# Unsupervised Disentanglement Without Compromises : How Functional Orthogonality Enforces Identifiability

**Mathieu Cyrille Simon** [1]   **Pascal Frossard** [2]   **Christophe De Vleeschouwer** [1]

## Abstract

This paper explores unsupervised disentangled representation learning from a functional perspective. We define latent concepts as factors that influence observations through locally orthogonal directions, formalized as an orthogonality constraint on the Jacobian of the generative mapping. We prove that this condition yields identifiability of general nonlinear generative models, without requiring statistical independence or causal assumptions, provided the latent domain admits all combinations of factor values. Experiments with orthogonality-regularized normalizing flows empirically confirm the theory, demonstrate reliable recovery of ground-truth factors, and shed light on the success of VAEs. These findings challenge the prevailing impossibility claims for unsupervised disentanglement and provide a principled alternative foundation.

## 1. Introduction

In *Disentangled Representation Learning* (DRL), the main assumption is that data is generated from a small number of underlying low-dimensional latent factors, often referred to as concepts. The objective is to recover a representation that separates these concepts, ideally in an unsupervised manner (Bengio et al., 2013; Wang et al., 2024). For instance, in facial images, underlying concepts might include hair length, hair color, or face orientation. But what exactly constitutes a concept? If unsupervised disentanglement is to be meaningful, it presupposes some form of intrinsic, human-free definition of the factors. Many prior works define concepts as statistically independent factors of variation (Higgins et al., 2017; Kim & Mnih, 2018; Chen et al., 2018) similarly to ICA (Lee, 1998; Oja et al., 2003). However, this assumption frequently fails to align with human intuition. Some semantically distinct concepts can be statistically dependent, while spurious correlations in the data may give rise to apparent dependencies between otherwise unrelated factors. To address these limitations, some approaches introduce causal modeling (Yang et al., 2020; Brehmer et al., 2022; Lippe et al., 2022; Xu et al., 2024; von Kügelgen et al., 2023; Lachapelle et al., 2022), treating factors as nodes in a causal graph. Yet, enforcing causal assumptions requires strong priors and may obscure the practical goal of capturing semantically meaningful factors. Often, the aim is not to recover the true causal structure, but rather to identify and represent the factors and their relationships, especially in cases where the causal link is unclear or difficult to justify. For example, while hair length and gender are clearly distinct concepts, they often exhibit statistical dependence, yet a direct causal link between them is not well-defined.

Thus, we propose an alternative perspective on what constitutes a concept. We argue that, although concepts may exhibit statistical dependence, they should be understood as factors that influence observations through distinct and independent variations, not in the statistical sense, but in terms of their functional effects on the generative process. This perspective aligns closely with the *Independent Causal Mechanisms* (ICM) principle (Schölkopf et al., 2021) from causal inference (Pearl, 2009; Peters et al., 2017), which posits that the causal generative process of a system is composed of autonomous modules that do not influence each other. Applied outside a causal context, it can be formalized as an orthogonality condition on the columns of the Jacobian of the generative function mapping latent factors to observation (Gresele et al., 2021). This captures a form of local decoupling in how each factor contributes to the observed data. Unlike prior works that treat this property as a sufficient condition to recover statistically independent sources (Buchholz et al., 2022; Gresele et al., 2021), we take a stronger stance: we argue that orthogonal influence is an *intrinsic defining property of meaningful concepts*. In contrast to these approaches, which rely on statistical independence and restricted functional families, we consider general nonlinear mixing functions with orthogonal Jacobians and explicitly allow for dependent latent factors. This perspective moves beyond the "Independent" in ICA and

[1]UCLouvain, ICTEAM, Louvain-la-Neuve, Belgium [2]EPFL, LTS4 laboratory, Lausanne, Switzerland. Correspondence to: Mathieu Cyrille Simon <mathieu.simon@uclouvain.be>.

*Proceedings of the 43rd International Conference on Machine Learning*, Seoul, South Korea. PMLR 306, 2026. Copyright 2026 by the author(s).

suggests that orthogonality, not independence, is the fundamental structural property underlying disentanglement.

Building on this framework, we show that *disentanglement emerges naturally from orthogonal influence*, without requiring statistical independence or supervision. This challenges the prevailing view that unsupervised disentanglement is fundamentally impossible, and demonstrates instead that meaningful disentanglement can arise *without compromise* provided that one adopts a functionally grounded notion of independence. We support our theoretical findings with empirical results and discuss implications for understanding why models such as Variational Autoencoders implicitly promote disentangled representations. More broadly, our work offers a principled step toward unifying disentanglement and causality under a common functional framework.

## 2. Related Works

Early approaches to disentangled representation learning were primarily inspired by *Independent Component Analysis* (ICA) (Lee, 1998; Oja et al., 2003), aiming to recover statistically independent factors of variation from observed data in an unsupervised manner (Kingma & Welling, 2013; Higgins et al., 2017). However, as formalized by (Locatello et al., 2019a), purely unsupervised disentanglement is fundamentally impossible without strong inductive biases. Specifically, there exist infinitely many equally valid latent representations that yield identical marginal distributions over observations, rendering the generative model non-identifiable. This impossibility statement mirrors classical findings in nonlinear ICA (Hyvärinen & Pajunen, 1999), where disentanglement has been proven to fail in the absence of additional assumptions or supervision (Hyvärinen et al., 2023; 2024; Zheng et al., 2022; Hyvarinen & Morioka, 2017). Consequently, subsequent works have sought to introduce suitable inductive biases, either through architectural constraints (Bouchacourt et al., 2018; Chen et al., 2016; Locatello et al., 2020; Eastwood et al., 2023; Simon et al., 2024), weighting terms (Chen et al., 2018; Kim & Mnih, 2018), or partial supervision via auxiliary variables (Locatello et al., 2019b; Mita et al., 2021; Hälvä & Hyvarinen, 2020; Gabbay et al., 2021; Hyvarinen & Morioka, 2016). Comparatively, in this paper, we focus on the fully unsupervised setting, emphasizing the need for a more principled notion of factors that goes beyond mere statistical independence as used previously and that would lead to an identifiable representation.

Several lines of research have explored alternative formulations of disentanglement through group theory (Zhu et al., 2021; Higgins et al., 2018), second-order Hessian-based penalties (Peebles et al., 2020), Jacobian regularization (Lezama, 2019) or directly encouraging orthogonality between latent directions (Wei et al., 2021; Cha & Thiya-

galingam, 2023). But, the first theoretical framework to formalize this idea of functional orthogonality was introduced by (Gresele et al., 2021), under the name *Independent Mechanism Analysis* (IMA). Subsequent works extended this framework to higher-dimensional latent spaces under the manifold hypothesis (Ghosh et al., 2023) and demonstrated that VAEs implicitly encourage IMA-like structure through their training objective (Reizinger et al., 2022; Allen, 2024), providing a partial explanation for their empirical disentangling ability despite the absence of explicit supervision. The original IMA formulation showed that enforcing orthogonality in the Jacobian eliminates spurious and degenerate ICA solutions in specific nonlinear settings. Later studies generalized these findings, proving identifiability results for restricted classes of transformations such as conformal mappings (Buchholz et al., 2022; Zheng et al., 2022) or local isometry (Horan et al., 2021). In contrast to these prior efforts, our work provides a comprehensive theoretical treatment of identifiability under the *general* case of orthogonal Jacobians, without restricting to specific functional families. Furthermore, whereas previous analyses assumed independent latent factors, we explicitly address the more realistic case of dependent concepts, thereby broadening the applicability of IMA to real-world generative processes and offering a unified framework for disentanglement beyond statistical independence.

## 3. Problem Formulation

We start by formalizing the data-generating process. We assume that observations $\mathbf{x} \in \mathbb{R}^n$ result from latent concepts $\mathbf{z} \in \mathbb{R}^d$, with $d \ll n$, that have undergone a smooth and invertible mapping $\mathbf{f} : \mathbb{R}^d \to \mathbb{R}^n$ which act as a mixing function

$$\mathbf{x} = \mathbf{f}(\mathbf{z}), \qquad p(\mathbf{z}) = \prod_{i=1}^{d} p(\mathbf{z}_i | \mathbf{z}_{<i}), \quad (1)$$

for *any* ordering of the components. Crucially, compared to *Blind Source Separation* (BSS) in (Gresele et al., 2021), we do not assume the underlying concepts $\mathbf{z}$ to be independent and their distribution might be modeled recursively by a triangular mapping from independent noise, $z = \phi(\epsilon)$, e.g. with the Darmois construction (Darmois, 1953; Hyvärinen & Pajunen, 1999). The goal is to learn an unmixing function $\hat{\mathbf{z}} = \mathbf{g}^{-1}(\mathbf{x})$ that would inverse $\mathbf{f}$ so that we recover the true concepts $\mathbf{z}$ up to some tolerable ambiguities. Formally, we adopt the definition of identifiability introduced in (Gresele et al., 2021).

**Definition 1.** Let $\mathcal{F}$ be the set of all smooth and invertible functions $\mathbf{f} : \mathbb{R}^d \to \mathbb{R}^n$ and $\mathcal{P}$ be the set of all densities $p_{\mathbf{z}}$ with simply connected support on $\mathrm{R}^d$. $\mathcal{M} \subseteq \mathcal{F} \times \mathcal{P}$ defines the subspace of models. The generative process is said to

be *identifiable* on $\mathcal{M}$ if

$$\forall (\mathbf{f}, \mathbf{z}), (\mathbf{g}, \hat{\mathbf{z}}) \in \mathcal{M}: \ \mathbf{f}_* p_{\mathbf{z}} = \mathbf{g}_* p_{\hat{\mathbf{z}}}$$
$$\Rightarrow \ \exists \mathbf{P}, \mathbf{t} \ \text{s.t.} \ (\mathbf{f}, \mathbf{z}) = (\mathbf{g} \circ \mathbf{P}^{-1} \circ \mathbf{t}^{-1}, (\mathbf{P} \circ \mathbf{t})_* \hat{\mathbf{z}}). \quad (2)$$

Where $\mathbf{f}_* p_{\mathbf{z}}$ denotes the pushforward of $p_{\mathbf{z}}$ through $\mathbf{f}$, $\mathbf{P}$ is an arbitrary permutation, and $\mathbf{t}$ is an element-wise reparameterization of the concepts.

Identifiability (Lehmann & Casella, 1998) expresses the idea that, under suitable assumptions, a learned representation should coincide with the true generative factors up to well-characterized ambiguities: a permutation and a dimension-wise reparameterization.Such ambiguities are acceptable as they preserve the disentangled structure of the representation. Since there is no canonical choice of coordinate system, different parameterizations remain equally valid.

Unfortunately, such identifiability cannot be achieved under the current formulation without further constraints. To illustrate this, consider the case where the latent variables $\mathbf{z}$ are generated from independent noise via an arbitrary function $\phi$, and subsequently mapped to observations via another bijection $\mathbf{f}$. In the absence of additional assumptions, any pair $(\phi, \mathbf{f})$ yielding the same distribution over $\mathbf{x}$ is equally valid. This means that $\mathbf{z}$ could be any arbitrary entangled transformation of the true underlying factors, with dependencies entirely absorbed by $\phi$. Consequently, the same observed distribution can be explained by many distinct and incompatible latent structures. The model is thus underdetermined. This lack of identifiability undermines the interpretability and meaning of the latent factors: if the ground-truth generative process is not uniquely recoverable, then the notion of concepts as fundamental explanatory variables loses its significance. To resolve this ambiguity, additional conditions are necessary to restrict the solution space and render the problem well-posed. We argue, following prior works (Buchholz et al., 2022; Gresele et al., 2021), that what is missing is a notion of independence, not in the statistical sense, but in terms of the functional influence that each factor exerts on the observations. Mathematically, we can formalize this as

**Assumption 1.** Let $\mathbf{J_f}$ denote the Jacobian of the generative function $\mathbf{f}$, and let $\partial \mathbf{f}/\partial z_i$ denote its $i$-th column. Each concept $z_i$ affects the observations through a distinct mode of variation:

$$\forall \mathbf{f} \in \mathcal{F}, \quad \forall i \neq j: \quad \frac{\partial \mathbf{f}}{\partial z_i}^T \frac{\partial \mathbf{f}}{\partial z_j} = 0, \quad (3)$$

i.e., the partial derivatives of $\mathbf{f}$ with respect to distinct latent variables are orthogonal at every point.

This orthogonality condition enforces that perturbations along different latent dimensions induce locally independent variations in the data space. It provides a geometric

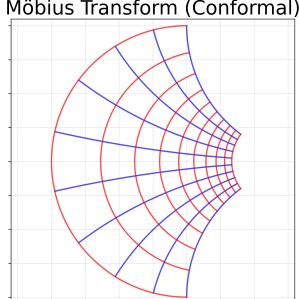 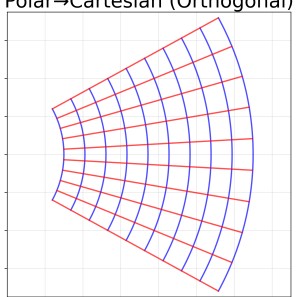

*Figure 1.* Grid Transformations: Visualizing Orthogonal Jacobians.

form of disentanglement by requiring that the effect of each latent factor on the observations be decoupled from the others. Importantly, this assumption applies to the entire model class $\mathcal{M}$ and is posited as a defining property of meaningful factors. In doing so, it shifts the inductive bias from statistical independence of the latent variables to independence of their functional contributions, which we argue is more aligned with the way humans intuitively define and distinguish concepts. Among the functions satisfying Asm.1, a notable subclass is formed by conformal maps, characterized by Jacobians of the form $\mathbf{J_f} = \lambda(\mathbf{z})\mathbf{Q}(\mathbf{z})$, where $\lambda : \mathbb{R}^d \to \mathbb{R}^+$ is a smooth scalar field and $\mathbf{Q}(\mathbf{z}) \in O(d)$ is an orthogonal matrix. Conformal maps preserve local angles. A visualization is provided in Fig.1.

Another way to understand this assumption is through the lens of generative invertibility. When the latent variables $\mathbf{z}$ are constructed recursively via a function $\phi$ from independent noise $\boldsymbol{\epsilon}$, the Jacobian of $\phi$ becomes lower triangular due to the autoregressive structure of the dependencies. In this setting, the full generative mapping from noise to observations is $\mathbf{x} = \mathbf{f} \circ \phi(\boldsymbol{\epsilon})$, and its Jacobian is the product $\mathbf{J_f} \cdot \mathbf{J}_\phi$. If $\mathbf{J}_\phi$ is lower triangular and $\mathbf{J_f}$ has orthogonal columns, then this product is reminiscent of a QR decomposition of an invertible matrix. Since the QR decomposition of an invertible matrix is unique up to signs (Golub & Van Loan, 2013), this analogy hints at a form of uniqueness in the factorization of the generative process. By encouraging orthogonality in $\mathbf{f}$, we effectively constrain the overall generative model in a way that helps disambiguate the latent representation. This supports the idea that the orthogonality of functional contributions is not merely a heuristic, but a theoretically grounded principle that facilitates identifiability in the absence of statistical independence.

## 4. Identifiability Conditions

We now turn to the central theoretical contribution of this work: establishing that the orthogonality constraint on the Jacobian of the generative function yields a uniquely recoverable generative model, up to the usual ambiguities. This

result highlights that orthogonality is not merely a convenient regularization, but rather a *fundamental property that defines what constitutes a factor in the generative process*. Without identifiability, multiple entangled latent representations can explain the same observed distribution, rendering the notion of "factors" devoid of semantic meaning. Demonstrating identifiability therefore establishes the conceptual and mathematical significance of orthogonal functional influence as the defining criterion for meaningful disentanglement. In the following, we analyze identifiability in two stages. We first consider the case where the latent factors are statistically independent. We then relax this assumption and extend the analysis to the more general and realistic case where the latent factors may exhibit statistical dependencies.

### 4.1. Identifiability Under the Independence Assumption

We first consider the case where the latent factors are statistically independent. This setting is closely related to classical BSS and ICA, where the goal is to recover the independent sources that generate the observed data. Here, we extend this principle to the nonlinear setting under the orthogonality constraint introduced in Asm.1.

**Proposition 1.** *Let the observed data* $\mathbf{x} \in \mathbb{R}^n$ *lie on a $d$-dimensional manifold generated by latent variables* $\mathbf{z} \in \mathbb{R}^d$, *with* $d \ll n$, *through a smooth and invertible mapping* $\mathbf{f} : \mathbb{R}^d \to \mathbb{R}^n$ *satisfying Asm.1. Each latent coordinate* $z_i$ *corresponds to a distinct underlying concept. Suppose further that the latent variables are statistically independent,* $p(\mathbf{z}) = \prod_i p(\mathbf{z}_i)$, *and that at most one component* $z_i$ *follows a Gaussian distribution. Then, the generative process* $(\mathbf{f}, p_{\mathbf{z}})$ *is identifiable in the sense of Def.1.*

The complete proof is provided in App.A. Intuitively, we can illustrate it in the special case where $\mathbf{f}$ is conformal and show that identifiability follows directly. Indeed, using the same notations as in Def.1, denote $\mathbf{h} = \mathbf{f}^{-1} \circ \mathbf{g}$ the function mapping $\hat{\mathbf{z}}$ to $\mathbf{z}$. Independence implies $\det \mathbf{Jh} = 1$ up to a coordinate-wise reparameterization. This, combined with the conformal structure of $\mathbf{h}$ (composition of conformal mappings), forces $\mathbf{h}$ to be both angle and volume preserving. Hence, $\mathbf{h}$ reduces to a rigid transformation (rotation and translation), yielding the same identifiability condition as in linear ICA. This provides a simple and direct proof of identifiability in the conformal case compared to (Buchholz et al., 2022) or (Zheng et al., 2022).

More generally, Prop.1 shows that the orthogonality constraint ensures identifiability even beyond the conformal case for the general orthogonal Jacobian. However, that simple proof reveals a conceptual link between our framework and classical ICA : both aim to recover latent directions that are statistically independent and Asm.1 naturally collapses the nonlinear problem to the identifiable linear ICA regime, recovering the condition that identifiability holds when at most one source is Gaussian. This is consistent with prior results in (Gresele et al., 2021) for restricted functional classes and explains why the same conclusions as ICA apply. In this sense, Asm.1 generalizes the linear orthogonality condition underlying ICA to the nonlinear regime. This connection also relates to Principal Component Analysis (PCA) (Abdi & Williams, 2010), where orthogonality plays a central role, the orthogonal Jacobian assumption can be interpreted as a local, nonlinear analogue of PCA, while statistical independence extends it globally to ICA.

Overall, this result goes beyond SoTA, as it directly challenges the widely accepted impossibility result which asserts that unsupervised disentanglement is unachievable (Hyvärinen & Pajunen, 1999; Locatello et al., 2019a). Our findings reveal that, when the generative process satisfies a functional independence condition, the generative process becomes identifiable even in the absence of supervision. This helps explain why methods such as VAEs and other architectures implicitly or explicitly enforcing local orthogonality often succeed empirically in disentangling latent factors, particularly on synthetic or structured datasets that directly satisfy the assumptions.

Finally, Prop.1 emphasizes the dual nature of the identifiability problem: (1) a **local** aspect, governed by the Jacobian orthogonality that determines how latent factors influence the observations; and (2) a **global** aspect, governed by the distribution $p(\mathbf{z})$ that constrains how these factors are statistically organized. Together, these complementary constraints define a principled pathway to achieving meaningful and identifiable disentanglement.

### 4.2. Identifiability Under Dependent Factors

In practice, and as discussed in the introduction, the assumption of statistically independent latent factors rarely holds, either due to inherent relationships or spurious correlations in the dataset. It is therefore essential to study identifiability when factors are potentially dependent. In this setting, the data $\mathbf{x}$ is still generated by a smooth and invertible mapping $\mathbf{f} : \mathbb{R}^d \to \mathbb{R}^n$ satisfying Asm.1, but the latent variables $\mathbf{z}$ no longer follow a factorized distribution. Instead, their joint distribution can be modeled recursively as $p(\mathbf{z}) = \prod_{i=1}^{d} p(z_i \mid z_{<i})$ which can equivalently be expressed through a triangular transformation $\phi : \mathbb{R}^d \to \mathbb{R}^d$ of independent noise variables $\boldsymbol{\epsilon}$, i.e., $\mathbf{z} = \phi(\boldsymbol{\epsilon})$. Importantly, we do not impose any causal interpretation on $\phi$; it merely serves as a generic mechanism to model dependent factors.

However, in this more general case, identifiability is no longer guaranteed. As established in Prop.1, identifiability arises from the interaction of two complementary constraints: a *local geometric constraint* and a *global statistical constraint*. When independence is removed, the global constraint disappears, leaving the model underdetermined.

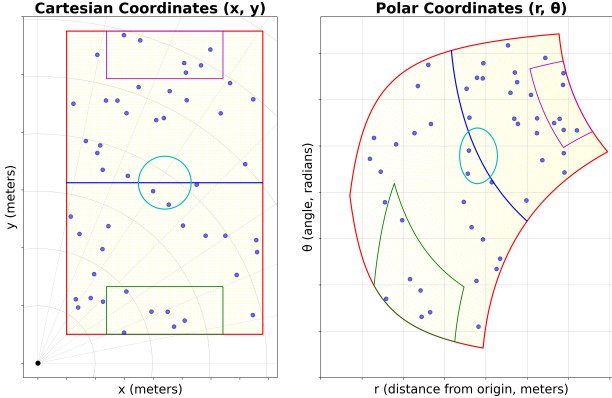

**Cartesian Coordinates (x, y)**     **Polar Coordinates (r, θ)**

*Figure 2.* Assume we aim to represent the position of a ball on a field as a two-dimensional latent variable. The most direct representation uses Cartesian coordinates $(x, y)$ (left), regardless of the distribution of positions. However, an equally valid alternative could be a polar representation $(r, \theta)$ (right). Without additional assumptions, there is no principled way to identify which representation is the "true" one: $(r, \theta)$ is simply an entangled reparameterization of $(x, y)$. Under an independence assumption, however, a privileged representation emerges; for example, if positions are uniformly distributed over the field, this constraint favors the Cartesian coordinates $(x, y)$. In the absence of such assumptions, these entangled representations are indistinguishable. Nevertheless, we argue that even without independence, certain representations are preferable. Specifically, those in which all combinations of factor values can in principle occur, as with the $(x, y)$ coordinates in this example (Asm.2).

Indeed, a trivial counterexample illustrates this failure: consider **f** as the identity mapping, which trivially satisfies the orthogonality condition. In this case, $\mathbf{x} = \mathbf{f}(\mathbf{z}) = \mathbf{z}$, and the overall generative model $\mathbf{x} = \mathbf{f} \circ \phi(\epsilon)$ can represent any data distribution $p(\mathbf{x})$ by appropriately choosing $\phi$ which can represent any distribution e.g., with the Darmois construction (Darmois, 1953). Thus, the model perfectly explains the data without uncovering any meaningful latent structure. This demonstrates that, in the absence of additional constraints, the concept of latent factors becomes vacuous: if multiple equally valid generative decompositions exist, then the notion of disentanglement loses its interpretability. An illustration of this degeneracy is provided in Fig.2.

A natural way to restore identifiability would be to incorporate auxiliary variables or weak supervision, such as labels or side information, to anchor the latent space and help align the learned axes with meaningful factors (Khemakhem et al., 2020). However, such information is not always available and introduces an undesirable degree of supervision.

These limitations motivate the search for a weaker, yet sufficiently informative, *global* constraint that can complement the *local* orthogonality condition. The key intuition is that orthogonality of the Jacobian already enforces strong geometric rigidity. Consequently, even limited global informa-

tion about the latent domain may suffice to fully determine the mapping. In the theory of conformal mappings for dimensions $d > 2$, Liouville's theorem (Blair, 2000) implies that any conformal transformation of $\mathbb{R}^d$ must be a Möbius transformation, i.e., a composition of translations, rotations, dilations, and inversions. When the data lies on a finite, simply connected domain $\Omega \subset \mathbb{R}^d$ that is not invariant under inversion (a property satisfied by a broad class of supports, including all polytopes and most bounded smooth domains) non-rigid Möbius transformations are ruled out. This implies that, *when the latent support is known, the generative process becomes identifiable, up to the natural rigid symmetries of that support*. Formal statements and illustrative examples are provided in App.B, where we also discuss the special case $d = 2$ via the Riemann Mapping Theorem.

Of course, such detailed geometric knowledge of the latent domain is rarely known a priori. Nevertheless, the above discussion suggests that identifiability can be recovered by introducing a structural assumption on the latent distribution that serves a similar purpose, thereby providing the required global constraint.

**Assumption 2.** The latent variables $\mathbf{z} \in \mathbb{R}^d$ follow a distribution $p_\mathbf{z}$ supported on a finite, simply connected region $\Omega \subset \mathbb{R}^d$ such that, up to a coordinate-wise reparameterization, $\Omega$ is a hypercube. Equivalently, the copula associated with $p_\mathbf{z}$ has full support on $[0, 1]^d$.

Intuitively, Asm.2 states that all combinations of latent factors are possible, that is, each latent coordinate can vary within its range, regardless of the values taken by the others. One may think of the latent dimensions as concept "sliders" that can be adjusted to produce every possible combination of factors. This implies that each factor corresponds to a distinct, manipulable degree of freedom. For example, consider the concepts of altitude and temperature. Although these quantities are correlated in nature, it is perfectly possible to observe configurations that break this dependency (e.g., a heated room at high altitude). The ability to realize or observe all such combinations is precisely what gives meaning to the notion of separate factors. Hence, Asm.2 can be viewed as a non-causal analog of interventions in causal models, ensuring that the dataset is rich enough to cover the joint range of the latent variables. A detailed discussion of both assumptions is provided in App.A.4.

Interestingly, if the factors are statistically independent and supported on a bounded domain, Asm.2 is fulfilled and thus this assumption is a direct generalization of the case of independent factors for bounded latent spaces. Critically, without such a global constraint, many different valid representations may satisfy the local functional independence enforced by an orthogonal Jacobian. Among these representations, however, there typically exists a *preferential* one that aligns with the combinatorial structure of the latent

domain; this phenomenon is illustrated in Fig.2. Using this conclusion, we obtain the following result.

**Proposition 2.** *Let the observed data* $\mathbf{x} \in \mathbb{R}^n$ *lie on a d-dimensional manifold generated by latent variables* $\mathbf{z} \in \mathbb{R}^d$, *with* $d \ll n$, *through a smooth and invertible mapping* $\mathbf{f} : \mathbb{R}^d \to \mathbb{R}^n$ *satisfying Asm.1. Suppose further that the latent distribution* $p_{\mathbf{z}}$ *satisfies Asm.2. Then, the generative process* $(\mathbf{f}, p_{\mathbf{z}})$ *is identifiable in the sense of Def.1.*

The proof is given in App.A. Together with the local orthogonality condition, Asm.2 ensures a unique alignment. This result can be interpreted as a general principle:

*Identifiability, and hence the very notion of a "concept" or "factor", emerges from the dual constraints that define it:*

1. **Local constraint (Assumption 1):** factors influence the data through orthogonal directions, capturing functional independence in the generative mechanism;
2. **Global constraint (Assumption 2):** the latent domain allows all possible factor combinations;each factor corresponds to a distinct manipulable degree of freedom.

Once these defining properties of latent factors are in place, identifiability follows naturally. This observation forms the central insight of our theoretical framework: *disentanglement is not an artifact of independence or causality, but a structural property emerging from the interplay between local orthogonality and global combinatorial completeness.*

We next turn to the empirical validation of these results.

## 5. Experiments

Theoretical Prop.1 and 2 establish that, under orthogonal functional influence, the generative model becomes identifiable. In this section, we empirically validate these results by investigating whether models trained solely on observations $\mathbf{x}$ can recover the underlying latent concepts $\mathbf{z}$ up to the admissible ambiguities specified in Def.1. Concretely, we compare models trained with and without an orthogonality constraint on the Jacobian of the generative mapping and evaluate their ability to disentangle latent factors. Details on the experiments are provided in App.C.

**Data Generation** : We consider two regimes for the latent variables $\mathbf{z} \in \mathbb{R}^d$, with $d \in \{3, 6, 9\}$ : (i) Independent factors, latent variables are sampled independently from a uniform distribution on $[0, 1]^d$ (ii) Dependent factors, latent variables are generated using a flexible *Normalizing Flow* (NF) (Kobyzev et al., 2020), that induces strong statistical dependencies while ensuring the resulting copula has full support on $[0, 1]^d$. This construction explicitly violates independence while satisfying Asm.2. In both cases, latent variables are mapped to observations via a smooth and invertible ground-truth mixing $\mathbf{f}$. We consider two classes of generative mappings: (i) conformal Möbius transformations;

and (ii) more general nonlinear mappings with orthogonal Jacobians of the form $\mathbf{J_f} = \mathbf{Q}(\mathbf{z})\mathbf{D}(\mathbf{z})$, obtained by composing conformal and non-conformal transformations. This second class goes strictly beyond conformal mappings and tests identifiability under general orthogonal Jacobians.

**Models** : To model the data distribution $p(\mathbf{x})$, we train expressive Residual Flows (Chen et al., 2019) with full Jacobians. We learn both *Unconstrained baselines* with no structural constraints on the Jacobian and *Orthogonally constrained models* using the $C_{\mathrm{IMA}}$ loss introduced by (Gresele et al., 2021) to regularize the maximum likelihood objective, enforcing Asm.1. In the independent setting, the base distribution is a standard Gaussian with diagonal covariance. In the dependent setting, the base distribution is itself learned using a RealNVP flow (Dinh et al., 2016) with a triangular Jacobian, ensuring a full-support copula while modeling dependencies between latent variables.

**Evaluation Metrics** : For evaluation, we compute (i) the KL divergence between the learned model and the true data-generating distribution to assess density estimation quality; (ii) the *Mean Correlation Coefficient* (MCC) between the recovered and ground-truth latent variables (Khemakhem et al., 2020); and (iii) a nonlinear extension of the Amari distance (Gresele et al., 2021) between the true mixing and the learned unmixing. The latter equals zero if and only if the model is identifiable in the sense of Def.1. As all models managed to fit the data and have equivalent KL divergence metrics we only report the MCC and Amari distance to measure disentanglement.

**Results** : Fig.3 reports MCC and Amari distances for unconstrained and orthogonally constrained models, across independent and dependent latent variables, for both conformal and general $\mathbf{QD}$ generative mappings. For each configuration, 30 independent runs are performed, yielding a total of 720 trained models. As expected, standard NFs consistently fail to recover the latent factors despite achieving good likelihoods, confirming that the problem remains underdetermined. In contrast, introducing the orthogonality constraint leads to substantial improvements in both MCC and Amari distance, often approaching perfect identifiability. These results empirically validate Prop.1 and 2. Notably, while prior work largely focused on conformal mappings, our results demonstrate that identifiability extends to significantly more general nonlinear transformations with orthogonal Jacobians. Strikingly, performance is also comparable in the independent and dependent settings. This provides strong empirical evidence that disentanglement does not fundamentally rely on statistical independence or causal assumptions. Instead, meaningful and identifiable representations of dependent concepts can be recovered purely from orthogonal functional influence, supporting the central thesis of this work. Additional results and qualitative visual-

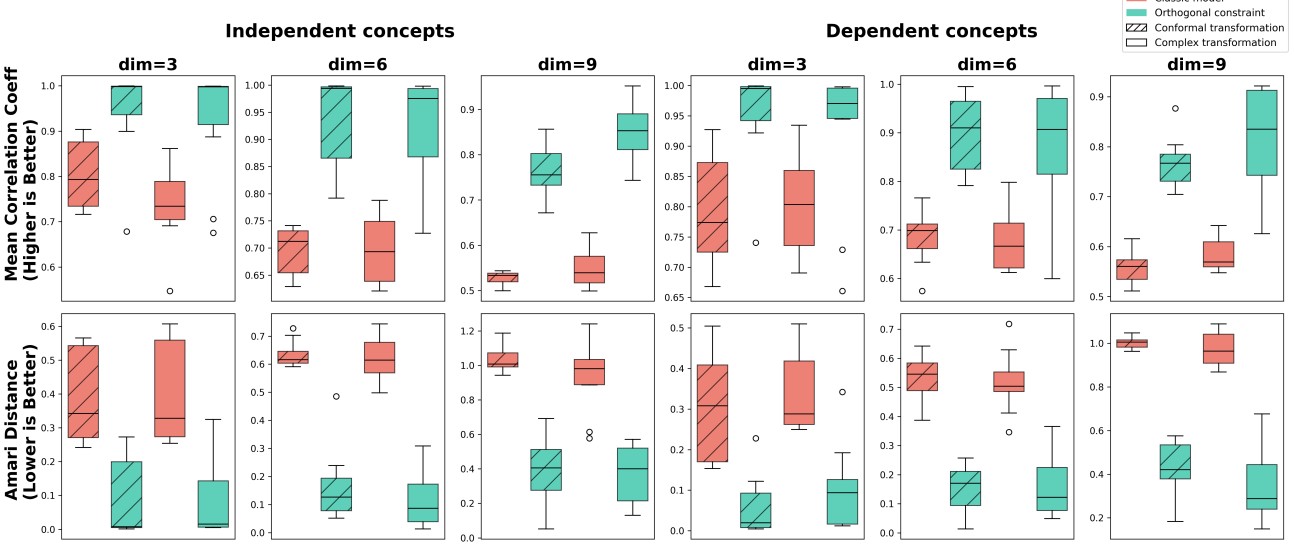

*Figure 3.* MCC and Amari distances quantitative results for $d \in \{3, 6, 9\}$ in the independent/dependent cases.

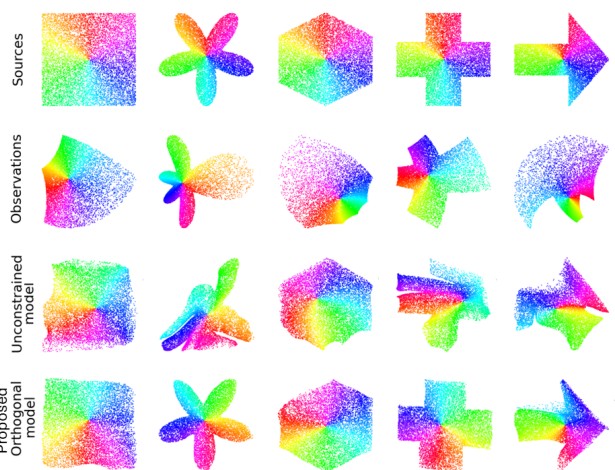

*Figure 4.* Qualitative results under various sources domain.

izations are provided in App.D, including comparison with VAEs across standard disentanglement metrics.

Finally, Fig.4 presents some qualitative results illustrating the role of global domain information. We generate data with varying latent supports and train a Neural Spline Flow (Durkan et al., 2019) equipped with orthogonal Jacobian constraints. Consistent with the discussion in Sec.4.2, the learned representations are identifiable up to the rigid symmetries of the latent domain when its structure is known. This behavior aligns with the theoretical analysis in App.B and further illustrates the interplay between local orthogonality and global constraints.

Overall, these experiments demonstrate that enforcing orthogonal functional influence is sufficient to enable unsu-

pervised disentanglement. The empirical findings closely match the theoretical predictions, providing strong support for the proposed notion of concepts.

# 6. Discussion and Perspectives

We now discuss some broader implications of our findings and outline several promising directions for future work.

**Causal Representation** : Throughout the paper, we have drawn connections to causal representation learning, whose aim is to recover latent factors together with their causal structure. However, in many applications, enforcing a full causal model is often unrealistic: the causal relations between factors are unclear or not identifiable from observational data alone. Our results show that meaningful factors can nevertheless be recovered by leveraging causally inspired principles, such as ICM, without committing to an explicit causal graph. This suggests an alternative perspective on causal representation learning, in which causal assumptions are relaxed to prioritize factor recovery. If causal structure is ultimately required, it can be inferred *after* disentanglement, using additional assumptions, domain knowledge, or limited interventions, effectively deferring causal resolution to a second stage. This "best effort with the available data" philosophy aligns with the practical reality of many machine-learning applications, where full causal supervision is rarely accessible. Overall, our results emphasize that it is neither statistical independence nor causal directionality that fundamentally defines a latent factor.

Importantly, our generative model remains compatible with causal analysis when such structure is desired. In our framework, the observation map **f** has an orthogonal Jacobian,

while dependencies between latent variables are modeled via a triangular transformation $\phi$. In this setting, results by Xi et al. (2023) imply that such models recover representations up to the Markov Equivalence Class. Consequently, our approach can be viewed as a strong starting point for causal discovery, with causal ordering potentially resolvable using only limited interventional information.

**VAE Disentanglement** : Finally, we revisit *Variational Autoencoders* (VAEs) through the lens of our framework. VAEs are among the most widely used models for unsupervised disentanglement. A VAE learns a latent variable model by jointly optimizing a reconstruction term, $-\log p(x \mid z)$, and a regularization term enforcing the approximate posterior $q(z \mid x)$ to match a factorized Gaussian prior $p(z) = \mathcal{N}(0, I)$. A well-established belief is that the factorized prior is the primary mechanism responsible for disentanglement. Works such as (Chen et al., 2018; Kim & Mnih, 2018) formalize this viewpoint by decomposing the evidence lower bound (ELBO) as

$$\mathcal{L}_{\text{VAE}} = -\log p(x \mid z) + \text{KL}(q(z \mid x) \| p(z)) \qquad (4)$$

$$= \underbrace{-\log p(x \mid z)}_{\text{Reconstruction}} + \underbrace{\text{KL}\Big(q(z) \| \prod_j q(z_j)\Big)}_{\text{Total Correlation}}$$

$$+ \underbrace{\sum_j \text{KL}(q(z_j) \| p(z_j))}_{\text{Dimension-wise KL}} + \underbrace{I(x; z)}_{\text{Mutual Information}} . \qquad (5)$$

The dimension-wise KL encourages each latent dimension to match a univariate Gaussian, hence simply affecting the reparameterization, and the mutual-information term is undesirable. In this decomposition, the total correlation term penalizes statistical dependence between latent coordinates and is therefore viewed as the component driving disentanglement. However, this explanation is incomplete. Indeed, we show here below that models such as Normalizing Flows share exactly the same total-correlation penalty while they do **not** generally produce disentangled representations.

Consider a NF, which learns an invertible mapping $\mathbf{h}$ s.t.

$$\log p(x) = \log p(z) + \log\big|\det(\mathrm{d}h(x)/\mathrm{d}x)\big|, \qquad (6)$$

with $p(z) = \mathcal{N}(0, I)$. Since the pushforward distribution $q(z)$ induced by $h$ does not necessarily match $p(z)$, training effectively minimizes cross-entropy:

$$\mathcal{L}_{\text{NF}} = -\log q(z) + \text{KL}(q(z) \| p(z)) - \log\big|\det(J)\big| \quad (7)$$

$$= -\log p(x) + \underbrace{\text{KL}\Big(q(z) \| \prod_j q(z_j)\Big)}_{\text{Total Correlation}}$$

$$+ \underbrace{\sum_j \text{KL}(q(z_j) \| p(z_j))}_{\text{Dimension-wise KL}} . \qquad (8)$$

Thus, flows exhibit the same global independence pressure as VAEs. Yet, in practice, flows remain non-disentangled

in their standard form (Dinh et al., 2016; Zhai et al., 2024). This discrepancy highlights that factorized priors, or equivalently, total-correlation minimization, are insufficient to explain VAE's natural bias toward disentanglement.

The crucial missing ingredient is the *factorized approximate posterior* which is often overlooked. VAEs simultaneously enforce:

1. a factorized prior $p(z)$, promoting independence across data points (global structure), and
2. a diagonal-covariance approximate posterior $q(z \mid x) = \mathcal{N}(\mu(x), \sigma(x)^2 I)$, promoting locally axis-aligned variations within each data point, hence orthogonality.

The fact that the factorized posterior implicitly induces an orthogonal Jacobian of the encoder has been formally proven in (Reizinger et al., 2022; Allen, 2024). Consequently, disentanglement in VAEs arises not merely from independence constraints but from the *joint* imposition of (i) global independence via the prior and (ii) local orthogonality via the posterior. This dual constraint aligns precisely with the structure promoted by our framework. It also explains why VAEs succeed on datasets with well-separated generative factors, typically synthetic data, and fail on more complex settings, e.g., with dependent structure.

To further validate this dual-mechanism explanation, we design additional experiments on standard image disentanglement benchmarks, namely dSprites (Matthey et al., 2017) and 3DShapes (Burgess & Kim, 2018). In Sec.5, we considered models without orthogonality bias (NFs) and showed that adding this bias improves disentanglement. Here, we follow the complementary approach: we start from a model that includes both independence and orthogonality (a VAE), and we remove the orthogonality bias to evaluate its effect.

The mechanism by which VAEs implicitly enforce Jacobian orthogonality has been analyzed in prior work (Reizinger et al., 2022). In simplified terms, the factorized Gaussian posterior enforces a row-orthogonal encoder Jacobian, and in the near-deterministic regime the decoder approximately inverts the encoder, resulting in a column-orthogonal decoder Jacobian. Based on this observation, we construct a variant of the $\beta$-VAE where we replace the diagonal Gaussian posterior with a NF posterior. This is a standard technique in the VAE literature to increase expressiveness and better match the prior, thereby improving density modeling (Kingma et al., 2016). However, unlike the factorized Gaussian case, this more expressive posterior no longer enforces a diagonal covariance structure, and thus removes the implicit Jacobian orthogonality constraint. Importantly, this change leaves the overall objective unchanged, particularly the total-correlation/independence pressure imposed by the prior, thereby isolating the effect of removing the orthogo-

*Table 1.* Disentanglement performance averaged over 10 random seeds across datasets and models.

| Dataset | Metric | Standard VAE models W/ Ortho. J. | | | W/O Ortho. J. |
|---------|--------|--------|--------|--------|--------|
| | | $\beta$-VAE | FactorVAE | $\beta$-TCVAE | $\beta$-FlowVAE |
| dSprites | DCI dis. | $0.1879 \pm 0.0967$ | $0.2527 \pm 0.0617$ | $0.2963 \pm 0.0905$ | $0.0975 \pm 0.0307$ |
| | MIG | $0.1070 \pm 0.0706$ | $0.1643 \pm 0.0654$ | $0.1783 \pm 0.0689$ | $0.0443 \pm 0.0194$ |
| | SAP | $0.0465 \pm 0.0338$ | $0.0673 \pm 0.0119$ | $0.0663 \pm 0.0141$ | $0.0195 \pm 0.0062$ |
| | $\beta$-VAE score | $0.8238 \pm 0.0758$ | $0.8548 \pm 0.0225$ | $0.8628 \pm 0.0345$ | $0.7329 \pm 0.0665$ |
| | FactorVAE score | $0.6523 \pm 0.1018$ | $0.7354 \pm 0.0973$ | $0.7312 \pm 0.1204$ | $0.6020 \pm 0.0997$ |
| 3DShapes | DCI dis. | $0.5349 \pm 0.3034$ | $0.6762 \pm 0.0847$ | $0.6539 \pm 0.3033$ | $0.1844 \pm 0.0386$ |
| | MIG | $0.2279 \pm 0.1791$ | $0.2977 \pm 0.1344$ | $0.2899 \pm 0.2412$ | $0.0617 \pm 0.0215$ |
| | SAP | $0.0662 \pm 0.0460$ | $0.0707 \pm 0.0320$ | $0.0811 \pm 0.0490$ | $0.0265 \pm 0.0112$ |
| | $\beta$-VAE score | $0.9675 \pm 0.0621$ | $0.9477 \pm 0.0741$ | $0.9756 \pm 0.0520$ | $0.8031 \pm 0.0596$ |
| | FactorVAE score | $0.8386 \pm 0.1014$ | $0.8234 \pm 0.0676$ | $0.8725 \pm 0.1403$ | $0.7142 \pm 0.0834$ |

nality bias while preserving the independence mechanism.

We implement this model following (Locatello et al., 2019a) and reproduce standard $\beta$-VAE results. We then train the exact same model, with identical architecture and training procedure, but using a flow-based posterior (denoted $\beta$-flowVAE). The results are reported in Tab.1. If orthogonality were not important, this more expressive model should yield improved disentanglement, since it better matches the prior hence improving independence. However, we observe a systematic substantial decrease in disentanglement performance. The key difference is precisely the removal of the orthogonality bias. This provides direct empirical evidence that independence alone is not sufficient, and that orthogonality plays a crucial role, further confirming our claims on the sources of disentanglement.

**Beyond VAE's** : Although VAEs remain the canonical model for disentanglement, they suffer from the well-documented *information preference* or *posterior collapse* phenomenon (Chen et al., 2016; Zhao et al., 2017). Intuitively, due to the explicit minimization of the mutual information $I(x; z)$ in Eq.4, VAEs trade off expressive reconstruction against informative latent representations, often resulting in blurry reconstructions, weak generative fidelity, and latent variables that carry minimal semantic signal.

Our analysis shows that disentanglement relies on independence and orthogonality constraints, not on the VAE objective per se. This opens the door to transplanting these inductive biases into alternative generative models that avoid VAE-specific limitations. In Fig.3, we demonstrated that NFs, which do not naturally produce disentangled representations, can be endowed with disentanglement capabilities simply by augmenting them with an orthogonal regularization. Unlike VAEs, flows optimize exact likelihoods and do not include a mutual information penalty in Eq.7, thereby avoiding the same reconstruction–information trade-off.

This perspective reinforces the relevance of recent approaches such as orthogonality regularized GANs (Wei et al., 2021) or diffusion models (Chen et al., 2024; Po et al., 2024). Our framework provides a theoretical justification for why these methods succeed. We argue that orthogonality should be viewed as a fundamental design principle for concept-structured and disentangled representations. By clarifying the mechanisms through which disentanglement arises, our framework illuminates how to systematically design new models that embody these principles. Overall, these insights indicate a promising direction for future research aimed at developing principled generative models with inherently disentangled representations. In particular, developing scalable methods to enforce or approximate Jacobian orthogonality efficiently in high-dimensional settings remains an important open problem, and a natural next step toward making the theoretical guarantees established in this work practically applicable at scale.

## 7. Conclusion

We showed that disentanglement can be achieved without statistical independence or causal supervision by defining concepts through orthogonal functional influence. Under this notion, nonlinear generative models with orthogonal Jacobians are identifiable up to standard ambiguities, even with dependent latent factors, extending prior identifiability results to the general orthogonal case. This framework explains the empirical disentanglement observed in models such as VAEs, clarifies why likelihood-based models fail without geometric constraints, and establishes orthogonality as a fundamental principle for identifiable and meaningful representation learning.

## Impact Statement

This paper presents work whose goal is to advance the field of Machine Learning. There are many potential societal consequences of our work, none which we feel must be specifically highlighted here.

## Acknowledgments

Mathieu Cyrille Simon is a Research Fellow of the Fonds de la Recherche Scientifique - FNRS of Belgium. Computational resources have been provided by the supercomputing facilities of the Université catholique de Louvain (CISM/UCL) and the Consortium des Équipements de Calcul Intensif en Fédération Wallonie Bruxelles (CÉCI) funded by the Fonds de la Recherche Scientifique de Belgique (F.R.S.-FNRS) under convention 2.5020.11 and by the Walloon Region

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

# Appendix

# A. Proofs and Detailed Derivations

This appendix provides the formal proofs of the main theoretical results stated in Sec.4. We begin by introducing auxiliary results that will be used throughout the proofs. We then establish identifiability under the independence assumption (Prop.1) and discuss its connection to classical Independent Component Analysis (ICA). Finally, we present the proof of identifiability under dependent latent factors (Prop.2) and highlight the key differences with respect to the independent case.

Throughout this appendix, as specified in Def.1, all functions are assumed to be sufficiently smooth, and all mappings are assumed to be invertible on their respective domains unless stated otherwise.

### A.1. Preliminary Results

**Preliminary.**   Let $\psi : [0,1]^d \to [0,1]^d$ be a smooth and invertible diffeomorphism that maps the hypercube onto itself and sends boundary points to boundary points. for every $u \in [0,1]^d$, the Jacobian of $\psi$ admits a factorization

$$J_\psi(u) = D_1(u)\, R(u)\, D_2(u), \tag{9}$$

where $D_1(u)$ and $D_2(u)$ are invertible diagonal matrices and $R(u) \in O(d)$. Additionally, the Jacobian field $J_\psi$ satisfies the integrability condition corresponding to a globally defined smooth mapping. Then there exists a signed permutation matrix $P$ and smooth, strictly monotone functions $\varphi_i : [0,1] \to [0,1]$ such that

$$\psi(u_1,\ldots,u_d) = P\big(\varphi_1(u_1),\ldots,\varphi_d(u_d)\big). \tag{10}$$

In particular, up to a fixed signed permutation of coordinates, $\psi$ is a global coordinate-wise (unitwise) reparametrization of the hypercube.

**Proof.**   By construction, the Jacobian of $\psi$ admits, at every point $u \in [0,1]^d$, the factorized form

$$J_\psi(u) = D_1(u)\, R(u)\, D_2(u), \tag{11}$$

where $D_1(u)$ and $D_2(u)$ are invertible diagonal matrices, and $R(u) \in O(d)$ is an orthogonal matrix. This decomposition holds smoothly on the entire domain, including its boundary. In particular, for each boundary point $u \in \partial[0,1]^d$, there exists a neighborhood on which the Jacobian preserves this diagonal–orthogonal–diagonal structure.

Let $F_i^{\pm}$ denote the $(d-1)$-dimensional faces of the hypercube defined by

$$F_i^- = \{u \in [0,1]^d \mid u_i = 0\}, \qquad F_i^+ = \{u \in [0,1]^d \mid u_i = 1\}. \tag{12}$$

For any point $p \in F_i^{\pm}$, the image $\psi(p)$ lies on the boundary $\partial[0,1]^d$, and hence belongs to some face $F_k^{\pm}$ for a certain index $k \in \{1, \ldots, d\}$. The tangent space at $p$ to the face $F_i^{\pm}$ is given by

$$T_p F_i^{\pm} = \mathrm{span}\{e_j \mid j \neq i\}, \tag{13}$$

where $\{e_1, \ldots, e_d\}$ is the canonical basis of $\mathbb{R}^d$. Since $\psi$ maps $\partial[0,1]^d$ onto itself, its differential must map tangent directions along a face to tangent directions of the image face. Equivalently, the differential maps the normal vector to a face to a vector normal to the image face, up to a nonzero scalar factor.

Let $\mathbf{n}_i = \pm e_i$ denote the unit normal vector to $F_i^{\pm}$ at $p$. Then

$$J_\psi(p)\,\mathbf{n}_i = D_1(p)\,R(p)\,D_2(p)\,\mathbf{n}_i = \lambda_i\,D_1(p)\,R(p)\,e_i, \tag{14}$$

for some nonzero scalar $\lambda_i$ corresponding to the $i$th diagonal entry of $D_2(p)$. Because the image of $\mathbf{n}_i$ must be normal to the image face $F_k^{\pm}$, the vector $R(p)e_i$ must be colinear with $\pm e_k$. As $R(p)$ is orthogonal, this implies that its $(k,i)$-entry satisfies $R_{ki}(p) = \pm 1$, and all other elements in the $i$th column and $k$th row of $R(p)$ vanish. Hence, locally, $R(p)$ acts as a signed permutation matrix exchanging coordinates $i$ and $k$ (possibly flipping their orientation).

Since $\psi$ (and therefore $R(u)$) is continuous, and the set of signed permutation matrices is discrete, the local signed permutation structure of $R(u)$ must remain constant on each connected component of the boundary where it is defined. As each face $F_i^{\pm}$ of the hypercube is connected, this implies that $\psi$ maps each face $F_i^{\pm}$ diffeomorphically onto a single face $F_k^{\pm}$, with the correspondence between indices $(i,k)$ determined by a fixed signed permutation.

To simplify the notation, we now restrict our attention to the three-dimensional case ($d = 3$); the argument extends verbatim to arbitrary dimensions. Moreover, without loss of generality, we assume that the signed permutation describing the correspondence between boundary faces is the identity. This assumption merely fixes the coordinate labeling and does not affect the generality of the proof, since any other signed permutation can be factored out and reintroduced at the end of the argument.

Let $F_1$, $F_2$, and $F_3$ denote the three faces of the cube $[0,1]^3$ intersecting at the origin, so that each face $F_i$ is orthogonal to the coordinate axis $e_i$, with outward normal vector $\mathbf{n}_i = -e_i$. The pairwise intersections of these faces correspond to the coordinate axes, and their triple intersection corresponds to the origin.

As established in the previous section (Eq.14), the Jacobian matrix of $\psi$ restricted to each face must preserve the tangent directions of that face. For $u \in F_1$, $F_2$, or $F_3$, the Jacobian therefore admits the following face-specific structures:

$$J_{\psi|F_1}(u) = \begin{pmatrix} d_{1,1}(u) & 0 & 0 \\ 0 & d_{1,2}(u) & 0 \\ 0 & 0 & d_{1,3}(u) \end{pmatrix} \begin{pmatrix} 1 & 0 & 0 \\ 0 & a & b \\ 0 & c & d \end{pmatrix} \begin{pmatrix} d_{2,1}(u) & 0 & 0 \\ 0 & d_{2,2}(u) & 0 \\ 0 & 0 & d_{2,3}(u) \end{pmatrix}, \tag{15}$$

$$J_{\psi|F_2}(u) = \begin{pmatrix} d_{1,1}(u) & 0 & 0 \\ 0 & d_{1,2}(u) & 0 \\ 0 & 0 & d_{1,3}(u) \end{pmatrix} \begin{pmatrix} a & 0 & b \\ 0 & 1 & 0 \\ c & 0 & d \end{pmatrix} \begin{pmatrix} d_{2,1}(u) & 0 & 0 \\ 0 & d_{2,2}(u) & 0 \\ 0 & 0 & d_{2,3}(u) \end{pmatrix}, \tag{16}$$

$$J_{\psi|F_3}(u) = \begin{pmatrix} d_{1,1}(u) & 0 & 0 \\ 0 & d_{1,2}(u) & 0 \\ 0 & 0 & d_{1,3}(u) \end{pmatrix} \begin{pmatrix} a & b & 0 \\ c & d & 0 \\ 0 & 0 & 1 \end{pmatrix} \begin{pmatrix} d_{2,1}(u) & 0 & 0 \\ 0 & d_{2,2}(u) & 0 \\ 0 & 0 & d_{2,3}(u) \end{pmatrix}. \tag{17}$$

At the pairwise intersections of these faces (that is, along the coordinate axes) and at their triple intersection (the origin), the orthogonal component necessarily satisfies $R(u) = I_3$. Consequently, the Jacobian at these intersection points simplifies to a purely diagonal form:

$$J_{\psi|\text{intersection}}(u) = \begin{pmatrix} 1 & 0 & 0 \\ 0 & 1 & 0 \\ 0 & 0 & 1 \end{pmatrix} \begin{pmatrix} d_1^*(u) & 0 & 0 \\ 0 & d_2^*(u) & 0 \\ 0 & 0 & d_3^*(u) \end{pmatrix}, \tag{18}$$

where $d_i^*(u)$ denote the diagonal entries of the product of $D_1(u)$ and $D_2(u)$. If the global signed permutation of faces were not the identity, the same conclusion would hold up to a fixed permutation and possible sign changes of these diagonal components.

While we must validate the orthogonal shape of the jacobian constraint everywhere, there is also another constraint that must be validated by the mapping. For $\psi$ to be globally realizable as a smooth diffeomorphism, the Jacobian field $J_\psi(u)$ must satisfy the standard *integrability condition*, which ensures that it corresponds to the gradient of a smooth mapping. Specifically, for each $i, j, k \in \{1, 2, 3\}$, the mixed partial derivatives must commute:

$$\frac{\partial^2 \psi_i}{\partial u_j\, \partial u_k} = \frac{\partial^2 \psi_i}{\partial u_k\, \partial u_j}. \tag{19}$$

When the second partial derivatives are continuous, this follows from Schwarz's theorem, which implies that the Hessian matrices of the coordinate functions of $\psi$ are symmetric. In differential form, this condition can be equivalently expressed as

$$\frac{\partial J_\psi}{\partial u_j} e_k = \frac{\partial J_\psi}{\partial u_k} e_j, \tag{20}$$

ensuring that the curl of each row of $J_\psi$ vanishes.

This integrability constraint imposes compatibility relations among the entries of $D_1(u)$, $D_2(u)$, and $R(u)$, ensuring that the Jacobian structure defined above is not only pointwise valid but also arises from a globally consistent smooth mapping $\psi$.

At the corner of the cube, where all three faces intersect, we have already established that $R(u) = I_3$. Furthermore, since $R(u)$ remains the identity along each pairwise intersection of faces, its value is fixed and constant in all directions emanating from the corner. Applying the integrability condition in this setting thus constrains the diagonal components of the Jacobian. Writing

$$J_{\psi|\text{corner}}(u) = \begin{pmatrix} d_1^*(u) & 0 & 0 \\ 0 & d_2^*(u) & 0 \\ 0 & 0 & d_3^*(u) \end{pmatrix}, \tag{21}$$

the mixed partial symmetry condition $\partial_{u_i} \partial_{u_j} \psi_k = \partial_{u_j} \partial_{u_i} \psi_k$ implies that cross-derivatives of distinct coordinates must vanish. Consequently, each diagonal entry $d_i^*(u)$ can depend only on its corresponding coordinate $u_i$, yielding

$$\begin{pmatrix} d_1^*(u) & 0 & 0 \\ 0 & d_2^*(u) & 0 \\ 0 & 0 & d_3^*(u) \end{pmatrix} = \begin{pmatrix} d_1^*(u_1) & 0 & 0 \\ 0 & d_2^*(u_2) & 0 \\ 0 & 0 & d_3^*(u_3) \end{pmatrix}. \tag{22}$$

Hence, near the corner, $\psi$ reduces to a *unitwise reparametrization* of the coordinates, meaning that each coordinate function $\psi_i$ depends solely on its corresponding input variable $u_i$. Showing how the integrability condition affects the possible mapping at the corners of the hypercube and thus the importance of this condition.

We now apply the integrability condition to the restriction of $\psi$ on a boundary face, for instance $F_1 = \{u_1 = 0\}$. On this face, the Jacobian admits the form

$$J_{\psi|F_1}(u) = \begin{pmatrix} d_{1,1}(u) & 0 & 0 \\ 0 & d_{1,2}(u) & 0 \\ 0 & 0 & d_{1,3}(u) \end{pmatrix} \begin{pmatrix} 1 & 0 & 0 \\ 0 & a & b \\ 0 & c & d \end{pmatrix} \begin{pmatrix} d_{2,1}(u) & 0 & 0 \\ 0 & d_{2,2}(u) & 0 \\ 0 & 0 & d_{2,3}(u) \end{pmatrix}, \tag{23}$$

where $R(u) = \begin{pmatrix} 1 & 0 & 0 \\ 0 & a & b \\ 0 & c & d \end{pmatrix}$ acts as the local orthogonal component on $F_1$. By construction, $R(u)$ fixes both its first column and its first row across the entire face, reflecting the fact that the normal direction $e_1$ is invariant under $\psi$ along $F_1$.

Since $R(u)$ is constant with respect to $u_2$ and $u_3$ in its first column and row, differentiating $J_{\psi|F_1}$ along the tangent directions $u_2$ and $u_3$ does not alter the components corresponding to $e_1$. The integrability condition,

$$\frac{\partial J_\psi}{\partial u_i} e_j = \frac{\partial J_\psi}{\partial u_j} e_i, \qquad \forall\, i, j \in \{1, 2, 3\}, \tag{24}$$

then enforces that the mixed derivatives involving $u_1$ and any tangent direction to $F_1$ vanish in the corresponding entries of $J_\psi$. In particular, the entries associated with the tangent scaling factors $(d_{1,2}, d_{1,3}, d_{2,2}, d_{2,3})$ must satisfy

$$\frac{\partial d_{1,2}}{\partial u_1} = \frac{\partial d_{1,3}}{\partial u_1} = \frac{\partial d_{2,2}}{\partial u_1} = \frac{\partial d_{2,3}}{\partial u_1} = 0. \tag{25}$$

Consequently, these quantities cannot depend on $u_1$, i.e.

$$d_{1,2}(u) = d_{1,2}(u_2, u_3), \quad d_{1,3}(u) = d_{1,3}(u_2, u_3), \tag{26}$$

$$d_{2,2}(u) = d_{2,2}(u_2, u_3), \quad d_{2,3}(u) = d_{2,3}(u_2, u_3). \tag{27}$$

Geometrically, this expresses the fact that along the face $F_1$, variations parallel to the face (in $u_2$ and $u_3$) occur independently of the normal coordinate $u_1$, since the orthogonal component $R(u)$ preserves the orientation of the normal direction. The integrability constraint thus enforces that the scaling along tangent directions to a face cannot depend on the coordinate normal to that face.

The foregoing discussion applies mutatis mutandis to every face, so that analogous conclusions hold for $F_2$ and $F_3$. In particular, on the pairwise intersection $F_1 \cap F_3$ we have (recall that $R = I$ on intersections)

$$J_{\psi|F_1 \cap F_3}(u) = I_3 \begin{pmatrix} d_1^*(u) & 0 & 0 \\ 0 & d_2^*(u) & 0 \\ 0 & 0 & d_3^*(u) \end{pmatrix} = I_3 \begin{pmatrix} d_1^*(u_1, u_2) & 0 & 0 \\ 0 & d_2^*(u_2) & 0 \\ 0 & 0 & d_3^*(u_2, u_3) \end{pmatrix}, \tag{28}$$

where the displayed functional dependencies record the constraints obtained from applying integrability on the adjacent faces. Thus, when one moves along the intersection curve (which is parametrized by $u_2$), the mapping acts as a unitwise reparametrization in the $u_2$–direction; perpendicular variations may still modify the scalings $d_1^*$ and $d_3^*$ through their explicit dependence on $u_2$. At the triple intersection (the corner) this specialization reduces to the purely unitwise reparametrization obtained earlier, which is coherent with what we obtained earlier.

Consider now, at that intersection $F_1 \cap F_3$, parameterized by the coordinate $u_2$, along which the Jacobian takes the simplified form (28), the integrability condition in the $e_3$-direction on this one-dimensional curve, namely

$$\frac{\partial J_\psi}{\partial u_2} e_3 = \frac{\partial J_\psi}{\partial u_3} e_2. \tag{29}$$

Because on $F_1 \cap F_3$ we have $R(u) = I_3$ (by construction) and $J_\psi(u) = D^*(u)$ is diagonal, we can explicitly compute for the left-hand side,

$$\frac{\partial J_\psi}{\partial u_2} e_3 = \frac{\partial}{\partial u_2} \begin{pmatrix} 0 \\ 0 \\ d_3^*(u_2, u_3) \end{pmatrix} = \begin{pmatrix} 0 \\ 0 \\ \dfrac{\partial d_3^*(u_2, u_3)}{\partial u_2} \end{pmatrix}. \tag{30}$$

For the right-hand side, differentiation is taken along the $u_3$-direction, which is tangent to the face $F_1$. In this case, $R(u)$ retains its first row and column fixed (as established earlier), but the remaining entries of $R(u)$ may vary smoothly with $u_3$. Thus we obtain

$$\frac{\partial J_\psi}{\partial u_3} e_2 = \frac{\partial}{\partial u_3} \big( R(u) D^*(u) \big) e_2 = \frac{\partial R(u)}{\partial u_3} D^*(u) e_2 + \frac{\partial D^*(u)}{\partial u_3} e_2. \tag{31}$$

The second term in (31) vanishes when evaluated on $e_2$ (because $d_2^*(u)$ depends only on $u_2$ along the intersection). Hence, we are left with

$$\frac{\partial J_\psi}{\partial u_3} e_2 = \frac{\partial R(u)}{\partial u_3} d_2^*(u_2) e_2, \tag{32}$$

Since $R(u)$ fixes $e_1$ and its first row/column and is fixed in $u_2$, $\frac{\partial R(u)}{\partial u_3}$ is a smooth matrix-valued function depending only on $u_3$ along the intersection.

Equating the two sides of (29) then yields the scalar constraint

$$\frac{\partial d_3^*(u_2, u_3)}{\partial u_2} = \xi(u_3) \, d_2^*(u_2), \tag{33}$$

where $\xi(u_3)$ is a scalar function determined by the infinitesimal rotation of $R(u)$ in the $(e_2, e_3)$-plane. Eq.33 expresses that $d_3^*$ must vary with $u_2$ according to a first-order linear relation whose coefficient depends only on $u_3$.

Finally, recall that the intersection $F_1 \cap F_3$ contains the corner point where $\psi$ reduces to a unitwise reparametrization. At this corner, the function $d_3^*$ must coincide with the corresponding unitwise scaling, which enforces $\frac{\partial d_3^*(u_2, u_3)}{\partial u_2} = 0$. However,

$d_2^*(u_2) \neq 0$ as the function is smooth and invertible everywhere. Hence, $\xi(u_3) = 0$ at the corner and thus also on the whole intersection $F_1 \cap F_3$ parametrized by $u_2$. As the same proof can be applied to $d_1^*$, overall we obtain that

$$J_{\psi|F_1 \cap F_3}(u) = I_3 \begin{pmatrix} d_1^*(u_1) & 0 & 0 \\ 0 & d_2^*(u_2) & 0 \\ 0 & 0 & d_3^*(u_3) \end{pmatrix}. \tag{34}$$

This shows that along $F_1 \cap F_3$, the mapping remains a unitwise reparametrization: each coordinate evolves independently. This conclusion is consistent with the earlier analysis at the corner and extends the unitwise reparametrization property to all intersection curves between adjacent faces $F_1, F_2, F_3$.

Eq.28 established that on the intersections of adjacent faces, the Jacobian reduces to a diagonal form, corresponding to a coordinate-wise (unitwise) reparametrization, possibly composed with a fixed signed permutation. However, this constraint initially applies only along the intersection curves themselves; a priori, when moving in a direction perpendicular to such a curve—i.e., transversally to the intersection of two faces—the mapping need not remain a pure reparametrization. The refined condition (34) strengthens this conclusion. Indeed, along the intersection $F_1 \cap F_3$, the integrability condition has shown that not only along the curve (parameterized by $u_2$) but also infinitesimally in directions orthogonal to it, the orthogonal component $R(u)$ remains fixed and equal to $I_3$. This means that the differential of $\psi$ is purely diagonal in a full neighborhood of each intersection curve:

$$J_\psi(u) = I_3 \begin{pmatrix} d_1(u_1) & 0 & 0 \\ 0 & d_2(u_2) & 0 \\ 0 & 0 & d_3(u_3) \end{pmatrix} \qquad \text{for } u \text{ in a neighborhood of } F_i \cap F_j. \tag{35}$$

In such a neighborhood, infinitesimal displacements along coordinate directions remain aligned with the coordinate axes. In particular, straight coordinate lines on the boundary, defined by fixing all but one coordinate, are mapped by $\psi$ to straight coordinate lines in the image, with no curvature introduced.

Now, consider moving an infinitesimal step $du$ away from the intersection curve into the interior of the cube. Because $R(u)$ is continuous, $R(u) = I_3$ along the intersection, and the set of orthogonal matrices is discrete in a neighborhood of the identity under the signed permutation constraint, the orthogonal component must remain constant in a connected neighborhood:

$$R(u) = I_3 \qquad \text{for all } u \text{ in a connected neighborhood of } F_i \cap F_j. \tag{36}$$

Hence, within this neighborhood, $\psi$ continues to act as a unitwise reparametrization:

$$J_\psi(u) = \text{diag}\big(d_1(u_1),\, d_2(u_2),\, d_3(u_3)\big). \tag{37}$$

Starting from one intersection curve (say $F_1 \cap F_3$), we can apply the same argument to adjacent intersection curves $F_1 \cap F_2$ and $F_2 \cap F_3$. Each of these intersections is connected and shares a corner point where $R(u) = I_3$, ensuring compatibility across overlaps. By continuity and connectedness of the cube $[0,1]^3$, these local neighborhoods form a connected cover of the entire domain. Consequently, the property (37) extends globally throughout $[0,1]^3$.

Therefore, on the whole cube, the Jacobian of $\psi$ has a globally diagonal structure with each diagonal element depending only on its corresponding coordinate:

$$J_\psi(u) = \begin{pmatrix} d_1(u_1) & 0 & 0 \\ 0 & d_2(u_2) & 0 \\ 0 & 0 & d_3(u_3) \end{pmatrix}. \tag{38}$$

Integrating this expression along each coordinate direction yields

$$\psi(u_1, u_2, u_3) = \big(\varphi_1(u_1),\, \varphi_2(u_2),\, \varphi_3(u_3)\big), \tag{39}$$

where each $\varphi_i : [0,1] \to [0,1]$ is a smooth, strictly monotone reparametrization function.

In summary, the integrability and continuity conditions jointly ensure that the local unitwise reparametrization property propagates from intersections of faces to their neighborhoods and ultimately to the entire cube. Thus, the diffeomorphism $\psi$ on $[0,1]^3$ must be a global coordinate-wise reparametrization (up to a fixed signed permutation, if present), concluding the proof. $\qquad\square$

## A.2. Proof of Proposition 1

**Proposition 1.** Let the observed data $\mathbf{x} \in \mathbb{R}^n$ lie on a $d$-dimensional manifold generated by latent variables $\mathbf{z} \in \mathbb{R}^d$, with $d \ll n$, through a smooth and invertible mapping $\mathbf{f} : \mathbb{R}^d \to \mathbb{R}^n$ satisfying Asm.1. Each latent coordinate $z_i$ corresponds to a distinct underlying concept. Suppose further that the latent variables are statistically independent, $p(\mathbf{z}) = \prod_i p(\mathbf{z}_i)$, and that at most one component $z_i$ follows a Gaussian distribution. Then, the generative process $(\mathbf{f}, p_{\mathbf{z}})$ is identifiable in the sense of Def.1.

**Proof.** Assume $\mathbf{g} : \mathbb{R}^d \to \mathbb{R}^n$ is a smooth diffeomorphism satisfying Asm.1 (the columns of its Jacobian are orthogonal) and that

$$\mathbf{f}_* p_{\mathbf{z}} = \mathbf{g}_* p_{\hat{\mathbf{z}}} \tag{40}$$

where $p_{\hat{\mathbf{z}}} = \prod_{i=1}^{d} p_{\hat{z}_i}$ i.e., the latent coordinates $\hat{\mathbf{z}} \in \mathbb{R}^d$ are independent. We must show that $\mathbf{g}$ recovers $\mathbf{f}$ up to the ambiguities allowed in Def.1 (a permutation and element-wise reparameterizations).

Because $\mathbf{f}$ and $\mathbf{g}$ are diffeomorphisms onto the same $d$-dimensional manifold, define

$$\mathbf{h} := \mathbf{f}^{-1} \circ \mathbf{g} : \mathbb{R}^d \to \mathbb{R}^d. \tag{41}$$

By the pushforward identity we have

$$\mathbf{h}_* p_{\hat{\mathbf{z}}} = p_{\mathbf{z}}. \tag{42}$$

Thus $\mathbf{h}$ is a smooth bijection that transforms the product density $p_{\hat{\mathbf{z}}}$ into the product density $p_{\mathbf{z}}$.

Fix $u \in \mathbb{R}^d$ and let $v = \mathbf{h}(u)$. By Asm.1, at the corresponding points the Jacobians of $\mathbf{f}$ and $\mathbf{g}$ admit the column-orthogonal factorization

$$J_{\mathbf{f}}(v) = Q_{\mathbf{f}}(v) D_{\mathbf{f}}(v), \qquad J_{\mathbf{g}}(u) = Q_{\mathbf{g}}(u) D_{\mathbf{g}}(u), \tag{43}$$

where for each argument $Q_{\mathbf{f}}(v), Q_{\mathbf{g}}(u) \in \mathbb{R}^{n \times d}$ have orthonormal columns ($Q^T Q = I_d$), and $D_{\mathbf{f}}(v), D_{\mathbf{g}}(u) \in \mathbb{R}^{d \times d}$ are diagonal matrices. Because $\mathbf{f}$ is a local diffeomorphism, the derivative of $\mathbf{f}^{-1}$ at $x = \mathbf{f}(v)$ equals the left inverse of $J_{\mathbf{f}}(v)$:

$$D(\mathbf{f}^{-1})(x) = (J_{\mathbf{f}}(v))^+ = (J_{\mathbf{f}}(v)^T J_{\mathbf{f}}(v))^{-1} J_{\mathbf{f}}(v)^T = D_{\mathbf{f}}(v)^{-1} Q_{\mathbf{f}}(v)^T \tag{44}$$

where we used $J_{\mathbf{f}} = Q_{\mathbf{f}} D_{\mathbf{f}}$ and $Q_{\mathbf{f}}^T Q_{\mathbf{f}} = I_d$. Consequently, the Jacobian of $\mathbf{h}$ at $u$ is

$$J_{\mathbf{h}}(u) = D(\mathbf{f}^{-1})(\mathbf{g}(u)) J_{\mathbf{g}}(u) = D_{\mathbf{f}}(v)^{-1} Q_{\mathbf{f}}(v)^T Q_{\mathbf{g}}(u), D_{\mathbf{g}}(u). \tag{45}$$

Since $\mathbf{f}(v) = \mathbf{g}(u)$ is the same point $x$ on the embedded manifold, the column spaces of $Q_{\mathbf{f}}(v)$ and $Q_{\mathbf{g}}(u)$ form both an orthogonal basis for the tangent space $T_x$ of the embedded manifold at $x$. In this tangent space they have orthonormal columns (rows) and thus also orthonormal rows (columns). Hence there exists a $d \times d$ orthogonal matrix

$$R(u) := Q_{\mathbf{f}}(v)^T Q_{\mathbf{g}}(u) \tag{46}$$

with $R(u)^T R(u) = R(u) R(u)^T = I_d$. Therefore the Jacobian of $\mathbf{h}$ admits the pointwise decomposition

$$J_{\mathbf{h}}(u) = D_{\mathbf{f}}(v)^{-1} R(u) D_{\mathbf{g}}(u), \tag{47}$$

i.e. a diagonal–orthogonal–diagonal factorization valid for every $u$.

Now, because both $p_{\mathbf{z}}$ and $p_{\hat{\mathbf{z}}}$ factorize into independent distributions, introduce the coordinatewise CDF maps (which are smooth, strictly monotone)

$$\tau := (F_{z_1}, ..., F_{z_d}) : \mathbb{R}^d \to (0,1)^d, \qquad \hat{\tau} := (F_{\hat{z}_1}, ..., F_{\hat{z}_d}) : \mathbb{R}^d \to (0,1)^d, \tag{48}$$

which push the marginals to independent $\mathrm{Unif}(0,1)$ coordinates. Define

$$\psi := \tau \circ \mathbf{h} \circ \hat{\tau}^{-1} : (0,1)^d \to (0,1)^d. \tag{49}$$

The maps $\tau$ and $\hat{\tau}$ are coordinatewise (their Jacobians are diagonal), hence the pointwise structure (47) is preserved up to diagonal pre- and post-multiplication: the Jacobian $J_\psi$ has the same diagonal–orthogonal–diagonal form as $J_{\mathbf{h}}$ (with possibly different diagonal factors).

Next, we will show that such a mapping $\psi$ will need to be a signed permutation. Recall that $\psi : (0,1)^d \to (0,1)^d$ is a smooth diffeomorphism whose Jacobian admits, at every point, a diagonal–orthogonal–diagonal factorization. This transformation extends continuously and smoothly to a diffeomorphism of the closed hypercube $[0,1]^d$ onto itself. In particular, $\psi$ is a continuous bijection with a continuous inverse on $[0,1]^d$, and thus a homeomorphism. By classical topological results (Brouwer (1911) Domain Invariance Theorem), any homeomorphism between subsets of $\mathbb{R}^d$ maps interiors onto interiors and boundaries onto boundaries. Consequently, $\psi$ maps the hypercube $[0,1]^d$ onto itself and sends boundary points of the hypercube to boundary points. Moreover, since $\mathbf{h}$ is globally defined and smooth, the Jacobian field $J_\psi$ satisfies the usual integrability conditions corresponding to a globally defined diffeomorphism. We are therefore exactly in the setting of the Preliminary result in Sec.A.1. Applying it to $\psi$, we conclude that there exists a fixed signed permutation matrix $P$ and smooth, strictly monotone univariate functions $\varphi_i : [0,1] \to [0,1]$ such that

$$\psi(u_1, \ldots, u_d) = P\big(\varphi_1(u_1), \ldots, \varphi_d(u_d)\big), \tag{50}$$

for all $(u_1, \ldots, u_d) \in (0,1)^d$. In other words, up to a fixed signed permutation of coordinates, $\psi$ is a global coordinate-wise reparameterization of the unit hypercube.

We now invoke the statistical definition of $\psi$: by construction, $\psi$ pushes each marginal of the original distribution to the uniform distribution on $(0,1)$. Formally, if $U = \psi(X)$ and $X = (X_1, X_2, X_3)$ denotes a random vector with continuous marginals, then for each $i \in \{1, 2, 3\}$,

$$U_i = \psi_i(X) \sim \mathrm{Unif}(0,1), \tag{51}$$

and the mapping $\psi$ is monotone and invertible along each coordinate.

From the geometric analysis above, we have established that $\psi$ must take the globally diagonal form

$$\psi(u_1, u_2, u_3) = \big(\varphi_1(u_{\sigma(1)}), \varphi_2(u_{\sigma(2)}), \varphi_3(u_{\sigma(3)})\big), \tag{52}$$

for some smooth, strictly increasing univariate functions $\varphi_i : [0,1] \to [0,1]$ and some permutation $\sigma$ of $\{1,2,3\}$ (encoding the fixed signed permutation of coordinate axes identified earlier).

Because $\psi$ maps each marginal distribution to the uniform law on $(0,1)$, each component $\varphi_i$ must itself transform a $\mathrm{Unif}(0,1)$ random variable into another $\mathrm{Unif}(0,1)$ random variable. The only smooth, strictly monotone maps on $(0,1)$ satisfying this invariance are the identity and its reflections $t \mapsto 1 - t$, corresponding to the two possible signed orientations. Hence,

$$\varphi_i(t) \in \{\, t,\, 1 - t\, \}, \qquad \forall i \in \{1, 2, 3\}. \tag{53}$$

Therefore, under the statistical and geometric constraints, the diffeomorphism $\psi$ reduces to a *signed permutation of the coordinates*.

Now, undoing the marginal transforms (i.e. returning from uniform coordinates to the original coordinates) gives

$$\mathbf{h} = \tau^{-1} \circ \psi \circ \hat{\tau} = \tau^{-1} \circ P \circ \hat{\tau} \tag{54}$$

Hence, $\mathbf{h}$ acts as a permutation of coordinates composed with elementwise reparameterizations. In other words, $\mathbf{h}$ operates independently on each coordinate of $\hat{\mathbf{z}}$ through a unitwise reparameterization $\mathbf{t}$ (given by the composition of $\tau$ and $\hat{\tau}$, up to permutation) s.t.

$$p_{\mathbf{z}} = \mathbf{h}_* p_{\hat{\mathbf{z}}} = \mathbf{t}_* p_{\hat{\mathbf{z}}}. \tag{55}$$

It is important to note that this is not equivalent to $\mathbf{z} = \mathbf{t}(\hat{\mathbf{z}})$. Indeed, the equality is between distributions. $\mathbf{h}$ acts as a unitwise reparameterization on $\hat{\mathbf{z}}$ such that the distributions match and thus the axis could still be unaligned. However, because both $p_{\mathbf{z}}$ and $p_{\hat{\mathbf{z}}}$ are factorized into independent components and because a unitwise reparameterization $\mathbf{t}$ preserves independence, the transformed variables $\mathbf{t}(\hat{\mathbf{z}})$ remain mutually independent. And thus, there is at most an additional global orthogonal transformation $Q$ acting on the coordinates,

$$\mathbf{z} = Q \circ \mathbf{t}(\hat{\mathbf{z}}). \tag{56}$$

This ambiguity corresponds to the well-known indeterminacy in Independent Component Analysis (ICA), where the independent components are recoverable only up to scaling, permutation, and orthogonal rotation when multiple Gaussian sources are present.

However, under the additional assumption that at most one latent variable follows a Gaussian distribution, the ICA indeterminacy reduces to permutations and elementwise reparameterizations only. In this case, the orthogonal rotation $Q$ must necessarily be a permutation matrix $\mathbf{P}$, since any non-trivial rotation would induce statistical dependence among the coordinates, contradicting the independence of $p_{\mathbf{z}}$. Consequently, we obtain

$$\mathbf{z} = \mathbf{P} \circ \mathbf{t}(\hat{\mathbf{z}}), \tag{57}$$

which is precisely the form of identifiability stated in Def.1.

Therefore, the generative process $(\mathbf{f}, p_{\mathbf{z}})$ is identifiable up to permutation and coordinate-wise invertible reparameterizations of the latent variables, completing the proof. □

**Discussion.**    This proof makes explicit that Prop.1 can be viewed as a nonlinear generalization of classical *Independent Component Analysis* (ICA), where identifiability emerges from the interplay between a geometric constraint on the mixing function and a statistical constraint on the latent distribution. In linear ICA, identifiability follows from the fact that a linear mixing matrix with orthogonal columns preserves angles, while statistical independence restricts the admissible transformations to permutations and scalings, except in the presence of multiple Gaussian sources. Our analysis shows that the same logic extends to the nonlinear setting when the Jacobian of the generative mapping satisfies a pointwise orthogonality constraint.

From a geometric perspective, Asm.1 enforces a strong form of functional independence: at every point in latent space, each latent coordinate influences the observations along an orthogonal direction in the tangent space of the data manifold. This local orthogonality plays the role of a nonlinear analogue of linear orthogonal mixing. The key technical step of the proof shows that any alternative generative explanation inducing the same observed distribution must differ from the true one by a diffeomorphism whose Jacobian admits a diagonal–orthogonal–diagonal factorization everywhere. Such mappings form a highly restricted class, and when combined with global statistical independence, they collapse to coordinate-wise transformations up to permutation.

The use of marginal CDF transforms highlights this collapse particularly clearly. By reducing the problem to diffeomorphisms of the unit hypercube that preserve independence and admit the same Jacobian structure, the identifiability question becomes purely geometric and topological. The result that such transformations must be signed permutations with coordinate-wise reparameterizations mirrors the classical ICA argument that only trivial symmetries preserve independence. The additional assumption that at most one latent variable is Gaussian then removes the residual orthogonal ambiguity, exactly as in linear ICA.

Importantly, this argument clarifies why nonlinear ICA is generally non-identifiable in the absence of further constraints, as shown in prior impossibility results, and why Asm.1 fundamentally changes the situation. Rather than attempting to recover independent components from arbitrary nonlinear mixtures, the orthogonal Jacobian assumption restricts the functional class of admissible generative models to those that behave locally like orthogonal linear maps. In this sense, Asm.1 can be interpreted as a structural prior that collapses the nonlinear ICA problem back into an identifiable regime.

Conceptually, Prop.1 demonstrates that identifiability does not rely on linearity per se, but on the preservation of orthogonal functional influence combined with a global statistical constraint. This perspective unifies linear ICA and the more general orthogonal-Jacobian setting studied here under a single principle: disentanglement becomes identifiable whenever local geometric independence and global statistical independence act in concert. This insight also explains why empirical methods that enforce orthogonality or decorrelation in the Jacobian often succeed in practice, even in nonlinear models, when the underlying generative process approximately satisfies these assumptions.

### A.3. Proof of Proposition 2

**Proposition 2.**    Let the observed data $\mathbf{x} \in \mathbb{R}^n$ lie on a $d$-dimensional manifold generated by latent variables $\mathbf{z} \in \mathbb{R}^d$, with $d \ll n$, through a smooth and invertible mapping $\mathbf{f} : \mathbb{R}^d \to \mathbb{R}^n$ satisfying Asm.1. Suppose further that the latent distribution $p_{\mathbf{z}}$ satisfies Asm.2. Then, the generative process $(\mathbf{f}, p_{\mathbf{z}})$ is identifiable in the sense of Def.1.

**Proof.**    Assume $\mathbf{g} : \mathbb{R}^d \to \mathbb{R}^n$ is a smooth diffeomorphism satisfying Asm.1 (the columns of its Jacobian are orthogonal) and that

$$\mathbf{f}_* p_{\mathbf{z}} = \mathbf{g}_* p_{\hat{\mathbf{z}}} \tag{58}$$

where $p_{\hat{\mathbf{z}}}$ satisfies Asm.2. We must show that $\mathbf{g}$ recovers $\mathbf{f}$ up to the ambiguities allowed in Def.1 (a permutation and element-wise reparameterizations).

Because $\mathbf{f}$ and $\mathbf{g}$ are diffeomorphisms onto the same $d$-dimensional manifold, define

$$\mathbf{h} := \mathbf{f}^{-1} \circ \mathbf{g} : \mathbb{R}^d \to \mathbb{R}^d. \tag{59}$$

By the pushforward identity we have

$$\mathbf{h}_* p_{\hat{\mathbf{z}}} = p_{\mathbf{z}}. \tag{60}$$

Thus $\mathbf{h}$ is a smooth bijection that transforms the density $p_{\hat{\mathbf{z}}}$ into the density $p_{\mathbf{z}}$.

Fix $u \in \mathbb{R}^d$ and let $v = \mathbf{h}(u)$. By Asm.1, at the corresponding points the Jacobians of $\mathbf{f}$ and $\mathbf{g}$ admit the column-orthogonal factorization

$$J_{\mathbf{f}}(v) = Q_{\mathbf{f}}(v) D_{\mathbf{f}}(v), \qquad J_{\mathbf{g}}(u) = Q_{\mathbf{g}}(u) D_{\mathbf{g}}(u), \tag{61}$$

where for each argument $Q_{\mathbf{f}}(v), Q_{\mathbf{g}}(u) \in \mathbb{R}^{n \times d}$ have orthonormal columns ($Q^T Q = I_d$), and $D_{\mathbf{f}}(v), D_{\mathbf{g}}(u) \in \mathbb{R}^{d \times d}$ are diagonal matrices. Because $\mathbf{f}$ is a local diffeomorphism, the derivative of $\mathbf{f}^{-1}$ at $x = \mathbf{f}(v)$ equals the left inverse of $J_{\mathbf{f}}(v)$:

$$D(\mathbf{f}^{-1})(x) = (J_{\mathbf{f}}(v))^+ = (J_{\mathbf{f}}(v)^T J_{\mathbf{f}}(v))^{-1} J_{\mathbf{f}}(v)^T = D_{\mathbf{f}}(v)^{-1} Q_{\mathbf{f}}(v)^T \tag{62}$$

where we used $J_{\mathbf{f}} = Q_{\mathbf{f}} D_{\mathbf{f}}$ and $Q_{\mathbf{f}}^T Q_{\mathbf{f}} = I_d$. Consequently, the Jacobian of $\mathbf{h}$ at $u$ is

$$J_{\mathbf{h}}(u) = D(\mathbf{f}^{-1})(\mathbf{g}(u)) J_{\mathbf{g}}(u) = D_{\mathbf{f}}(v)^{-1} Q_{\mathbf{f}}(v)^T Q_{\mathbf{g}}(u), D_{\mathbf{g}}(u). \tag{63}$$

Since $\mathbf{f}(v) = \mathbf{g}(u)$ is the same point $x$ on the embedded manifold, the column spaces of $Q_{\mathbf{f}}(v)$ and $Q_{\mathbf{g}}(u)$ form both an orthogonal basis for the tangent space $T_x$ of the embedded manifold at $x$. In this tangent space they have orthonormal columns (rows) and thus also orthonormal rows (columns). Hence there exists a $d \times d$ orthogonal matrix

$$R(u) := Q_{\mathbf{f}}(v)^T Q_{\mathbf{g}}(u) \tag{64}$$

with $R(u)^T R(u) = R(u) R(u)^T = I_d$. Therefore the Jacobian of $\mathbf{h}$ admits the pointwise decomposition

$$J_{\mathbf{h}}(u) = D_{\mathbf{f}}(v)^{-1} R(u) D_{\mathbf{g}}(u), \tag{65}$$

i.e. a diagonal–orthogonal–diagonal factorization valid for every $u$.

Now, because both $p_{\mathbf{z}}$ and $p_{\hat{\mathbf{z}}}$ satisfy Asm.2, there exist smooth, strictly monotone coordinate-wise reparameterizations mapping their supports to the unit interval. More precisely, Asm.2 ensures the existence of smooth diffeomorphisms

$$\tau := (\tau_1, \ldots, \tau_d) : \mathbb{R}^d \to (0,1)^d, \qquad \hat{\tau} := (\hat{\tau}_1, \ldots, \hat{\tau}_d) : \mathbb{R}^d \to (0,1)^d, \tag{66}$$

where each $\tau_i, \hat{\tau}_i : \mathbb{R} \to (0,1)$ is strictly monotone. These maps are not assumed to be unique, nor to push $p_{\mathbf{z}}$ and $p_{\hat{\mathbf{z}}}$ to the same distribution on $(0,1)^d$; they merely provide a common normalized range.

Define the mapping

$$\psi := \tau \circ \mathbf{h} \circ \hat{\tau}^{-1} : (0,1)^d \to (0,1)^d. \tag{67}$$

By construction, $\psi$ is a smooth diffeomorphism of the open unit hypercube. Since both $\tau$ and $\hat{\tau}$ act independently on each coordinate, their Jacobians are diagonal everywhere. Consequently, conjugation by $\tau$ and $\hat{\tau}$ preserves the diagonal–orthogonal–diagonal structure of the Jacobian (with possibly different diagonal factors).

Next, we will show that such a mapping $\psi$ will need to be a signed permutation and coordinate-wise reparametrization. Recall that $\psi : (0,1)^d \to (0,1)^d$ is a smooth diffeomorphism whose Jacobian admits, at every point, a diagonal–orthogonal–diagonal factorization. This transformation extends continuously and smoothly to a diffeomorphism of the closed hypercube $[0,1]^d$ onto itself. In particular, $\psi$ is a continuous bijection with a continuous inverse on $[0,1]^d$, and thus a homeomorphism. By classical topological results (Brouwer (1911) Domain Invariance Theorem), any homeomorphism between subsets of $\mathbb{R}^d$ maps interiors onto interiors and boundaries onto boundaries. Consequently, $\psi$ maps the hypercube $[0,1]^d$ onto itself and sends boundary points of the hypercube to boundary points. Moreover, since $\mathbf{h}$ is globally defined and smooth, the Jacobian field $J_\psi$ satisfies the usual integrability conditions corresponding to a globally defined diffeomorphism. We are therefore

exactly in the setting of the Preliminary result in Sec.A.1. Applying it to $\psi$, we conclude that there exists a fixed signed permutation matrix $P$ and smooth, strictly monotone univariate functions $\varphi_i : [0, 1] \to [0, 1]$ such that

$$\psi(u_1, \ldots, u_d) = P\big(\varphi_1(u_1), \ldots, \varphi_d(u_d)\big), \tag{68}$$

for all $(u_1, \ldots, u_d) \in (0, 1)^d$. In other words, *up to a fixed signed permutation of coordinates, $\psi$ is a global coordinate-wise reparameterization of the unit hypercube.*

Now, undoing the marginal normalizations (i.e. returning to the original coordinates) yields

$$\mathbf{h} = \tau^{-1} \circ \psi \circ \hat{\tau} = \tau^{-1} \circ P \circ \phi \circ \hat{\tau}, \tag{69}$$

where $P$ is a fixed permutation matrix and $\phi(u_1, \ldots, u_d) = (\phi_1(u_1), \ldots, \phi_d(u_d))$ is a coordinate-wise smooth, strictly monotone reparameterization of $(0, 1)^d$.

Consequently, $\mathbf{h}$ acts as a permutation of coordinates composed with element-wise reparameterizations. Equivalently, there exists a coordinate-wise invertible map $\mathbf{t} : \mathbb{R}^d \to \mathbb{R}^d$ such that

$$p_{\mathbf{z}} = \mathbf{h}_* p_{\hat{\mathbf{z}}} = \mathbf{t}_* p_{\hat{\mathbf{z}}}. \tag{70}$$

It is important to emphasize that this equality holds at the level of distributions: it does not imply the pointwise identity $\mathbf{z} = \mathbf{t}(\hat{\mathbf{z}})$. Indeed, a unit-wise reparameterization may map distributions correctly even if the coordinate axes of $\mathbf{z}$ and $\hat{\mathbf{z}}$ are not aligned.

However, both $p_{\mathbf{z}}$ and $p_{\hat{\mathbf{z}}}$ satisfy Asm.2. In particular, since they have finite support that are hypercubes up to a reparametrization, their supports must coincide with a compact hyperrectangle, and their copulas have full support on the interior of this domain. As a consequence, the diffeomorphism $\mathbf{h}$ extends continuously to the boundary of the support and maps boundary points to boundary points and interior points to interior points. Under these conditions, any smooth bijection mapping $p_{\hat{\mathbf{z}}}$ to $p_{\mathbf{z}}$ through a coordinate-wise reparameterization must preserve the axis-aligned structure of the support. Indeed, any nontrivial mixing of coordinates would necessarily map a boundary face of the hyperrectangle to a set intersecting the interior, or would destroy the full-support property of the copula, contradicting Asm.2. Therefore, the only admissible transformations are coordinate-wise reparameterizations composed with permutations of the coordinate axes.

We can thus conclude that $\mathbf{h}$ is identifiable up to a permutation and coordinate-wise invertible reparameterizations of the latent variables. Consequently, the generative process $(\mathbf{f}, p_{\mathbf{z}})$ is identifiable in the sense of Def.1, completing the proof. $\square$

**Discussion.** The proof of Prop.2 makes explicit the mechanisms underlying identifiability in the dependent case, and in particular clarifies the respective roles played by support geometry and distributional assumptions. Building on these insights, the full-support copula assumption can be viewed as a natural generalization of the independence setting: requiring that all combinations of latent coordinates occur ensures that no factor is functionally constrained by the others, which is precisely what enables disentanglement. However, in contrast to the independent case, this assumption alone is not sufficient unless the latent variables are supported on a *finite* domain.

This additional requirement is essential for identifiability. When the latent space is $\mathbb{R}^d$, there exist infinitely many smooth, monotone transformations that map $\mathbb{R}^d$ onto a bounded domain such as the hypercube $(0, 1)^d$. For instance, applying sigmoid-type transformations along arbitrary directions yields different embeddings of an unbounded domain into the same bounded support, all of which are compatible with the same observed distribution. As a result, the geometry of the latent space cannot be uniquely recovered from data in this setting.

By contrast, when the latent variables are supported on a bounded domain with full copula support, the boundary structure of the support becomes identifiable and constrains the class of admissible transformations, leading to disentanglement up to coordinate-wise reparametrization and permutation. The independent case does not require such a boundedness assumption because independence provides strictly stronger information: it fixes not only the support geometry but also the factorization structure of the distribution, thereby ruling out these additional ambiguities even when the latent space is $\mathbb{R}^d$.

### A.4. Discussion on Assumption 1 and Assumption 1

This section elaborates on the practical scope and interpretation of Asm.1 and 2. These assumptions are not only theoretically motivated but also well-grounded in widely accepted principles and common practices in representation learning; we

therefore view them as structural abstractions that enable studying identifiability beyond the classical independence setting. In what follows, we discuss each assumption in turn, clarifying its conceptual grounding, its relationship to standard causal representation learning settings, and the conditions under which it can be expected to hold in practice.

**Assumption 1 (Orthogonal Jacobian).** Asm.1 should be understood not as an empirical property to be verified in a given dataset, but as a *definitional criterion* for what constitutes a meaningful latent concept: each factor contributes a functionally independent mode of variation, decoupled from all others in its local effect on the observations. This view is strongly aligned with the Independent Causal Mechanisms (ICM) principle from causal inference (Schölkopf et al., 2021), a widely accepted foundation which posits that the mechanisms governing the components of a system operate autonomously and do not inform one another. Applied outside a strictly causal context, Asm.1 can be understood as a functional, non-parametric instantiation of ICM: instead of requiring independence in a causal graph, it requires decoupling in the generative Jacobian. This connection is well-supported by the theoretical analysis in Gresele et al. (2021), where orthogonal Jacobian structure was introduced precisely to capture this notion.

Beyond its theoretical motivation, Asm.1 finds direct empirical support in practical settings. Wei et al. (2021) successfully leveraged Jacobian orthogonality constraints for unsupervised disentanglement in image generation, demonstrating that such structural priors are not only theoretically meaningful but also empirically effective on realistic data. Moreover, as discussed in Sec.6, VAEs implicitly encourage a related structure through their factorized approximate posterior (Reizinger et al., 2022; Allen, 2024), which helps explain their empirical disentanglement ability on structured datasets.

In summary, Asm.1 is best understood not as an empirical property to be verified in any given dataset, but as a *structural design criterion* for what kind of representation one seeks to learn: one in which each factor's functional influence on the observations is orthogonal to that of all others.

**Assumption 2 (Full-Support Copula).** Asm.2 formalizes the combinatorial richness of the latent factors, capturing the idea that each factor should, in principle, be independently manipulable. Many latent variables (e.g., position, orientation, lighting) naturally vary across bounded ranges, and datasets are often explicitly constructed to explore these variations, making this assumption well-aligned with common practices in representation learning.

A fundamental distinction, however, is that Asm.2 is a *dataset-level property* of the support of the latent distribution, rather than a property of any underlying causal or generative model. The existence of a causal model does not, by itself, guarantee that the resulting dataset will be rich enough to satisfy it. To illustrate this, consider two latent factors $(Z_1, Z_2)$ with a causal dependency $Z_1 \rightarrow Z_2$:

- *Deterministic mechanism*: if $Z_1 = 0$ implies $Z_2 = 0$ and $Z_1 = 1$ implies $Z_2 = 1$, then combinations such as $(Z_1 = 0, Z_2 = 1)$ never occur. The dataset lacks full combinatorial coverage, violating Asm.2, even though a valid causal model is present.
- *Probabilistic mechanism*: if $Z_1 = 0$ yields $P(Z_2 = 0) = 0.9$ and $P(Z_2 = 1) = 0.1$, then every combination of $(Z_1, Z_2)$ appears with non-zero probability. Causal dependencies exist and the variables are statistically dependent, yet Asm.2 is perfectly satisfied because the joint latent space has full support.

This shows that Asm.2 can be satisfied in typical causal representation learning settings, provided the dataset is sufficiently diverse. Moreover, even when natural observational data lacks certain combinations, interventions — either explicit or through deliberate dataset design — can restore the required combinatorial richness.

Crucially, this highlights that causal structure and Asm.2 are *orthogonal*: our work does not rely on any causal assumptions, but rather focuses on the combinatorial richness of the dataset, regardless of how the data was generated (causally or not). This formalizes the idea that each latent factor should correspond to a distinct, manipulable degree of freedom ("concept slider"). If two variables can never be varied separately in the observed data, they cannot be identified as separate entities without further supervision. Thus, while Asm.2 defines the conditions under which disentanglement is theoretically possible, it may not be achieved in every real-world scenario, particularly if the dataset is not sufficiently rich or lacks the necessary interventional samples to make these separate degrees of freedom visible to the disentanglement model. For instance, variables such as temperature and altitude may not span all possible combinations in observed data, even though such combinations are physically plausible. So the assumption should be interpreted as a property of the underlying generative domain, not a guarantee about any finite dataset. Even so, the assumption provides a critical theoretical foundation: it delineates when disentanglement is possible, by ensuring that distinct latent factors are identifiable in principle. In other words, while actual datasets may be incomplete, Asm.2 highlights the necessary richness required to make independent degrees of freedom visible to the model, grounding identifiability results in realistic, physically plausible factor spaces.

# B. Disentanglement Under General Latent Domains

In the main text, we argued that identifiability does not necessarily require strong distributional assumptions such as statistical independence. Instead, *limited but appropriate global information about the latent domain* may already suffice to fully determine the generative mapping. In particular, when the latent variables are supported on a known, finite, simply connected domain $\Omega \subset \mathbb{R}^d$, the generative process can become identifiable up to the natural rigid symmetries of that support. In the main body of the paper, this idea was instantiated through Asm.2, which restricts the latent domain (up to coordinate-wise reparameterization) to a hypercube. However, the above observation suggests that identifiability can be achieved under substantially more general geometric conditions. The purpose of this appendix is to formalize and extend this claim by studying disentanglement when the latent domain is known but not necessarily a hypercube.

For clarity, we restrict our analysis in this appendix to the class of *conformal* generative mappings. This setting allows us to leverage classical results from geometric function theory and to make precise statements about how domain geometry constrains admissible transformations. In particular, for conformal mappings in dimensions $d \geq 3$, the knowledge of a small amount of global geometric information—such as the support of the latent variables or the correspondence of a few salient points—is already sufficient to uniquely determine the mapping up to rigid symmetries.

This intuition is formalized in the following proposition, which establishes identifiability under mild geometric assumptions on the latent support.

**Proposition 3.** *Let the observed data $\mathbf{x} \in \mathbb{R}^n$ lie on a $d$-dimensional manifold generated by latent variables $\mathbf{z} \in \mathbb{R}^d$, with $3 \leq d \ll n$, through a smooth and invertible mapping $\mathbf{f} : \mathbb{R}^d \to \mathbb{R}^n$ that is conformal everywhere on the domain in an open subset. Suppose further that the latent variables follow a distribution $p_{\mathbf{z}}$ supported on a known, finite, simply connected region $\Omega \subset \mathbb{R}^d$. For any $c \in \mathbb{R}^d$, there exist $u, v \in \mathbb{S}^{d-1}$ such that, denoting by $(r_1(u), r_2(u))$ and $(r_1(v), r_2(v))$ the intersections of $\Omega$ with the rays emanating from $c$, one has $r_1(u)r_2(u) \neq r_1(v)r_2(v)$. Then, the generative process $(\mathbf{f}, p_{\mathbf{z}})$ is identifiable in the sense of Def.1, up to a rigid transformation that is a symmetry of $\Omega$.*

The key theoretical ingredient underlying Prop.3 is Liouville's theorem for conformal mappings in dimensions $d > 2$ (Blair, 2000). This result implies that any conformal transformation of $\mathbb{R}^d$ must be a Möbius transformation, i.e., a composition of translations, rotations, dilations, and inversions. Among these components, inversion is the only source of nonlinearity. When the latent domain $\Omega$ is finite, simply connected, and not invariant under inversion (a property satisfied by a broad class of supports including all polytopes and most bounded smooth domains), non-rigid Möbius transformations cannot preserve $\Omega$. The geometric condition imposed in Prop.3 precisely excludes this possibility by ensuring that no inversion can map $\Omega$ onto itself. As a result, the only admissible transformations are rigid motions (rotations and translations) that are symmetries of the domain.

This establishes that identifiability can be achieved under substantially weaker assumptions than those imposed in the main text. In particular, it shows that disentanglement is possible for a wide class of known latent supports—not only hypercubes, provided that the domain geometry sufficiently constrains admissible conformal symmetries. Only highly symmetric sets, such as perfect spheres, violate this condition and admit non-rigid self-transformations.

We empirically illustrate this phenomenon in Fig.4 of the main paper and in Figs.15 and 16 of App.D, where we observe identifiability up to rigid rotations for several non-cubic latent domains.

In the following sections, we provide a proof of Prop.3 along with additional geometric intuition and visualizations. We then treat separately the special case $d = 2$, where conformal mappings exhibit fundamentally different behavior.

## B.1. Proof and Geometric Interpretation

We now prove Prop.3 and provide geometric intuition for the role played by the latent domain geometry.

Let $\mathbf{f}, \mathbf{g} : \mathbb{R}^d \to \mathbb{R}^n$ be smooth, invertible mappings that are conformal on an open neighborhood of a finite, simply connected domain $\Omega \subset \mathbb{R}^d$, with $d \geq 3$. Assume that the induced data distributions coincide, i.e.,

$$\mathbf{f}_* p_{\mathbf{z}} = \mathbf{g}_* p_{\hat{\mathbf{z}}}, \tag{71}$$

where both $p_{\mathbf{z}}$ and $p_{\hat{\mathbf{z}}}$ are supported on $\Omega$.

Since $\mathbf{f}$ and $\mathbf{g}$ are diffeomorphisms onto the same $d$-dimensional data manifold, we define

$$\mathbf{h} := \mathbf{f}^{-1} \circ \mathbf{g} : \mathbb{R}^d \to \mathbb{R}^d. \tag{72}$$

By construction, $\mathbf{h}$ is a smooth bijection. Using the pushforward identity, we obtain

$$\mathbf{h}_* p_{\hat{\mathbf{z}}} = p_{\mathbf{z}}, \tag{73}$$

which implies that $\mathbf{h}$ maps $\Omega$ onto itself almost everywhere. Hence, $\mathbf{h}$ is an automorphism of the latent domain $\Omega$.

Because conformality is preserved under inversion and composition, and both $\mathbf{f}$ and $\mathbf{g}$ are conformal, the map $\mathbf{h}$ is itself conformal on an open neighborhood of $\Omega$. The identifiability problem therefore reduces to characterizing all conformal automorphisms of $\Omega$.

For dimensions $d \geq 3$, Liouville's theorem (Blair, 2000) states that any conformal map between open subsets of $\mathbb{R}^d$ must be a Möbius transformation. Consequently, $\mathbf{h}$ must be of the form

$$\mathbf{h}(x) = b + \frac{\alpha A(x-a)}{\|x-a\|^\varepsilon}, \tag{74}$$

where $a, b \in \mathbb{R}^d$, $\alpha \in \mathbb{R} \setminus \{0\}$, $A \in O(d)$ is an orthogonal matrix, and $\varepsilon \in \{0, 2\}$. Here, $a$ denotes the center of inversion, $b$ a translation, $\alpha$ a uniform dilation, and $A$ a rotation or reflection. The case $\varepsilon = 0$ corresponds to affine conformal maps (no inversion), while $\varepsilon = 2$ corresponds to Möbius transformations involving inversion.

Geometrically, Möbius transformations admit a classical interpretation via stereographic projection. Specifically, they can be obtained by projecting $\mathbb{R}^d$ onto the $d$-sphere, applying a rigid motion of the sphere, and projecting back via stereographic projection. A visualization of this construction is provided in Fig.5. In this representation, inversion corresponds to a transformation that exchanges a point $a$ with infinity: when $\varepsilon = 2$, the point $x = a$ is mapped to infinity, and infinity is mapped to $b$. When $\varepsilon = 0$, infinity is preserved.

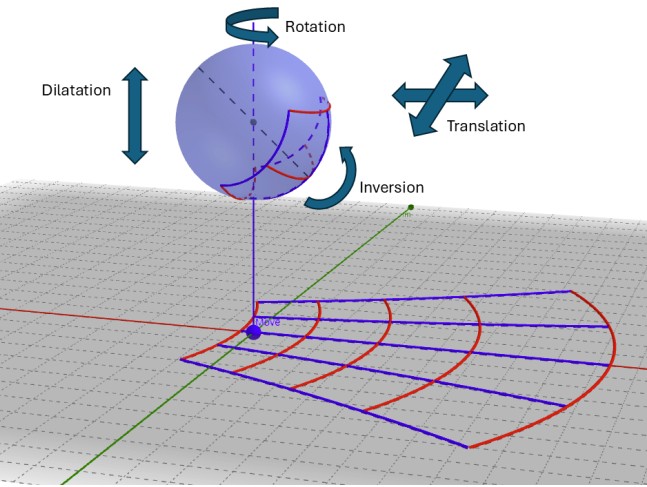

*Figure 5.* Geometric interpretation of Möbius transformations via stereographic projection. Points in $\mathbb{R}^d$ are projected onto the sphere, transformed by a rigid motion of the sphere, and mapped back to $\mathbb{R}^d$ via inverse stereographic projection. Plot generated using a visualization tool created by Juan Carlos Ponce Campuzano on Geogebra

We now analyze which Möbius transformations can be automorphisms of the finite domain $\Omega$.

First, consider the case $\varepsilon = 0$. In this case, $\mathbf{h}$ reduces to an affine conformal map,

$$\mathbf{h}(x) = b + \alpha A(x-a). \tag{75}$$

Translations, rotations, and dilations preserve the global shape of a domain. Since $\Omega$ is finite, $\mathbf{h}$ cannot involve arbitrary translations or dilations while remaining an automorphism. Consequently, $\mathbf{h}$ must reduce to a rigid transformation belonging to the symmetry group of $\Omega$. This yields identifiability up to rigid symmetries.

We now turn to the only source of nonlinearity: the case $\varepsilon = 2$, corresponding to inversion. All other components of a Möbius transformation—translations, rotations, and dilations—preserve the qualitative shape of a domain. Therefore, if a Möbius transformation is to act as a nontrivial automorphism of $\Omega$, the inversion component must itself preserve the domain.

Without loss of generality, we may reduce the analysis to inversion with respect to the unit sphere,

$$\mathcal{I}(x) = \frac{x}{\|x\|^2}, \tag{76}$$

since any other inversion can be obtained by composing $\mathcal{I}$ with translations, rotations, and dilations.

Inversion has two fundamental geometric effects. First, it exchanges the interior and exterior of the unit sphere: points inside the sphere are mapped outside, and vice versa. Second, the center of inversion is sent to infinity, and infinity is sent to the center. Since $\Omega$ is finite, the center of inversion must lie outside $\Omega$; otherwise, points in $\Omega$ would be mapped to infinity, which is incompatible with $\mathbf{h}$ being an automorphism.

For inversion to preserve $\Omega$, the domain must therefore intersect the inversion sphere. Moreover, the portion of $\Omega$ inside the sphere must be exactly the inversion image of the portion outside the sphere, while the intersection of $\Omega$ with the sphere itself remains invariant. This implies that $\Omega$ must cross the inversion sphere in a highly structured way: one part of the domain lies inside, one part lies outside, and these two parts are mapped onto each other by the inversion. This construction characterizes all possible counterexamples in which a nontrivial inversion acts as a domain automorphism. We visualize such configurations in Fig.6, where the interior and exterior components are exact inversion images of one another.

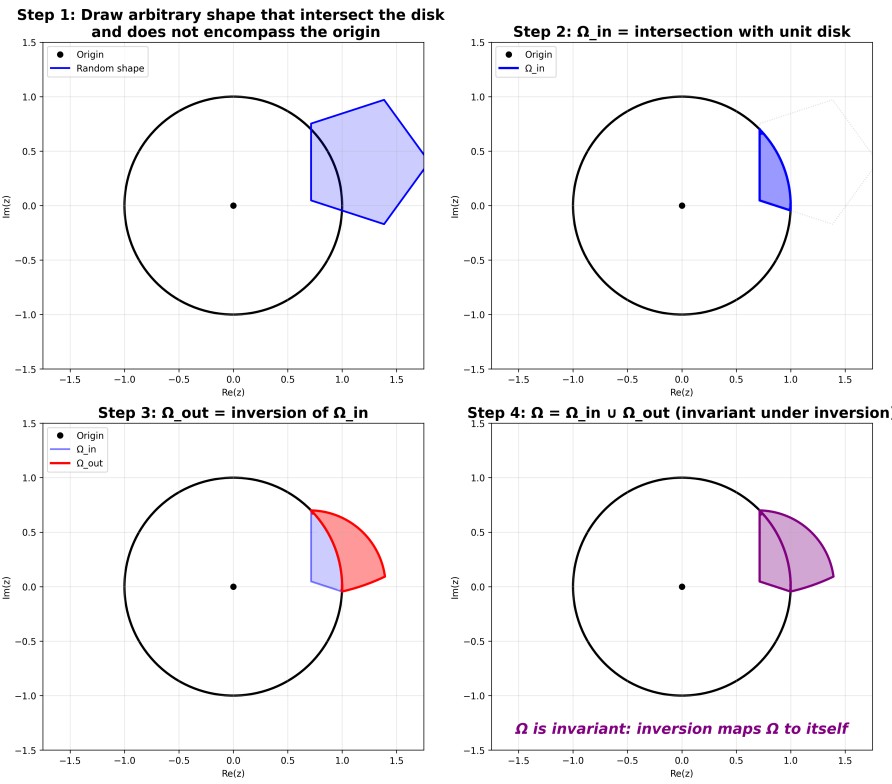

*Figure 6.* Method for constructing counterexamples. Any domain exhibiting inversion symmetry can be generated in this way.

Crucially, aside from the points lying exactly on the inversion sphere, no point can remain fixed. Thus, for any other point in $\Omega$, the existence of an automorphism involving inversion requires the existence of a corresponding point on the opposite side of the inversion sphere. It is precisely this requirement that is excluded by the geometric condition of Prop.3.

Indeed, the condition in Prop.3 ensures that for any potential inversion center $c$, there exist directions along which the radial structure of $\Omega$ is incompatible with inversion symmetry. Prop.3 excludes all domains that are invariant under an inversion. As a result, no inversion can map $\Omega$ onto itself. This situation is illustrated in Fig.7.

We conclude that the inversion case $\varepsilon = 2$ is ruled out under the assumptions of Prop.3. Therefore, any conformal automorphism of $\Omega$ must be affine, and hence a rigid symmetry of the domain. It follows that

$$\mathbf{g} = \mathbf{f} \circ \mathbf{h}, \tag{77}$$

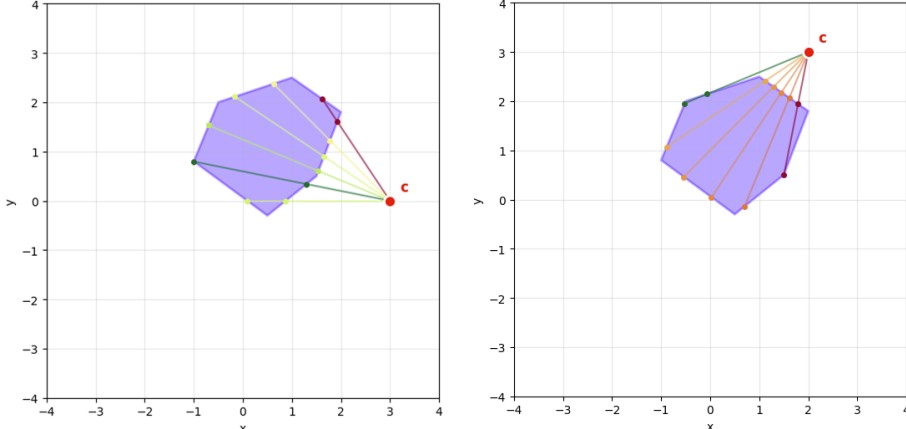

*Figure 7.* The condition in Prop.3 directly targets the defining geometric invariant of inversion. For a fixed center $c$, inversion about a sphere centered at $c$ preserves, along every ray direction $u$, the product of the two intersection distances $r_1(u)r_2(u)$ with the domain boundary. Domains that are invariant under inversion, such as disks or spheres, are characterized by the fact that this product is constant across all directions. Prop.3 explicitly excludes this possibility by requiring the existence of at least two directions for which the radial products differ, thereby breaking the invariance required for inversion symmetry. Since any Möbius transformation involving inversion must preserve this radial structure for some center $c$, the condition rules out all non-rigid Möbius automorphisms. Consequently, only affine conformal maps can act as automorphisms of the latent domain.

where $\mathbf{h}$ is a rigid transformation preserving $\Omega$. This establishes identifiability of the generative process up to the natural rigid symmetries of the latent domain, completing the proof of Prop.3.

Most latent domains do not admit invariance under inversion; only highly specific and strongly symmetric sets, such as perfect spheres or disks, possess this property. For such exceptional geometries, non-rigid Möbius symmetries may persist, leading to multiple valid generative solutions. We illustrate this phenomenon in Fig.8 with the case of a disk, whose inversion symmetry yields multiple admissible solutions, and in Fig.9 with a crescent-shaped domain, where inversion symmetry gives rise to exactly two distinct solutions. Importantly, such cases are rare: for the vast majority of finite, simply connected domains, inversion invariance does not hold. Consequently, identifiability extends well beyond the hypercube setting considered in the main text, demonstrating that disentanglement can be achieved for a broad class of latent supports whenever the domain geometry sufficiently constrains admissible conformal symmetries.

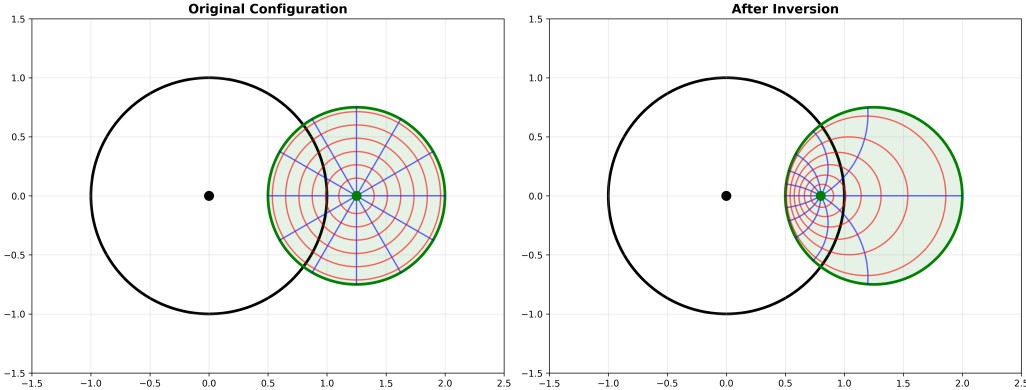

*Figure 8.* Visualization of a disk automorphism induced by an inversion. The disk is mapped onto itself, confirming that the transformation is indeed an automorphism. The mapping is nonlinear and produces a visibly deformed geometry.

### B.2. The Two-Dimensional Case

The two-dimensional case exhibits fundamentally different behavior from the higher-dimensional setting discussed above. This distinction arises from the markedly greater flexibility of conformal mappings in dimension $d = 2$, which contrasts

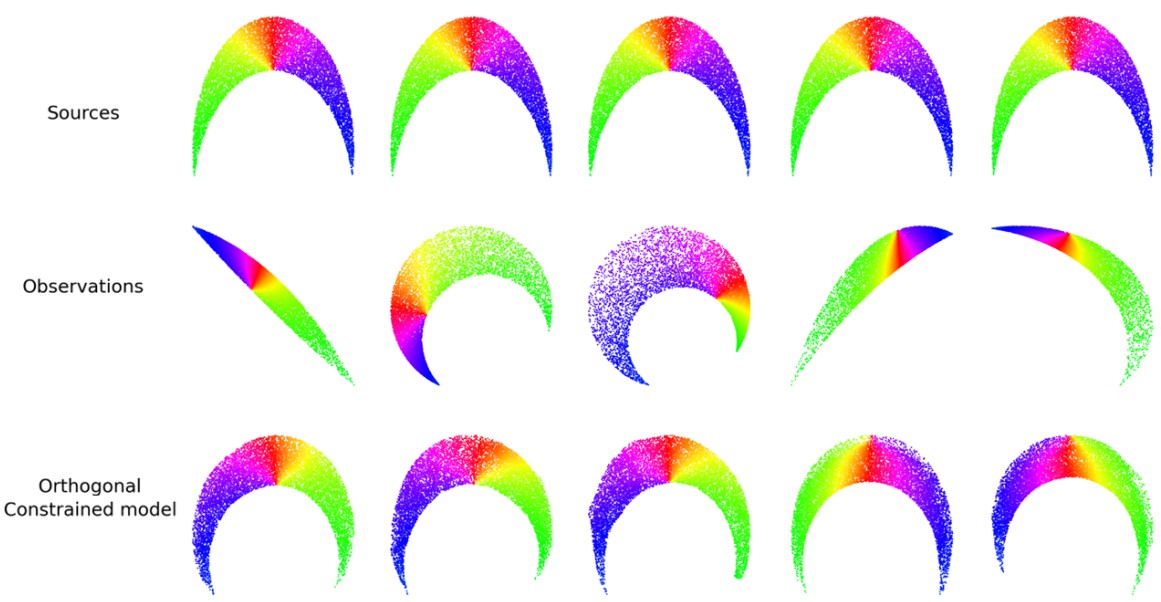

*Figure 9.* Qualitative disentanglement results for crescent-shaped domains. From top to bottom: ground-truth sources sampled from the structured domain, corresponding observations, and reconstructions obtained using the orthogonally constrained model. Due to inversion symmetry, two equivalent solutions are observed.

sharply with the rigidity imposed by Liouville's theorem for $d \geq 3$.

In two dimensions, conformal mappings coincide with holomorphic or anti-holomorphic functions with non-vanishing derivative. As a result, the space of conformal self-maps of planar domains is infinite-dimensional, rather than being restricted to Möbius transformations as in higher dimensions. This increased flexibility is formalized by the classical *Riemann Mapping Theorem* (RMT) ([Walsh, 1973](#)).

**Riemann Mapping Theorem.** The Riemann Mapping Theorem states that any simply connected, proper open subset $\Omega \subset \mathbb{C}$ (i.e., $\Omega \neq \mathbb{C}$) is conformally equivalent to the open unit disk $\mathbb{D}$. That is, there exists a bijective conformal map

$$\phi : \Omega \to \mathbb{D}. \tag{78}$$

Moreover, such a map is unique up to composition with a conformal automorphism of the disk, i.e., a Möbius transformation preserving $\mathbb{D}$.

This result implies that, in dimension two, any simply connected domain can be conformally deformed into a disk, regardless of its geometric shape. Consequently, unlike the case $d \geq 3$, domain geometry alone is insufficient to restrict conformal symmetries to rigid motions. The group of conformal automorphisms of the disk is infinite-dimensional, allowing for highly non-rigid transformations.

At first glance, this suggests that identifiability under conformal generative models should fail in two dimensions. However, this conclusion is overly pessimistic in the present setting, for several important reasons.

**1) Role of boundary behavior.** The Riemann Mapping Theorem applies to *open* domains and makes no guarantees about the behavior of the conformal map on the boundary $\partial\Omega$. In general, the conformal equivalence provided by the RMT does *not* extend continuously, let alone smoothly, to the boundary. As a result, many conformal maps that exist in the interior of a domain are incompatible with prescribed boundary correspondences.

In our setting, the generative mapping $\mathbf{f}$ is assumed to be smooth and invertible on an open neighborhood of the latent domain, which implicitly constrains its behavior near and on the boundary. When the boundary geometry of $\Omega$ is taken into

account—particularly when it contains salient geometric features such as corners, edges, or extremal points—the class of admissible conformal automorphisms is drastically reduced.

**2) Fixing boundary points and rigidity.**    A classical result in complex analysis states that a conformal automorphism of the unit disk is uniquely determined by the images of three distinct boundary points. Equivalently, any conformal automorphism of a planar domain that fixes three non-collinear points must be the identity.

This observation has direct implications for identifiability. If the latent domain $\Omega$ possesses three or more geometrically distinguished boundary points—such as the vertices of a triangle, the corners of a square or rectangle, or other extremal points—then any conformal automorphism preserving these points is necessarily rigid. In such cases, the freedom predicted by the Riemann Mapping Theorem disappears once boundary correspondence is enforced.

For example, if $\Omega$ is a polygonal domain, the interior angles at its vertices are conformal invariants. Any conformal automorphism that preserves the domain must therefore map vertices to vertices while preserving angles, which leaves only rigid motions as admissible transformations. Similar reasoning applies to domains with sufficiently rich boundary structure.

**Implications for identifiability in practice.**    Although two-dimensional conformal geometry admits greater theoretical flexibility, the combination of smoothness, invertibility, and implicit boundary constraints imposed by the generative model typically rules out non-rigid conformal automorphisms. As a result, even in $d = 2$, identifiability often holds up to rigid transformations for most practically relevant latent domains.

In this sense, the two-dimensional case represents a boundary regime: while conformal maps are theoretically more expressive, the conditions required for non-identifiability are fragile and rarely satisfied in structured or bounded domains. Thus, once boundary regularity or correspondence is taken into account, the identifiability conclusions obtained for $d \geq 3$ extend, in practice, to the two-dimensional setting as well.

This completes the discussion of the $d = 2$ case and highlights that the apparent loss of rigidity predicted by the Riemann Mapping Theorem does not fundamentally undermine identifiability in the present framework.

# C. Experimental Setup

This appendix details the experimental setup used throughout the paper, including dataset generation, ground-truth mixing functions, model architectures, training objectives, and evaluation metrics. All experiments are implemented in Python using PyTorch (Paszke et al., 2019), leveraging automatic differentiation to compute full Jacobians and enforce orthogonality constraints during training.

## C.1. Datasets and Generative Processes

To make the dataset for the experiments we start by generating data sources. We consider latent variables $\mathbf{z} \in \mathbb{R}^d$ with $d \in \{3, 6, 9\}$ and generate datasets under two distinct regimes: independent and dependent factors.

In the **independent setting**, latent variables are sampled i.i.d. from a uniform distribution on the hypercube,

$$\mathbf{z} \sim \mathcal{U}([0, 1]^d). \tag{79}$$

In the **dependent setting**, we generate complex statistical dependencies while preserving Asm.2 (full-support copula). We first sample $\boldsymbol{\epsilon} \sim \mathcal{N}(0, I)$ and transform it using an untrained autoregressive normalizing flow composed of 20 RealNVP-style coupling layers (Dinh et al., 2016). A coordinate-wise sigmoid nonlinearity maps the samples to $[0, 1]^d$. To ensure full support, we mix this distribution with a uniform component:

$$p_{\mathbf{z}} = \alpha \, p_{\mathrm{NF}} + (1 - \alpha) \mathcal{U}([0, 1]^d), \tag{80}$$

with $\alpha \in [0, 1]$. This guarantees a minimum density of $1 - \alpha$ everywhere on the hypercube, hence full support. Typically $\alpha = 0.75$ is chosen. This construction explicitly violates independence while satisfying Asm.2. For each configuration, we sample $10\,000$ latent points.

Once we have the sources, latent variables are then mapped to observations $\mathbf{x} \in \mathbb{R}^n$ using smooth, invertible transformations satisfying Asm.1.

We consider two classes of mixing functions. First, we use **conformal mappings** based on generalized Möbius transformations,

$$f(\mathbf{z}) = \mathbf{b} + \frac{\alpha\, A(\mathbf{z} - \mathbf{a})}{\|\mathbf{z} - \mathbf{a}\|^{\varepsilon}}, \tag{81}$$

where $A$ is orthogonal, $\alpha > 0$, $\mathbf{a}, \mathbf{b} \in \mathbb{R}^d$, and $\varepsilon = 2$. This choice yields non-isometric but conformal transformations, preserving angles while allowing local scaling.

Second, to test identifiability beyond the conformal case, we construct a broad class of nonlinear mixing functions whose Jacobians are of the form $J_f(\mathbf{z}) = Q(\mathbf{z})D(\mathbf{z})$, but which are not globally conformal. These mappings are generated compositionally, exploiting the fact that the composition of $QD$ transformations remains $QD$ if composed block-wise.

Concretely, we partition the latent space into blocks of three dimensions. Within each block, we first apply a nonlinear two-dimensional transformation acting on the first two coordinates, chosen among polar, elliptic, parabolic, or rigid transformations. This 2D mapping is then composed with a 2D Möbius transformation, yielding a flexible yet analytically tractable deformation with orthogonal Jacobian structure. The resulting 2D surface is subsequently embedded into three dimensions, either by directly appending the third coordinate or by constructing a surface of revolution around it. In the latter case, a point $(u, v, \phi)$ is mapped according to

$$x = X(u,v)\cos\phi, \qquad y = X(u,v)\sin\phi, \qquad z = Z(u,v), \tag{82}$$

where $(X(u,v), Z(u,v))$ denotes the output of the preceding 2D transformation. Each three-dimensional block is then further composed with a 3D Möbius transformation. After processing all blocks independently, we apply an additional global Möbius transformation across all dimensions to induce cross-block interactions and ensure full mixing. At each stage, transformation parameters are sampled within bounded ranges to avoid singularities (e.g., degeneracies associated with Möbius inversions), and intermediate outputs are normalized to ensure comparable scales across dimensions. Examples of transformations from the hypercube are provided in Fig.10

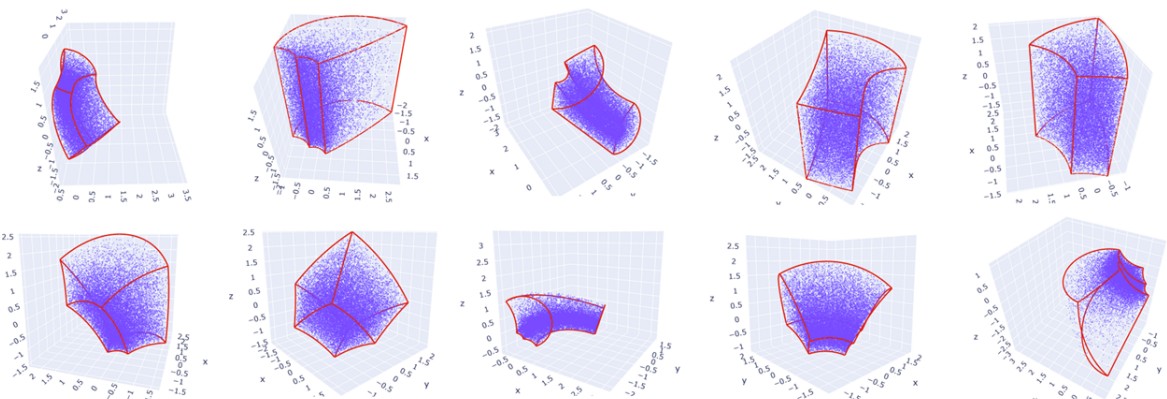

*Figure 10.* Example of transformations in 3D.

This construction yields highly expressive nonlinear mixing functions with orthogonal Jacobians that go far beyond simple coordinate reparameterizations such as spherical or cylindrical coordinates. While Möbius transformations alone form a relatively restricted subclass, their block-wise and hierarchical composition produces a much larger family of admissible $QD$ mappings. As a result, these generative processes are substantially more challenging to disentangle than purely conformal ones, providing a stringent empirical test of the proposed identifiability theory.

### C.2. Evaluation Metrics

We evaluate both density estimation accuracy and disentanglement quality.

Since the true data-generating distribution is known, we compute the **Kullback–Leibler divergence** between the true distribution and the learned model by evaluating likelihoods using the inverse ground-truth mapping. Concretely, since

we have access to the ground-truth mixing function $\mathbf{f}$ and can compute its Jacobian via automatic differentiation, the true data log-density is tractable at any point. The learned normalizing flow provides a tractable model density. The KL divergence is then estimated by a standard Monte Carlo average over samples from the true distribution. This metric purely assesses goodness of fit and is not used to measure disentanglement. All models achieve comparable KL values, confirming successful density estimation.

Disentanglement is assessed using the **Mean Correlation Coefficient (MCC)** (Khemakhem et al., 2020). We compute the pairwise correlation matrix between ground-truth latent variables and recovered representations, solve a maximum-weight assignment problem, and report the mean of the matched correlations. We use Spearman correlation (Hauke & Kossowski, 2011), which is invariant to monotonic coordinate-wise reparameterizations, consistent with Def.1.

We also report the **nonlinear Amari distance** introduced by Gresele et al. (2021) that directly compares the Jacobians of the true mixing and the learned unmixing. Given the true mixing $f$ and learned unmixing $g$, it is defined as

$$d_{\text{n-Amari}}(g, f) = \mathbb{E}_{\mathbf{x} \sim p_{\mathbf{x}}} \Big[ d_{\text{Amari}} \big( J_g(\mathbf{x}) \, J_f(f^{-1}(\mathbf{x})) \big) \Big]. \tag{83}$$

where $d_{\text{Amari}}$ is the classical Amari distance metric used in linear ICA (Amari et al., 1995). This metric equals zero if and only if $g \circ f$ differs from the identity by a permutation and coordinate-wise reparameterization (i.e., the product of Jacobians is a scale and permutation matrix), exactly matching the equivalence class of Def.1. Together, MCC and nonlinear Amari distance quantify whether the learned representation recovers the true factors up to the admissible ambiguities.

### C.3. Models, Architectures, and Training Objectives

Given observations $\mathbf{x}$, the objective is to recover latent representations corresponding to the unknown sources, without access to the true generative factors. To this end, we train expressive normalizing flows on the observed data and evaluate whether the resulting latent variables exhibit a disentangled structure.

Our main architecture is a **Residual Flow** (Chen et al., 2019), defined by the transformation

$$\mathbf{y} = f(\mathbf{x}) = \mathbf{x} + h(\mathbf{x}), \tag{84}$$

where $h : \mathbb{R}^d \to \mathbb{R}^d$ is a neural network constrained to be Lipschitz-continuous, which guarantees invertibility and ensures that the Jacobian

$$J_f(\mathbf{x}) = I + J_h(\mathbf{x}) \tag{85}$$

is full and expressive. Residual flows are universal density approximators and allow for highly flexible transformations while retaining tractable log-determinants.

We use 64 residual flow layers, each parameterized by a 3-layer multilayer perceptron with hidden dimension 128. Models are trained using Adam with learning rate $10^{-3}$ and batch size 256 until convergence.

We consider two settings depending on whether the latent sources are independent or dependent. In the *independent* setting, the base distribution of the residual flow is a logistic distribution with diagonal scale matrix,

$$p_{\text{base}}(\mathbf{z}) = \text{Logistic}(\mathbf{0}, I), \tag{86}$$

which provides full support on $\mathbb{R}^d$ and matches standard assumptions in nonlinear ICA.

In the *dependent* setting, we allow the base distribution to be complex and learned. Specifically, we train an additional normalizing flow to model the prior distribution. This prior flow consists of three affine coupling layers, each implemented by a 3-layer MLP with hidden size 64. To guarantee full support and avoid degeneracies, we define the base distribution as a mixture

$$p_{\text{base}} = \beta \, p_{\text{logistic}} + (1 - \beta) \, p_{\text{prior NF}}, \qquad \beta \in (0, 1). \tag{87}$$

This construction exploits the fact that pushforwards preserve mixtures:

$$f_{\#}(\beta P + (1 - \beta) Q) = \beta \, f_{\#} P + (1 - \beta) \, f_{\#} Q. \tag{88}$$

As a result, the learned latent distribution is a mixture between a flexible learned component and a simple reference distribution. Intuitively, after applying a sigmoid map, the logistic component corresponds to a uniform distribution on $[0, 1]^d$, ensuring non-vanishing density everywhere. This guarantees full support in the latent space and therefore enforces Asm.2 inherently during training.

It is important to note that the base distribution lives on $\mathbb{R}^d$, whereas the true sources are supported on $[0, 1]^d$. Consequently, even in the ideal case, recovery is only possible up to coordinate-wise reparameterizations. The learned latent variables are therefore not expected to match the source distribution directly. Sigmoid mappings to $[0, 1]^d$ are applied solely for visualization purposes and are never used during training.

Unconstrained models are trained by maximizing the standard log-likelihood

$$\log p(\mathbf{x}) = \log p(g(\mathbf{x})) + \log|\det J_g(\mathbf{x})|. \tag{89}$$

Orthogonally constrained models use the regularized objective proposed by Gresele et al. (2021),

$$\mathcal{L} = \log p(\mathbf{x}) - C_{\text{IMA}}, \qquad C_{\text{IMA}} = \sum_{i=1}^{d} \log \|[J_{g^{-1}}]_i\| - \log|\det J_{g^{-1}}|. \tag{90}$$

This penalty is non-negative and vanishes if and only if the Jacobian is of the form $QD$, i.e., its columns are mutually orthogonal up to scaling. Geometrically, the columns of the Jacobian $J_g$ correspond to the partial derivatives $\partial g/\partial s_i$. The determinant $|J_g|$ measures the volume of the $d$-dimensional parallelepiped spanned by these vectors, while the product of their norms corresponds to the volume of an axis-aligned box with side lengths $\|\partial g/\partial s_i\|$. These two volumes coincide if and only if the columns are orthogonal (Gresele et al., 2021). Minimizing $C_{\text{IMA}}$ therefore explicitly pushes the Jacobian toward an orthogonal-diagonal structure.

In our experiments, orthogonality is enforced by recovering the full Jacobian of the learned unmixing $g$ via automatic differentiation by feeding canonical basis vectors through vector-Jacobian products using PyTorch `autograd`, and then applying the CIMA regularization term to the resulting columns. This procedure guarantees that the orthogonality condition is effectively enforced during training and that no confounding factors are introduced. While this allows us to precisely control the conditions and observe the effect of orthogonality, it requires computing the full $d \times d$ Jacobian at each training step, which scales poorly with the dimensionality of the latent space and renders direct application to high-dimensional observations such as images computationally prohibitive. In practice, efficiently enforcing orthogonal Jacobians in a scalable way is still an open research question, one that goes beyond the scope of this paper.

Overall, we train expressive residual flows that accurately model the data distribution. In the orthogonal setting, the additional regularization enforces a $QD$ Jacobian structure, aligning the learned transformation with the assumptions of our theoretical framework. Despite the complexity of the data-generating process, this approach yields disentangled latent representations up to coordinate-wise reparameterization.

### C.4. Boundary Behavior and Optimization Issues

In theory, identifiability requires the learned latent distribution to exactly match the assumed prior and, in particular, to satisfy Asm.2 on the support of the latent variables. In practice, however, optimization and finite model capacity can prevent this condition from being met. Even when using a prior with full support, training may converge to suboptimal solutions in which the learned latent domain folds onto itself. Such configurations can achieve relatively high likelihood, remain close to the target prior, and yet violate the assumptions required by the theory. Once trapped in these configurations, continued training typically results in further local deformations of the latent space rather than convergence to the true prior, explaining most of the observed failure cases.

This also explains the observed degradation in MCC for higher latent dimensions ($d = 9$): as dimensionality increases, the $C_{\text{IMA}}$ loss becomes harder to satisfy globally, and the optimization landscape grows more complex, making it more likely to converge to solutions where the latent domain is locally consistent with orthogonality but globally distorted. This is a limitation of the practical implementation rather than of the theory itself.

This gap highlights a fundamental difference between the theoretical setting, which assumes exact matching of the prior, and practical optimization, where approximate matching may be insufficient. To better align practice with theory, we introduce an additional mechanism to explicitly enforce correct domain geometry. The key observation is that for smooth invertible

mappings, the boundary of a domain must be mapped to the boundary, and interior points must be mapped to interior points. Consequently, by enforcing correct behavior on the boundary, one indirectly constrains the entire interior of the domain.

Operationally, we identify samples lying close to the empirical boundary of the latent domain and introduce an auxiliary loss that penalizes mappings sending these points outside the expected support. This boundary-aware regularization encourages the learned transformation to respect the global geometry of the domain and prevents self-intersections and holes. While this procedure requires mild prior knowledge about the data domain, it is only used to ensure that the assumptions of the identifiability theorem are met. Importantly, the method can still succeed without this regularization, but in such cases training is more prone to becoming trapped in suboptimal solutions that fall outside the theoretical regime we aim to validate empirically.

### C.5. Experiments with Varying Latent Domains

As discussed in Sec.4 and further detailed in App.B, when the latent support is known, the generative process becomes identifiable, up to the natural rigid symmetries of that support. To support that analysis, we perform additional experiments where the latent support is known but non-cubical. Latent variables are sampled uniformly from various bounded domains and mixed using conformal Möbius transformations. For each configuration, $10\,000$ samples are generated. The task of the model is to recover the original latent structure solely from the observations.

For these experiments, we use Neural Spline Flows (Durkan et al., 2019), which parameterize monotonic rational-quadratic spline transformations and provide high expressivity with stable training. The model consists of 12 flow layers, each implemented by a 3-layer MLP with hidden size 256, using 27 spline bins and a tail bound of 12.

To avoid optimization issues associated with sharp, bounded supports, we design a shape-aware base distribution that matches the geometry of the true latent domain. Specifically, the base density is defined as a soft-edged distribution whose density is proportional to

$$\exp\left(-\frac{\text{dist}(\mathbf{z}, \Omega)^2}{2\varepsilon^2}\right), \tag{91}$$

where $\Omega$ denotes the target latent domain and $\varepsilon$ controls boundary sharpness. For large values of $\varepsilon$, the distribution has very smooth edges and provides gradients far outside the domain, while for small $\varepsilon$ it approaches a uniform distribution over $\Omega$. Points lying outside the domain experience strong gradients pulling them inward. During training, we progressively decrease $\varepsilon$, making the boundary increasingly sharp and, at the end of training, enforcing the target domain almost exactly. Orthogonality regularization is applied as in previous experiments.

Overall, this setup corresponds to a standard Neural Spline Flow trained with a specially designed base distribution encoding the known latent support, combined with the same orthogonality regularization used in previous experiments. This controlled setting allows us to systematically investigate the role of domain knowledge in identifiability and to empirically validate the theoretical predictions under varying and nontrivial latent geometries.

## D. Additional Results and Experiments

In this appendix, we present additional quantitative and qualitative results complementing the experiments reported in the main paper.

### D.1. Quantitative Comparison

The primary goal of our experiments was to validate our theoretical results, not to establish normalizing flows as state-of-the-art disentanglement models. Flows were used because they satisfy the conditions of the propositions. Since VAEs already incorporate an implicit orthogonal bias (Sec.6), including them as a baseline was less central to our theoretical validation. Nonetheless, we provide here a comparison for $d = 6$ with independent latent variables, since the VAE prior imposes independent terms, making this the most natural setting for such a comparison.

Our initial choice of Amari distance and MCC was deliberate and aligned with prior work (Gresele et al., 2021), where these metrics are commonly used and sufficient to assess whether latent factors are successfully recovered. To further strengthen our evaluation, we additionally report standard disentanglement metrics: FactorVAE (Kim & Mnih, 2018), MIG (Chen et al., 2018), SAP (Kumar et al., 2017), and DCI (Eastwood & Williams, 2018).

Results are reported in Table 2. As expected from the analysis in Sec. 6, VAEs achieve stronger disentanglement than unconstrained flows, likely due to their inherent orthogonal bias, but perform worse than constrained normalizing flows, likely due to the information preference property and the detrimental MI term. Across all additional metrics, the conclusions are consistent with those drawn from Amari distance and MCC, confirming that the learned representations are well disentangled and supporting the central claims of this work.

$\beta$-TCVAE obtains results closer to constrained NFs than standard VAEs, which is consistent with our explanation in Sec. 6. Indeed, $\beta$-TCVAE reduces or removes the MI term while retaining (i) the total correlation penalty and (ii) a factorized posterior, thereby alleviating posterior collapse without affecting the two mechanisms responsible for disentanglement: global independence and local orthogonality. This further supports our claim that disentanglement does not arise from MI minimization, but from the joint imposition of these two inductive biases, and that they can be made explicit and transferred beyond VAEs through orthogonality regularization. This also clarifies the comparison with unconstrained NFs: despite sharing the same total correlation pressure as VAEs, they lack the implicit orthogonality constraint induced by the factorized posterior, explaining why adding an explicit orthogonality regularization is sufficient to recover strong disentanglement.

*Table 2.* Averaged disentanglement metrics across experiments.

| Metric d=6, ind. | NF Unconstrained | NF Constrained (Ortho.) | VAE | $\beta$-TCVAE |
|---|---|---|---|---|
| MCC_spearman | $0.7091 \pm 0.0492$ | $0.9575 \pm 0.0544$ | $0.8213 \pm 0.0868$ | $0.8880 \pm 0.0759$ |
| MCC_pearson | $0.7086 \pm 0.0496$ | $0.9510 \pm 0.0545$ | $0.8138 \pm 0.0846$ | $0.8799 \pm 0.0785$ |
| MIG | $0.0760 \pm 0.0389$ | $0.6138 \pm 0.1429$ | $0.2011 \pm 0.0907$ | $0.3279 \pm 0.1201$ |
| SAP | $0.0266 \pm 0.0133$ | $0.1356 \pm 0.0236$ | $0.0609 \pm 0.0245$ | $0.1078 \pm 0.0348$ |
| DCI_D | $0.1474 \pm 0.0596$ | $0.8887 \pm 0.1346$ | $0.4819 \pm 0.1536$ | $0.6766 \pm 0.1665$ |
| DCI_C | $0.1545 \pm 0.0623$ | $0.8898 \pm 0.1326$ | $0.5106 \pm 0.1299$ | $0.6824 \pm 0.1469$ |
| DCI_I | $0.9753 \pm 0.0049$ | $0.9978 \pm 0.0019$ | $0.9589 \pm 0.0456$ | $0.9734 \pm 0.0335$ |
| FactorVAE | $0.7195 \pm 0.1279$ | $0.9770 \pm 0.0629$ | $0.8463 \pm 0.1182$ | $0.9040 \pm 0.0893$ |

### D.2. Qualitative Results

Figs.11 and 12 show representative reconstructions of latent variables in the case of *independent* sources, comparing unconstrained normalizing flows with their orthogonally constrained counterparts. Similarly, Figs.13 and 14 report qualitative results for the *dependent* sources setting.

Across all cases, the results are consistent with the quantitative boxplot comparisons presented in Fig.3. Models trained with orthogonality constraints on the Jacobian are generally able to recover the underlying sources up to the expected ambiguities, whereas unconstrained models, while successfully learning the data distribution, typically fail to disentangle the true generative factors. Beyond confirming the behavior observed in the main experiments, these results demonstrate that our approach is not limited to conformal transformations or to independent source models, as considered in much of the previous literature. Instead, it successfully disentangles substantially more complex generative processes, including dependent factors and highly nonlinear, non-conformal mixings. This highlights that enforcing an orthogonal Jacobian is not merely an auxiliary regularization or architectural constraint, but rather captures a fundamental structural property of latent factors underlying identifiability in these settings.

In addition, we report results for latent domains with shapes other than the hypercube, demonstrating that our approach extends beyond standard axis-aligned supports. The experimental setup for these experiments is described in App.C.5, and the corresponding qualitative results are shown in Figs15 and 16. As predicted by the theory (Sec.4 and App.B), the learned representations recover the sources up to the rigid symmetries of the domain, namely global rotations and reflections. This confirms that identifiability holds whenever the latent support is known, even for nontrivial geometries.

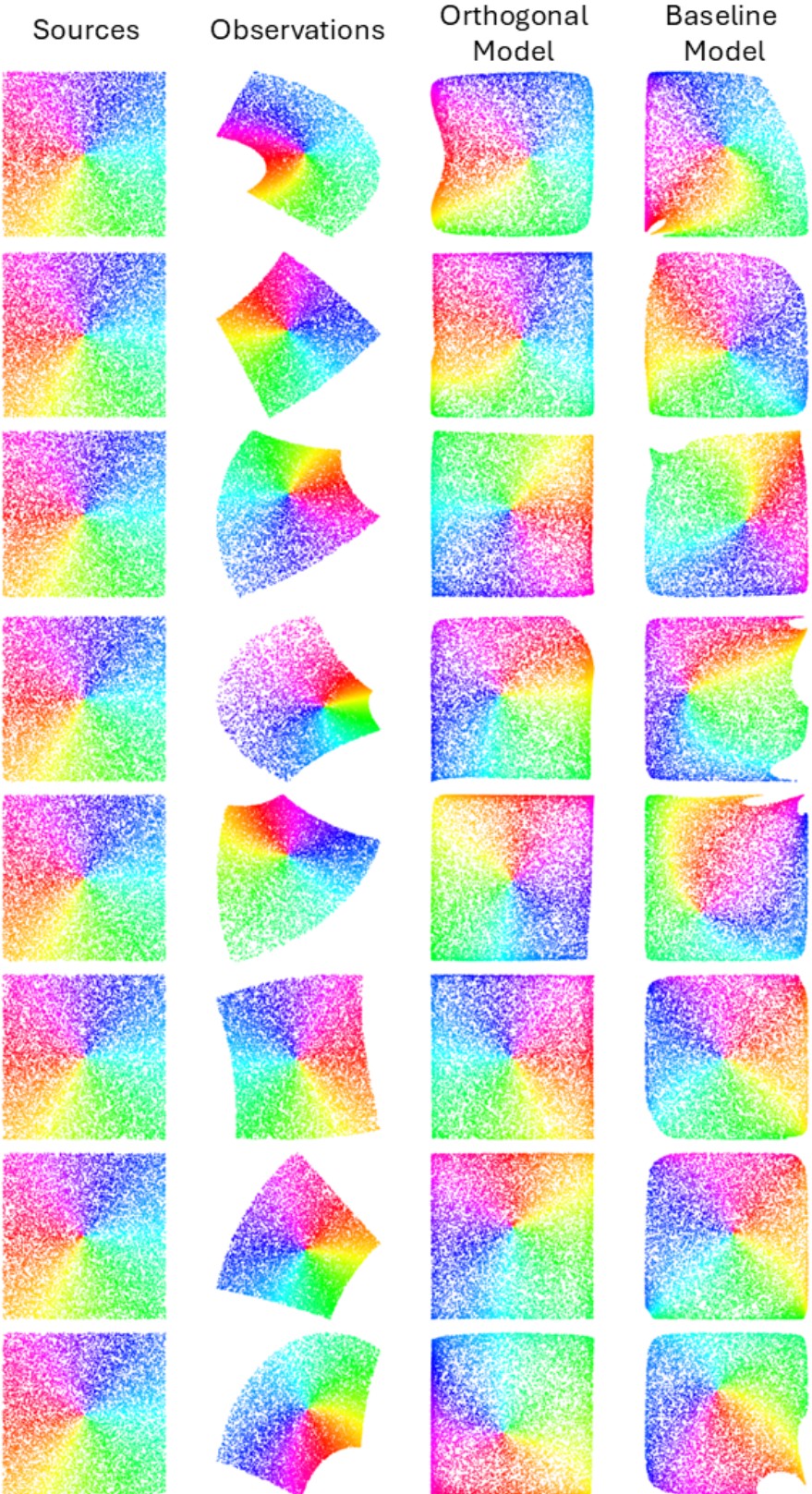

*Figure 11.* Qualitative disentanglement results for independent sources. From left to right: ground-truth sources, observed variables, latent reconstructions obtained with the orthogonally constrained model, and latent reconstructions obtained with the unconstrained model. Reconstructed latents are visualized after a sigmoid transformation for clarity.

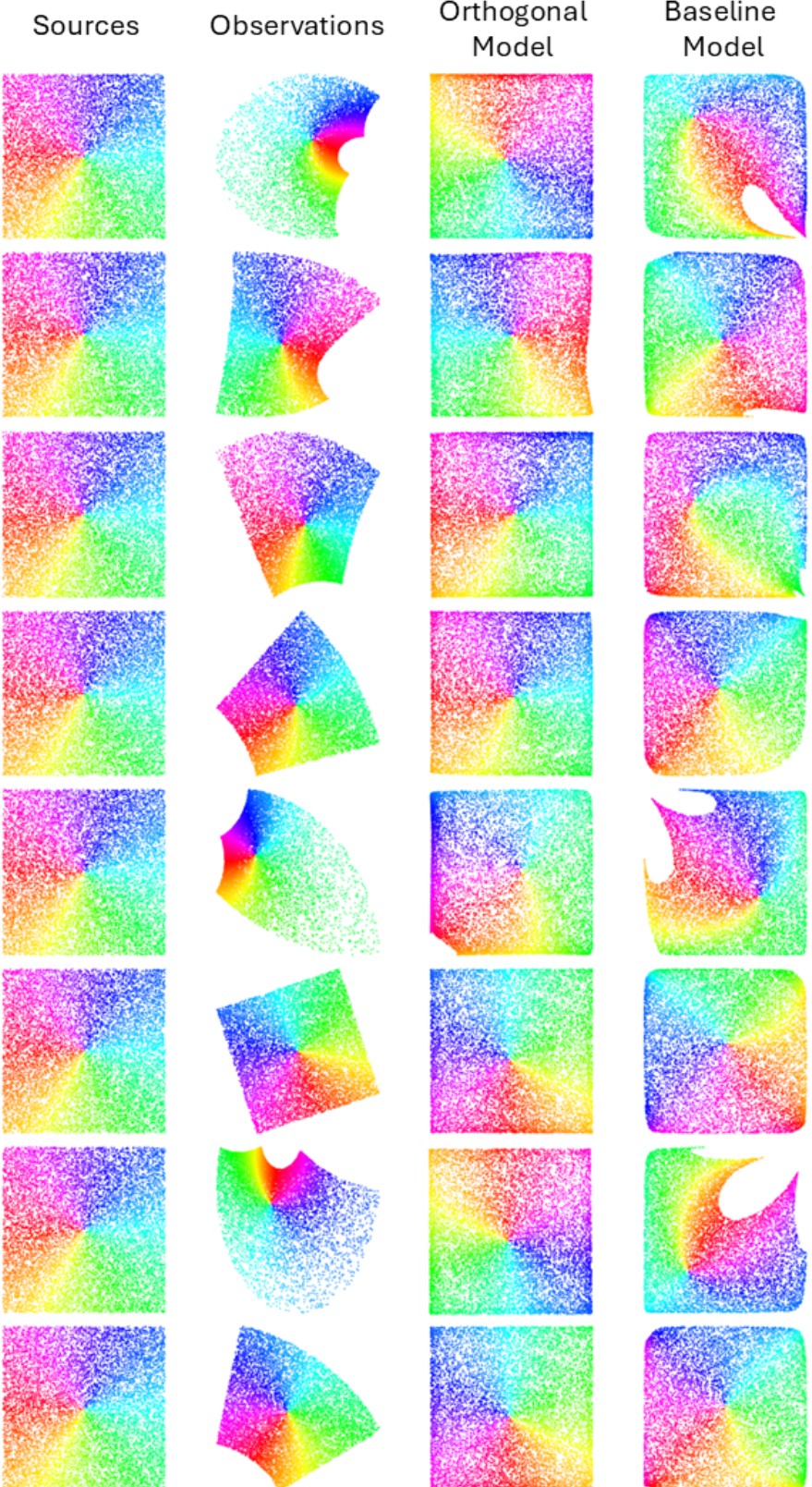

*Figure 12.* Additional qualitative results for independent sources. Columns correspond to ground-truth sources, observations, reconstructions from the constrained model, and reconstructions from the unconstrained model, respectively. Reconstructions are shown after sigmoid reparameterization.

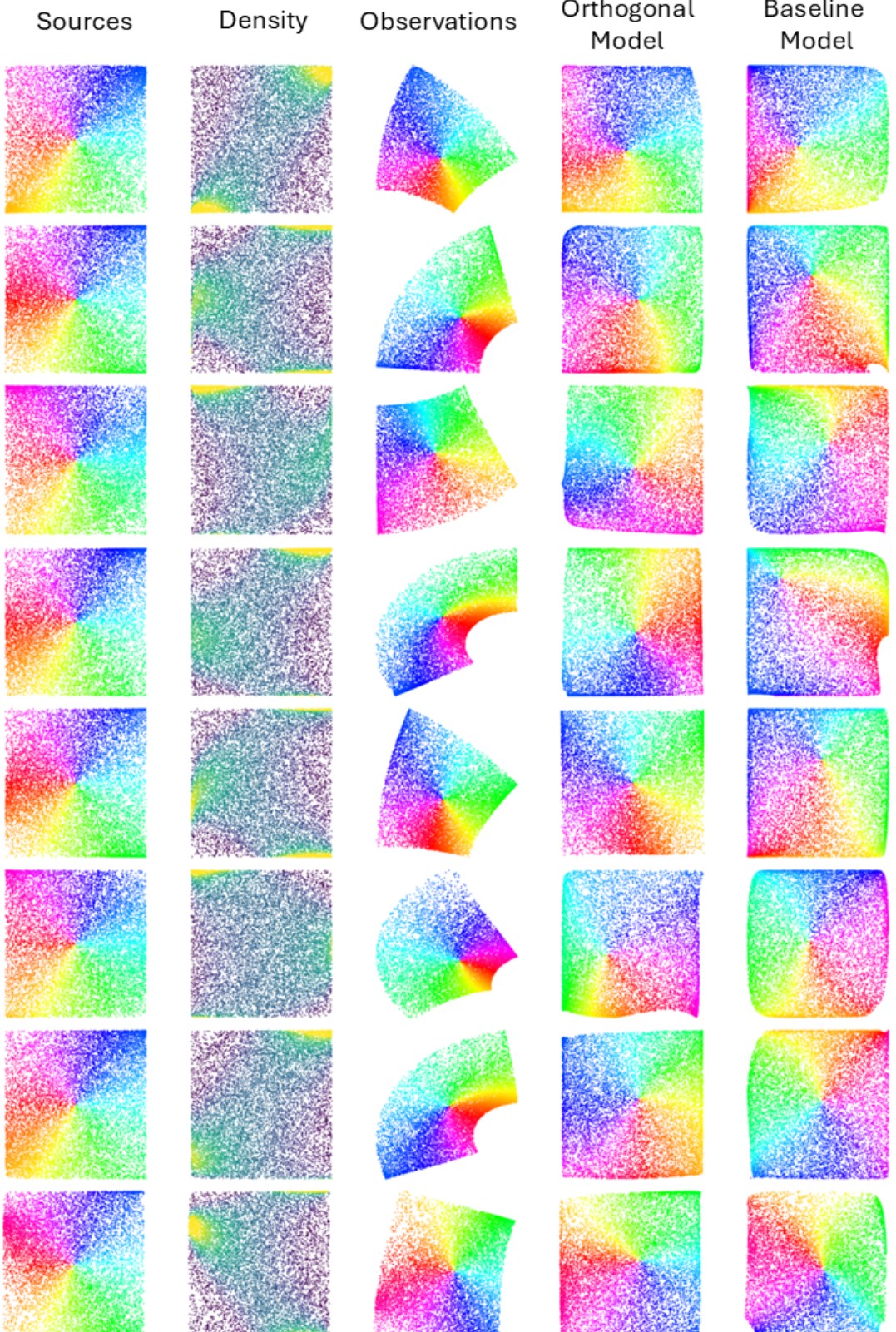

*Figure 13.* Qualitative disentanglement results for dependent sources. From left to right: ground-truth sources, source density, observed variables, latent reconstructions obtained with the orthogonally constrained model, and latent reconstructions obtained with the unconstrained model. Reconstructions are visualized after sigmoid transformation.

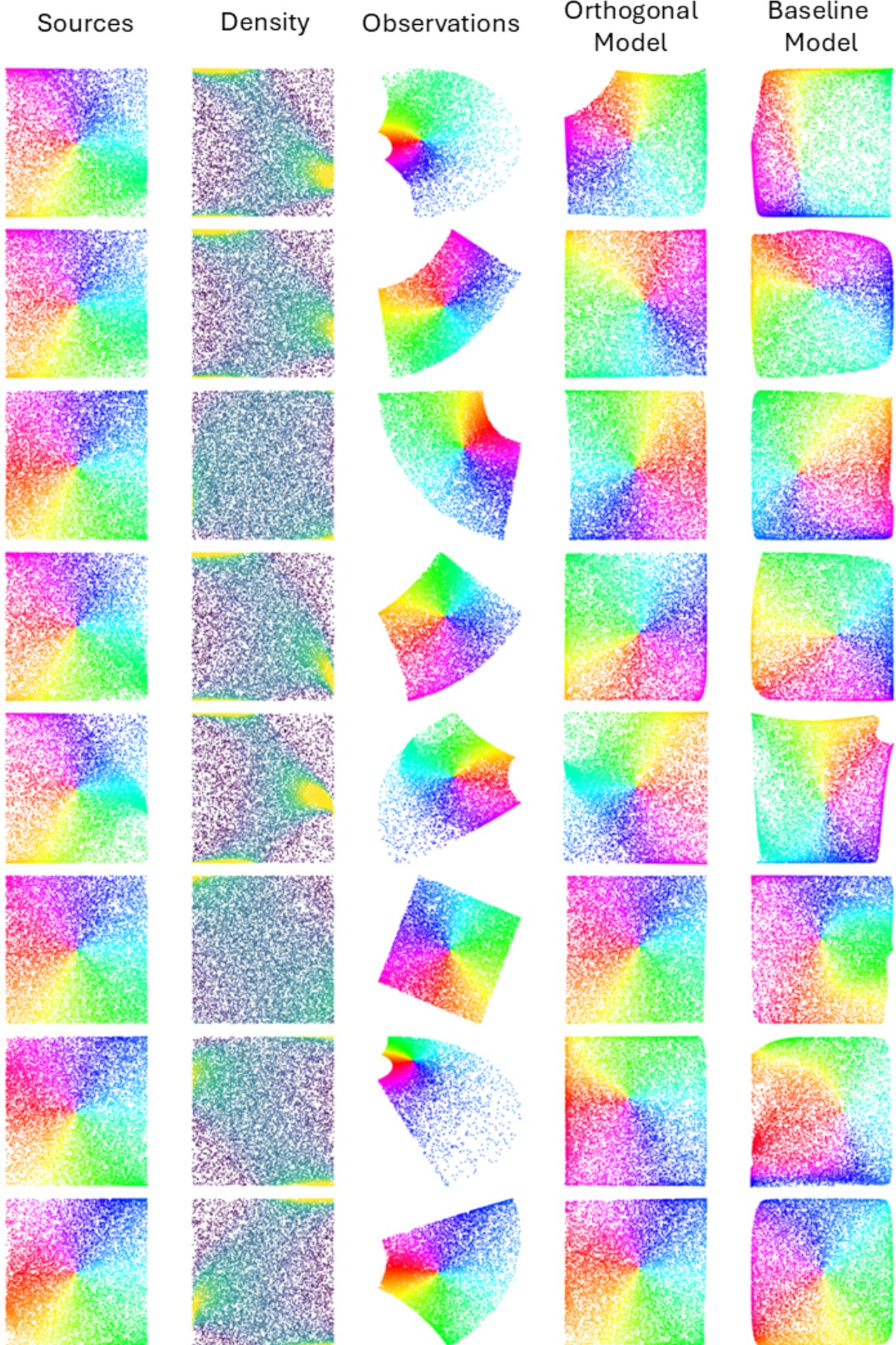

*Figure 14.* Additional qualitative results for dependent sources. Columns show ground-truth sources, source density, observations, and latent reconstructions from the constrained and unconstrained models, respectively.

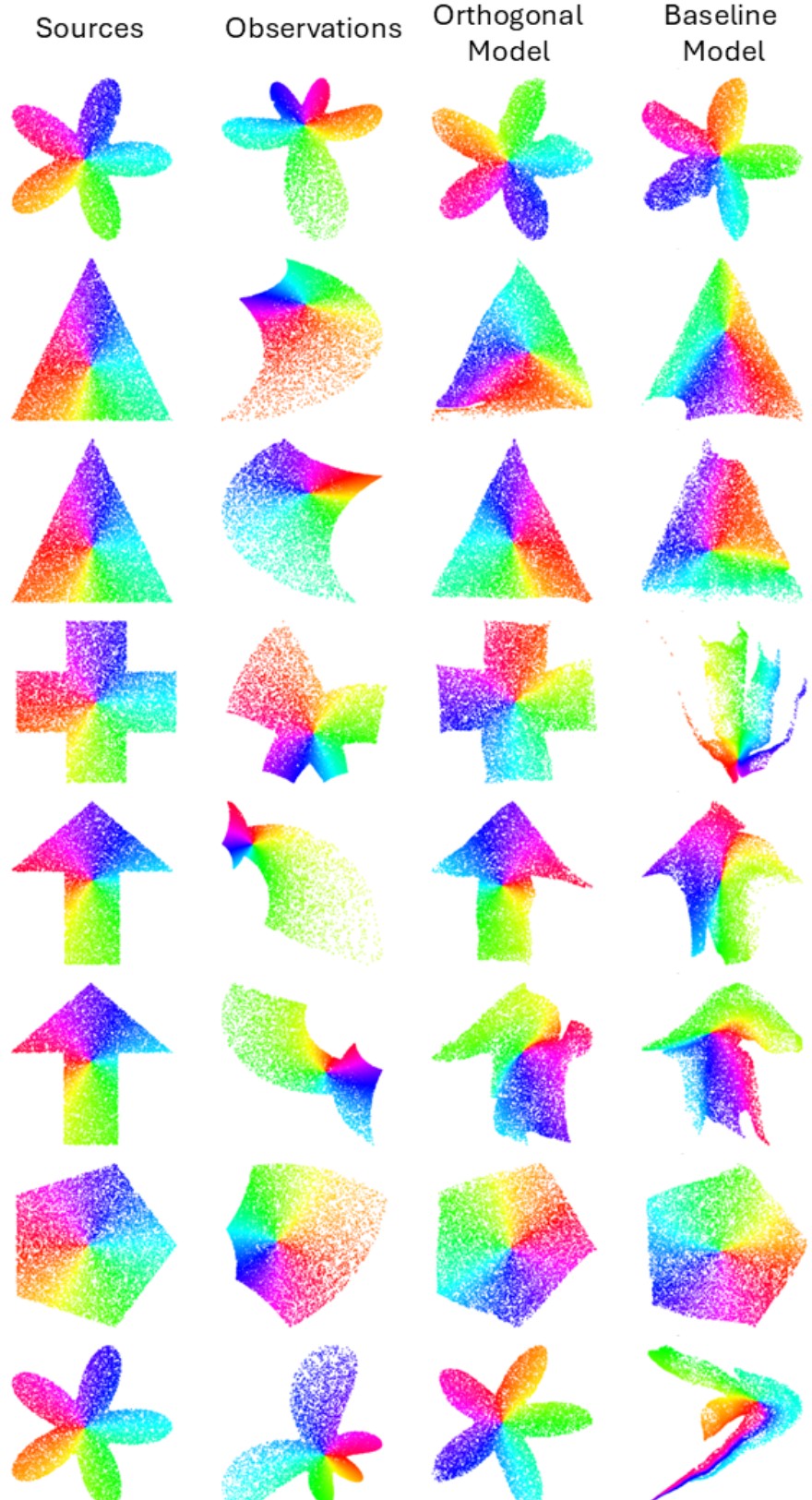

*Figure 15.* Qualitative disentanglement results for non-cubical latent domains. From left to right: ground-truth sources sampled from a structured domain, observations, reconstructions obtained with the orthogonally constrained model, and reconstructions obtained with the unconstrained model.

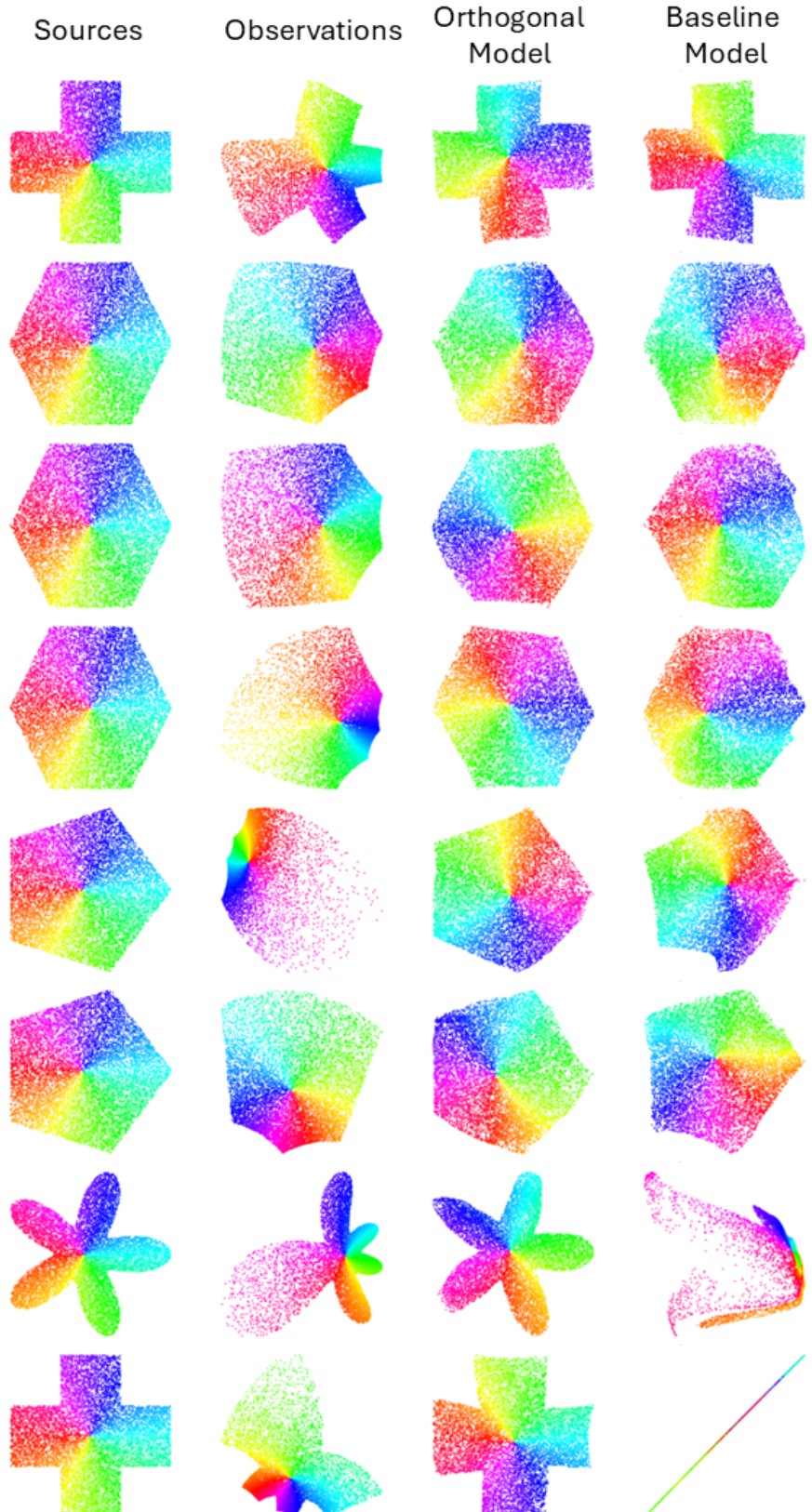

*Figure 16.* Additional results for non-cubical latent domains. The constrained model successfully recovers the sources up to rigid transformations of the domain, while the unconstrained model fails to preserve the latent geometry.

