# OpenReview forum: "Unsupervised Disentanglement Without Compromises : How Functional Orthogonality Enforces Identifiability"
_ICML.cc/2026/Conference — ICML 2026 regular_

### Official Review · Reviewer_5uqT · 2026-02-27

**Soundness:** 2
**Presentation:** 4
**Significance:** 3
**Originality:** 4
**Overall Recommendation:** 5
**Confidence:** 4

**Summary:**

This paper studies the disentanglement representations without assuming independent latent variables or supervision information, for an arbitrary smooth and invertible mixing function. It provides two crucial conditions to ensure the identifiability of latent concepts: 1. the orthogonality condition of the mixing function, 2. The domain of latent is a hypercube.  Experiments on synthetic data validate that the orthogonality condition is sufficient to identify latent variables.

**Compliance With Llm Reviewing Policy:**

Affirmed.

**Final Justification:**

I keep my positive assessment of this work

**Key Questions For Authors:**

1.	In the problem setting, the joint distribution of $z$ is modeled recursively into a factorization following some topological order. Does this structural assumption play a crucial role in the identifiability results? Or do the results still hold under arbitrary dependencies among the components of $z$? The latter setting seems more general, as it would also cover cases where no clear topological ordering exists among the latent variables.

2.	I understand this is a strong theoretical work, and it is still far from applying it to an application. Is there any way to know how realistic the dataset satisfies assumptions 1 or 2?

**Limitations:**

See above

**Strengths And Weaknesses:**

## Strengths

**Significance**: This work addresses an important and fundamental problem in disentangled representation learning. Although the proposed framework may still be far from practical applicability in real-world scenarios, the orthogonality principle it introduces is theoretically inspiring and offers meaningful insights into the foundations of identifiability.

**Originality**: To the best of my knowledge, this paper presents a novel perspective on the conditions required to ensure identifiability of dependent latent concepts. While prior works have achieved identifiability by imposing structural constraints on the latent support or on the Jacobian of the mixing function, this is the first study to systematically examine both aspects in depth and clarify how they jointly contribute to identifiability. This comprehensive theoretical analysis clearly distinguishes the paper and represents a valuable contribution.

**Presentation**: The paper is well structured and clearly written. I particularly enjoyed reading the theoretical sections. The logical flow is smooth, and the results are developed layer by layer. I especially appreciate the authors’ effort to provide intuition behind Assumptions 1 and 2, which are explained clearly and thoughtfully.

## Weaknesses

**Soundness of empirical evaluation**: Although I did not verify the proofs in full detail, the intuition behind the main theorems seems sound, and the theoretical analysis is carried out rigorously. However, there is a noticeable gap between the theoretical results and the empirical findings. Even on synthetic datasets, the performance is not perfect. For example, the MCC is lower than 0.9 when the latent dimension increases to 9 only. One possible explanation is that the CIMA loss used to enforce orthogonality may be insufficient to fully satisfy the theoretical conditions in practice. It would be better if the authors explicitly discussed this gap and provided further analysis of the relationship between theory and empirical performance.

---

> ### Author Rebuttal · Authors · 2026-03-27
>
> **Given the strict 5000-character limit, this is a highly condensed answer. Please feel free to ask for details. See the general message for reviewers in hvD5.**
>
> **Empirical performance:** We thank the reviewer for highlighting the observed gap for higher latent dimensions. Our analysis indicates that this gap primarily arises from *optimization challenges rather than a limitation of the theory*. In practice, the orthogonality condition is enforced through the $C_{\text{IMA}}$ loss, which only approximates the ideal constraint. As the dimensionality increases, this constraint becomes significantly harder to satisfy globally. In particular, in higher dimensions the optimization landscape becomes more complex, and the model can converge to suboptimal local minima. A typical failure mode we observe is that the learned latent representation remains locally consistent with the orthogonality constraint, but the global structure of the latent domain becomes distorted (e.g., it may "fold onto itself"). This violates the theoretical requirement that the learned representation matches the full support of the latent domain, which is crucial for identifiability. As a consequence, the recovered factors are only partially aligned with the ground truth, leading to a degradation in MCC. Importantly, this behavior is not specific to our method: increasing the latent dimension makes disentanglement significantly harder in general, and we observe similar degradation trends across models (Fig.3).
>
> **Q1:** The recursive factorization used in the problem formulation is a general way to represent joint distributions and does not impose a restrictive structural assumption. Importantly, the identifiability results themselves do not rely on a specific topological ordering. The triangular construction is used as a convenient parametrization of dependencies, but the core of our theory lies elsewhere. In particular, Proposition 2 relies on the interplay between the local orthogonality of the mixing function and the global structural property captured by Assumption 2. From this perspective, the ordering is not fundamental: any joint distribution can be factorized in such a way, and different orderings are equally valid. Crucially, our framework does not assume that such an ordering is known, nor that it corresponds to a causal structure. This is precisely one of the motivations of our work: identifiability is achieved without relying on causal assumptions or a known generative order.
>
> **Q2 :** We appreciate the reviewer’s question regarding the practical relevance of the assumptions. These assumptions are not only theoretically motivated but also well-grounded in widely accepted principles and common practices in representation learning; we therefore view them as structural abstractions that enable studying identifiability beyond the classical independence setting.
> Regarding Assumption 1, orthogonality can be interpreted as a principled definition of what constitutes a meaningful latent "concept" by formalizing the idea that distinct factors should have decoupled functional effects on observations. This view is strongly aligned with the Independent Causal Mechanisms principle, which is a widely accepted foundation in causal inference and has been extensively studied in prior work. In this context, orthogonality can be seen as a functional instantiation of this principle. Moreover, related works have successfully leveraged orthogonality constraints on the Jacobian in practical settings (e.g., Wei et al., 2021), demonstrating that such assumptions are not only theoretically meaningful but also empirically impactful for realistic image data.
> Regarding Assumption 2, this condition formalizes the combinatorial richness of the latent factors, capturing the idea that each factor should, in principle, be independently manipulable. Many latent variables (e.g., position, orientation, lighting) naturally vary across bounded ranges, and datasets are often constructed to explore these variations. It is important to note, however, that real-world datasets may not fully satisfy this assumption. For instance, variables such as temperature and altitude may not span all possible combinations in observed data, even though such combinations are physically plausible. So the assumption should be interpreted as a property of the underlying generative domain, not a guarantee about any finite dataset. Even so, the assumption provides a critical theoretical foundation: it delineates when disentanglement is possible, by ensuring that distinct latent factors are identifiable in principle. In other words, while actual datasets may be incomplete, Assumption 2 highlights the necessary richness required to make independent degrees of freedom visible to the model, grounding identifiability results in realistic, physically plausible factor spaces.
>
> We hope these clarifications adequately address your concerns and look forward to your response.

---

> > ### Author Rebuttal · Reviewer_5uqT · 2026-04-02
> >
> > Thank the authors for the clarification. I keep my positive assessment.

---

### Official Review · Reviewer_hvD5 · 2026-03-12

**Soundness:** 2
**Presentation:** 3
**Significance:** 2
**Originality:** 3
**Overall Recommendation:** 4
**Confidence:** 4

**Summary:**

Instead of relying on the conventional statistical independence for disentanglement, this paper proposes new necessary conditions by defining a local constraint (orthogonality in the latent space) and a global constraint (combinatorial completeness of the latent domain). The authors provide theoretical proofs for both conditions and empirically validate them through qualitative and quantitative results on toy datasets. Furthermore, the paper provides a mathematical derivation explaining why likelihood-based models, such as VAEs, often fail to achieve proper disentanglement.

**Compliance With Llm Reviewing Policy:**

Affirmed.

**Final Justification:**

Thanks for sharing new links. My concerns have all been adequately addressed, and I have therefore increased my score.

**Key Questions For Authors:**

- Please address the points raised in the Weaknesses section.

**Limitations:**

Recent studies in the VAE framework have attempted to enforce orthogonality through specific objectives. It remains unclear whether these approaches fail for the same theoretical reasons discussed here. Although I have not reflected this in the final score, addressing this in future work would be a meaningful extension. It was a pleasure to read such an interestingly motivated paper.

**Strengths And Weaknesses:**

### Strengths

1. The paper defines the necessity of orthogonality from a structural perspective (local constraint) and provides mathematical verification. This offers a much clearer motivation compared to previous intuitive approaches and serves as a solid guide for modeling disentangled representations.

2. The global constraint for datasets is intuitively defined and supported by rigorous theoretical proofs.

3. The paper directly proves why disentanglement occurs through Equation (3). Moving beyond simple intuition to a direct theoretical derivation is a significant contribution.

---

### Weaknesses (Major Issues)

1. **Inconsistency with Claims**: While the authors mention "realistic cases" (lines 73 & 163), the experiments are limited to toy datasets. To support these claims, validation on standard benchmarks such as dSprites is essential. Expanding the experiments to 3D Shapes or MPI3D would further strengthen the paper's arguments.

2. **Absence of Disentanglement Metrics**: Although Section 5 shows the quality of representations, it lacks standard metrics used in the field. Results for Factor-VAE metric, MIG, SAP, and DCI are necessary for a fair comparison.

3. **Comparative Analysis with VAEs**: The authors attribute the failure of VAEs to the "mutual information" term in Eq. (5) (lines 395–396). However, models like $\beta$-TCVAE operate without this term, making their objective similar to Normalizing Flows (NF). This suggests that the failure might stem from the lack of invertibility (Assumption 1) rather than the objective function itself. A comparison with $\beta$-TCVAE is needed to clarify this distinction.

- **Note to Authors**: While the theoretical contribution is significant, the gap between the strong claims and the simplistic toy experiments is quite large. I am willing to raise the score if the empirical concerns are properly addressed.

---

### Minor Issues
- Line 148: "a uniquely" $\rightarrow$ "an uniquely".

- Line 402: "objective per se" (Please check the phrasing "what is the se").

- Notation/Alignment: Ensure all terms are clearly defined upon first use.

---

> ### Author Rebuttal · Authors · 2026-03-27
>
> **General message for reviewers**: I would like to sincerely thank all reviewers for their time and effort. I know the reviewing process can be demanding, and while it can sometimes be stressful or frustrating from the author’s side, I genuinely appreciate the depth and care reflected in these reviews.
> Even though it is always easier to receive shorter and more straightforward feedback, I found the level of engagement with the paper very encouraging. I truly appreciated that all reviewers seemed interested in the work and took the time to provide detailed and thoughtful comments. I’m especially grateful that the importance of the problem and significance of the work are recognized, and I welcome the challenge. I will do my best to address each point carefully below.
>
> **Q1 :** Thank you for raising this important point. We agree that the wording “realistic” may have been misleading, and we will revise it for clarity. Its use was strictly limited in the context of emphasizing that, unlike many prior works, we do not rely on statistical independence assumptions, which are often *unrealistic* in practice. Our primary objective is to validate the theoretical propositions under controlled conditions, rather than to benchmark performance on standard datasets. To rigorously test the theory, it is essential to ensure that all assumptions are satisfied and that no confounding factors are introduced. For this reason, we use invertible NFs and explicitly enforce the orthogonality condition by computing the Jacobian via PyTorch autograd (through vector-Jacobian products with basis vectors) and constraining it accordingly. While this allows us to precisely control the conditions and observe the effect of orthogonality, it is computationally expensive and does not scale to higher-dimensional image datasets. Importantly, efficiently enforcing orthogonal Jacobians in a scalable way remains an open research problem, beyond the scope of this work. However, as discussed in Sec.6, widely used models such as VAEs implicitly introduce a form of orthogonality bias. Their empirical success on datasets like dSprites, 3Dshapes or MPI3D can therefore serve as direct support of the practical relevance of our theoretical claims.
>
> **Q2 :** We thank the reviewer for this suggestion. Metrics such as FactorVAE, MIG, SAP, and DCI are widely used in the disentanglement literature. Our initial choice of Amari distance and MCC was deliberate and aligned with prior work (Gresele et al., 2021), where these metrics are commonly used and sufficient to assess whether latent factors are successfully recovered and disentangled. These metrics already provide a strong indication of whether the representation is disentangled in our setting, and an important question is whether additional metrics would lead to a different conclusion. To address this, we computed the suggested metrics within the limited time available for rebuttal. **New results** https://is.gd/ULwzEw are consistent with our findings and confirm that the learned representations are well disentangled.
> Overall, these additional evaluations support our conclusions and are coherent with the trends observed using Amari distance and MCC. We will include these metrics and a more detailed discussion in the final version of the paper.
>
> **Q3 :**
> We thank the reviewer for this insightful comment. We believe there is a slight misunderstanding of our claim, and we are happy to clarify.
> We do not attribute the failure of VAEs to the MI term. Our central claim is that disentanglement does not arise from MI regularization, but rather from the combination of (i) global independence (via the prior / total correlation) and (ii) local orthogonality (implicitly induced by the factorized posterior).
> The role of the MI term is different: it controls the amount of information stored in the latent variables.
> In this regard, the reviewer’s observation about β-TCVAE supports our claim. Indeed, β-TCVAE reduces or removes the MI term while retaining (i) the TC penalty and (ii) a factorized posterior. These models still produce disentangled representations, which is consistent with our explanation: disentanglement is driven by independence and orthogonality, not by MI minimization, and removing it can alleviate posterior collapse without affecting the mechanisms responsible for disentanglement.
> This perspective also clarifies the comparison with NFs. While NFs include the same TC pressure, they lack the implicit orthogonality constraint induced by the factorized posterior in VAEs, which explains why they typically fail to disentangle. Our framework shows that adding such an orthogonality constraint is sufficient to recover disentanglement in flows. It reinforces the view that disentanglement stems from independence and orthogonality, and can be transferred beyond VAEs once these inductive biases are made explicit.
>
> We hope these clarifications adequately address your concerns and look forward to your response.

---

> > ### Author Rebuttal · Reviewer_hvD5 · 2026-04-03
> >
> > Q-2 and Q-3 have been satisfactorily addressed. However, I still remain unconvinced regarding Q-1. I agree that recent studies suggesting that VAE-based models may exhibit an orthogonality bias provide a plausible motivation for the paper. That said, this serves primarily as motivation unless it is directly validated through experiments in the context of the present work. Without such empirical evidence, the argument remains suggestive rather than fully substantiated.
> >
> > I therefore believe that additional experimental validation is necessary. In particular, given that the current paper has only been evaluated in a highly limited setting, at minimum, validation on standard synthetic disentanglement datasets would be needed. If this concern is convincingly addressed, I would be open to increasing my score.
> >
> > ---
> > Response to 'Reply Rebuttal Comment by Authors': **Unable to Access the Anonymous Site**
> > > Thank you for sharing the anonymous link. **Unfortunately, the site does not appear to be accessible from my side**. If possible, I would greatly appreciate it if you could re-upload the additional experimental results through another anonymous platform, such as an anonymous GitHub repository. This would allow me to review the added results more carefully.

---

> > > ### Author Response · Authors · 2026-04-06
> > >
> > > We sincerely thank the reviewer for the follow-up and for acknowledging that Q2 and Q3 have been satisfactorily addressed. We appreciate the continued engagement on Q1 and have now generated results on dSprites and 3Dshapes, which indeed enrich the paper. We take this comment seriously and have made a substantial effort during the rebuttal phase to address it further as rigorously as possible.
> > >
> > > First, we would like to emphasize that our experimental protocol follows prior work (Gresele et al., 2021), where the evaluation is designed to validate theoretical results in controlled settings. In this context, toy datasets are commonly used and deemed sufficient, as they allow us to strictly enforce the assumptions required by the theory and avoid confounding factors. Nonetheless, to further address the reviewer’s concern, we provide additional empirical results both within our controlled framework and **on standard disentanglement datasets**
> > >
> > > ---
> > >
> > > ### Within our framework
> > >
> > > We provide **new results** [https://is.gd/6LbAJx] explicitly comparing VAEs and orthogonality-constrained NFs. The results are consistent with our claims: VAEs achieve stronger disentanglement than unconstrained NFs, likely due to their inherent orthogonality bias (as discussed in Sec.6), but perform worse than orthogonality-constrained flows, likely due to the information preference property. β-TCVAE obtains results closer to NFs, which is consistent with our explanation in Q3. These observations directly support our central claim that disentanglement arises from the combination of (i) global independence (via the prior) and (ii) local orthogonality (implicitly induced by the factorized posterior), thereby explaining the success of VAEs in a way that is both *theoretically grounded and empirically supported*.
> > >
> > > ---
> > >
> > > ### On standard disentanglement datasets
> > >
> > > We additionally provide new results on dSprites and 3DShapes. As explained previously, directly applying our constrained NF in this setting is challenging, since enforcing orthogonality requires explicit Jacobian computations, which are computationally expensive and do not scale to image data (beyond the scope of this paper). However, as highlighted in Sec.6, VAEs share the same objective as our constrained model (up to the MI term, which does not drive disentanglement) and implicitly encourage Jacobian orthogonality. They therefore provide a natural proxy to test our theory in higher-dimensional settings. Both VAEs and NFs impose independence through the TC term. In our manuscript, we considered models without orthogonality bias (NFs) and showed that adding this bias improves disentanglement. Here, we follow the *complementary approach*: we start from a model that includes both independence and orthogonality (a VAE), and we remove the orthogonality bias to evaluate its effect.
> > >
> > > The mechanism by which VAEs implicitly enforce Jacobian orthogonality has been analyzed in prior work (e.g., Reizinger et al.). In simplified terms, the factorized Gaussian posterior $z|x\sim\mathcal{N}(\mu(x),\sigma(x)I)$ enforces a row-orthogonal encoder Jacobian, and in the near-deterministic regime the decoder approximately inverts the encoder, resulting in a column-orthogonal decoder Jacobian. Based on this observation, we construct a variant of the β-VAE where we replace the diagonal Gaussian posterior with a NF posterior. This is a standard technique in the VAE literature to increase expressiveness and better match the prior, thereby improving density modeling (e.g., Kingma et al., 2016). However, unlike the factorized Gaussian case, this more expressive posterior no longer enforces a diagonal covariance structure, and thus removes the implicit Jacobian orthogonality constraint. Importantly, this change leaves the overall objective unchanged, particularly the TC/independence pressure imposed by the prior, thereby isolating the effect of removing the orthogonality bias while preserving the independence mechanism.
> > >
> > > We implement this model using disentanglement\_lib and reproduce standard β-VAE results on dSprites and 3DShapes. We then train the exact same model, with identical architecture and training procedure, but using a flow-based posterior (denoted β-flowVAE). **New results** are reported at [https://is.gd/GS9cmQ]. If orthogonality were not important, this more expressive model should yield improved disentanglement, since it better matches the prior hence improving independence. However, we observe a systematic substantial decrease in disentanglement performance. The key difference is precisely the removal of the orthogonality bias. This provides direct empirical evidence that independence alone is not sufficient, and that orthogonality plays a crucial role. Overall, these results confirm our theoretical claim on the sources of disentanglement. We believe these additional experiments address the reviewer’s concerns
> > >
> > > ---
> > >
> > > # New link : https://anonymous.4open.science/r/anonymous-supplementary-material-EAF2/

---

### Official Review · Reviewer_cicY · 2026-03-12

**Soundness:** 2
**Presentation:** 3
**Significance:** 3
**Originality:** 2
**Overall Recommendation:** 4
**Confidence:** 4

**Summary:**

This work considers the problem of unsupervised disentangled representation learning. Specifically, this translates to the problem of recovering a latent random vector through samples observed through a nonlinear function. The authors provide a condition under which identifiability w.r.t. the latent random vector can be claimed. This condition does not rely on stringent statistical assumptions. In contrast, it is expressed as a combination of pointwise properties that the Jacobian of a learned model has to satisfy and properties of the support of the latent distribution. The theoretical claims are also supported by experiments with synthetic results.

**Compliance With Llm Reviewing Policy:**

Affirmed.

**Final Justification:**

The rebuttal addressed most of my initial concerns, and several issues were clarified satisfactorily. There is an overall consensus that the paper is well written and mathematically sound, while most reviewers appear satisfied with the provided responses. Some of the suggestive arguments noted by both myself and reviewer hvD5 in the initial version have been addressed. Based on my understanding of the main results, I still believe the overall contribution is important. I also think that incorporating the additional experiments suggested in reviewer hvD5’s post-rebuttal comments, improving the positioning of certain arguments, and reorganizing the proof section would further strengthen the final version. Therefore, I increase my score to 4 (weak accept).

**Key Questions For Authors:**

Question 1: The proof section in the Appendix spans 16 pages but includes only two references. It is unclear which parts constitute original contributions by the authors and which are based on prior work. Are all of these details necessary to establish the main results? The current presentation and organization make it difficult for the reader to verify the validity of the claims, which are otherwise very interesting and potentially significant.

Question 2: Assumption 1 states an orthogonality requirement that model's Jacobian has to satisfy in a pointwise fashion (if I understand correctly). However, the paper provides very little detail on how this can be achieved in practice, or whether there exist any known models that can enforce this assumption. Even in the experimental section, the authors present it as a straightforward process, whereas in reality, it is far from trivial. Could the authors please elaborate on that?

Question 3: If I understand correctly, one can claim, based on your claims, that every smooth and invertible function can be "uniquely" (up to scaling and permutation) written as the composition of an invertible function where it’s Jabobian has orthogonal columns everywhere and a function consisting of univariate  invertible functions. Isn’t this a very strong claim to make? I would appreciate if the authors could provide further clarification here.

Question 4: In your experiments, how is the KL divergence computed as a metric? Wouldn’t it be intractable in the settings under consideration? Could the authors clarify the computation and any approximations used?

Question 5: Right below relation (8), the authors state: “Yet, in practice, flows remain notoriously nondisentangled.” Normalizing flows are a very popular approach and the learning objective actually encourages disentanglement through total correlation minimization. Has this phenomenon been observed and reported by others? This is an important point that supports your work, yet relevant references are missing. I would appreciate if the authors could provide supporting references.

Question 6: Based on Section 6, VAEs could also perform really well under some of the considered scenarios. However, they are not included as one of your baselines. How does you method compare to them in practice across all considered scenarios?

Typos and minor comments:

- Section 5, Model, second sentence “..Jacobians. We learn…”
- Many similar typos in the Appendix
- In some cases, it is possible to infer causal relations from observational data alone.
- In relation (1), the authors choose a very specific form for the latent density. Since the expression is valid for any density function by the chain rule, it is unclear why this particular form is highlighted as being of special interest.

**Limitations:**

Yes

**Strengths And Weaknesses:**

The main paper is very well written and a pleasure to read. It is well structured, balances technical detail with clear explanations, and provides strong intuition behind the theoretical claims. However, verifying these claims requires going through 16 pages of proofs, which include only two references to prior work. It is not clear which parts of these proofs are original and which are adapted from existing results. To improve clarity and accessibility, the authors should explicitly separate prior theoretical results from their novel contributions. Portions that closely follow earlier work could be replaced with concise references, allowing the reader to focus on the paper’s original advances. This would make the theoretical contributions more transparent and easier to assess, while reducing redundancy in the presentation.

I am not fully convinced of the validity of the proposed claims. Some aspects of the experimental and discussion sections also raise potential contradictions. While the theoretical claims are clearly stated, it is unclear how they are implemented in practice. Additionally, the authors present the imposition of the considered Jacobian-based structure as straightforward, but in my view, this is far from trivial. That said, the experimental section is well organized, and the detailed experiments effectively demonstrate the strengths of the proposed approach and help the authors substantiate their claims. The same also holds for the discussion section.

The problem that the authors consider is a very important one. If the provided claims are valid, then this work is capable to advance understanding, capabilities, or practice in machine learning. This could be definitely achieved through opening the door to new methods, theory, and perspectives. Regarding the originality of this work, the authors state, in the main paper, that this work provides an extension of previous works, however the structure of their appendix does not make it easy to understand what are their actual contributions.

---

> ### Author Rebuttal · Authors · 2026-03-28
>
> **Given the strict 5000-character limit, this is a highly condensed answer. Please feel free to ask for details. See the general message for reviewers in hvD5.**
>
> **Q1:** We thank the reviewer for this comment. We agree that the proof section is quite long and could perhaps be segmented for readability. We aimed to keep it concise, but due to the complexity of the problem, a shorter proof did not emerge. Regarding references, we only included two because most derivations are original mathematical work we developed for this problem. While we cannot completely rule out similar arguments may exist elsewhere, our approach was constructed from scratch specifically for our claims.
>
> **Q2:** Thank you for raising this point, you are indeed correct that imposing an orthogonal Jacobian in practice is far from trivial. There are previous works exploring *approximate* enforcement, e.g.,Wei et al., 2021. However, in our case, the goal was to validate the propositions, so we had to ensure the condition was *effectively enforced*. Specifically, we used invertible NFs and imposed orthogonality by recovering the Jacobian via PyTorch autograd with vector-Jacobian products, feeding in basis vectors, and then enforcing the orthogonality constraint. While computationally expensive, this allowed us to clearly observe the effect of the orthogonality condition and validate the theory. In practice, efficiently enforcing orthogonal Jacobians in a scalable way is still an open research question, one that goes beyond the scope of this paper. However, as discussed in Sec.6, our work highlights that finding such methods would be a very promising direction that would directly leverage our theory.
>
> **Q3:** We appreciate this question and would like to clarify a potential misconception. Our claim is not that every smooth and invertible function admits such a decomposition, this would be much stronger than our result and is generally false. Our claim is instead the following. Assuming that latent factors influence observations in distinct ways, formalized through Jacobian orthogonality, and that the learned function belongs to the same functional family as the true generative process, the mapping *between learned and true latent variables* can be expressed as the composition of a function with orthogonal Jacobian and a set of coordinate-wise invertible functions. Importantly, this does not imply that both functions decompose this way. Rather, this characterizes how the learned model inverts the true generative process within the shared functional family.
>
> **Q4:** In our experiments, we have access to the generating function and true latent factors. Using PyTorch autograd, we can compute the Jacobian and its log-determinant, yielding the true data distribution. For the model, the NF provides a tractable density. We then compute the KL divergence between the true and learned distributions in the standard way. This applies to all models and allows us to evaluate whether they fit the data distribution accurately. We emphasize that this evaluation is not intended to measure disentanglement, but purely the goodness of fit.
>
> **Q5:** The reviewer makes a good point, and we agree the original wording could be refined to better reflect the nuance (e.g., “do not naturally yield...”). The original phrasing stems from our strong familiarity with flow models. Disentanglement is not an inductive bias of standard flows trained purely with maximum likelihood. Empirically, canonical flows (RealNVP, Glow, Residual, TAR) do not report or exhibit disentangled latent representations. When disentanglement appears, it typically arises from introducing additional structure (e.g., GLOWin, SCFlow, StyleFlow) rather than from vanilla likelihood training. This contrasts with VAEs, which are widely used for disentanglement in their standard form and exhibit a natural bias toward factorized representations. This is in fact the main point of Sec.6. To our knowledge, our work is the first to explicitly highlight that NFs also contain a TC term in their objective, even though, as supported by the cited literature, they do not exhibit the same inherent bias toward disentanglement as VAEs.
>
> **Q6:** The primary goal of our experiments was to validate our theoretical results, not to establish NFs as SotA disentanglement models. Flows were used because they satisfy the conditions of the propositions. Since VAEs already incorporate an implicit orthogonal bias (Sec.6), including them was less central.
> Nonetheless, we ran a comparison for d=6 with independent latent variables, since the VAE prior imposes independent terms. **New results** https://is.gd/ULwzEw : As expected, VAEs achieve stronger disentanglement than unconstrained flows, likely due to their inherent orthogonal bias (as highlighted in Sec. 6), but perform worse than constrained normalizing flows, likely due to the information preference property and the detrimental MI term (also discussed in Sec. 6).

---

> > ### Author Rebuttal · Reviewer_cicY · 2026-04-02
> >
> > I would like to thank the authors for satisfactorily answering some of my questions, specifically Q2 and Q3. The current form of the proof section of the paper makes covering it in such a short time inaccessible. A more strategic structure of that section, allowing the reviewer to verify the validity of and the reader to follow your main claims, would be desired. The lack of such structure is a major disadvantage of this work.
> >
> > The response to my Q5 is not satisfying at all, as it covers many methods I did not ask about. My question was simple and directed, yet, based on the lack of provided references, I understand that this notorious claim has not been observed by others before. Regarding Q4, computing the KL divergence between the two distributions in the standard way required computing an integral, which in high dimensions is intractable. Please correct me if there is something I'm missing here. Finally, regarding Q6, I appreciate the additional experimental results. In my opinion, including VAE in your experiments strengthens the positioning of your work even more.

---

> > > ### Author Response · Authors · 2026-04-03
> > >
> > > We thank the reviewer for their constructive follow-up. We are pleased that they found the additional results valuable. We will incorporate them along with additional discussion in the final manuscript.
> > >
> > > We address the remaining points below.
> > >
> > > ---
> > >
> > > ### **On the Appendix structure**
> > >
> > > We thank the reviewer for this comment. We would like to clarify that the proof section presents novel, and complete derivations, which necessarily makes it long and complex. Nevertheless, it is not a single monolithic block, but is structured into four main sections:
> > >
> > > * Preliminaries (~4 pages proof)
> > > * Proposition 1 (~2 pages proof + ~1 page discussion sub section)
> > > * Proposition 2 (~1.5 pages proof + ~0.5 page discussion sub section)
> > > * Proposition 3 (~3 pages proof including >1 page of visual illustrations + ~1 page discussion on the 2D case)
> > >
> > > Each proposition is thus presented with its own self-contained proof and an accompanying discussion to improve interpretability. In the final version, we will further aid readability by adding a *brief “proof outline” at the beginning of each section*, summarizing the key ideas and steps.
> > >
> > > We also note that the overall length and level of detail are *consistent with standard theoretical work in disentanglement and causality*, where appendix proofs of comparable or greater length/complexity are common (e.g., Gresele et al., 2021; Ghosh et al., 2023; Zheng et al., 2022; Buchholz et al., 2022; Reizinger et al., 2022; Locatello et al., 2020). Several of these works contain substantially longer individual proofs (e.g., >7 pages for a single result with 26 pages total). In comparison, we believe our presentation is relatively compact given the technical scope.
> > >
> > > ---
> > >
> > > ### **On NFs and disentanglement**
> > >
> > > We apologize for the lack of clarity in our previous response. **Short answer:** To the best of our knowledge, there are **no references showing that vanilla NFs yield disentangled representations**, and this is precisely the point, such models are not used for disentanglement in their standard form.
> > >
> > > More specifically, the literature does not contain works explicitly stating that “vanilla flows do not disentangle,” because research typically does not focus on documenting the absence of a property. Instead, the evidence is indirect but consistent:
> > >
> > > * No prior work reports disentangled representations from standard likelihood-trained flows (e.g., RealNVP, Glow).
> > > * When disentanglement is pursued with flows, it is always achieved by introducing additional structure (supervision, architectural constraints, or modified objectives).
> > > * In contrast, models like VAEs are routinely used for disentanglement in their vanilla formulation, which highlights a clear difference in inductive bias.
> > >
> > > Therefore, the absence of such references reflects an established empirical practice: vanilla NFs are not considered disentanglement models.
> > >
> > > Our contribution is to make this limitation of vanilla NF more explicit:
> > >
> > > * *theoretically*, by identifying the role of the TC term in the flow objective, and
> > > * *empirically*, as our experiments confirm that vanilla flows do not disentangle, whereas adding the proposed structural constraint (or using VAEs) leads to disentangled representations.
> > >
> > > We will revise the manuscript to state this more clearly and avoid suggesting that this is a widely documented empirical claim; rather, it is an *implicit consensus reflected in how models are used in the literature*, and now *explicitly demonstrated in our experiments*.
> > >
> > > ---
> > >
> > > ### **On KL divergence computation**
> > >
> > > As clarified in the rebuttal, in our setting (primarily defined to validate our theory), we have access to both the ground-truth distribution ($p$) and the learned model ($p_\theta$). The KL divergence is defined as:
> > > $$KL(p|p_\theta)=\mathbb{E}\_{x\sim p}[\log p(x)-\log p_\theta(x)].$$
> > >
> > > While this expectation is indeed an integral in principle, it is estimated in practice via Monte Carlo sampling. Concretely, since our dataset consists of N samples $\{x_i\}\_{i=1}^N$ drawn i.i.d. from $p$, we approximate:
> > > $$KL(p|p_\theta)\approx\frac{1}{N}\sum_{i=1}^N\log p(x_i)-\log p_\theta(x_i).$$
> > >
> > > Both terms are thus computed as empirical averages over the dataset. This estimation strategy is standard in the literature: likelihood-based generative models are routinely evaluated via empirical averages over samples. This is the same principle underlying maximum likelihood training and evaluation of NFs, as introduced in Rezende and Mohamed 2015 and reviewed in Papamakarios et al. 2021. Similarly, disentanglement works such as Gresele et al. 2021 follow the exact same evaluation pipeline. As $N \to \infty$, the empirical estimate converges to the true KL divergence. In practice, for sufficiently large datasets, this provides an accurate and widely accepted proxy for comparing model fit.
> > >
> > > ---
> > >
> > > We thank the reviewer again for their careful and constructive feedback, and believe these clarifications address the remaining concerns.

---

### Official Review · Reviewer_AFsQ · 2026-03-14

**Soundness:** 2
**Presentation:** 2
**Significance:** 3
**Originality:** 2
**Overall Recommendation:** 4
**Confidence:** 4

**Summary:**

This paper studies the identifiability of latent variables (also called components or sources) under a specific class of nonlinear mixing function, known as the orthogonal map. First, the authors showed that the latent variables are identifiable (up to trivial indeterminacies) if they are assumed to be independent and that at most one latent variable is Gaussian. Then, the authors extend it to the setting with dependent latent variables, showing that they are identifiable under certain structural assumption on the latent distribution; the paper states that this is a direct generalization of the first setting with independent latent variables. Experiments on synthetic data are given to validate the theory.

**Compliance With Llm Reviewing Policy:**

Affirmed.

**Final Justification:**

Most of my main concerns have been addressed by the rebuttal

**Key Questions For Authors:**

Comments or questions:
1. To help readers understand the key Assumption 2, the paper should provide some examples of dependent latent variables such that the assumption is satisfied. Given that causal representation learning (CRL) is one of the major ways to identify dependent latent variables, I would suggest providing examples in CRL setting under which Assumption 2 can be satisfied. For example, if the latent variables follow a structural equation model corresponding to additive noise model, linear Gaussian model, or linear non-Gaussian model, can Assumption 2 be satisfied?
2. Is this statement "if the factors are statistically independent Assumption 2 is fulfilled and thus this assumption is a direct generalization of the case of independent factors" in Section 4.3 correct? If so, then independent latent variables $\implies$ Assumption 2 is satisfied $\implies$ identifiability in the sense of Definition 1. However, this is not consistent with Proposition 1 which additionally requires the assumption that at most one latent variable is Gaussian. So my understanding is either that statement is not correct, or Proposition 2 is not correct in the sense that it misses the assumption that at most one latent variable is Gaussian.

Other comments or questions about connection with highly relevant prior works:
1. The same orthogonality assumption stated in Assumption 1 has previously been proposed and adopted by Gresele et al. (2021) and Buchholz et al. (2022). I am therefore surprised that these works are not mentioned or acknowledged in Section 3, which is largely devoted to explaining and motivating this assumption. As currently written, Section 3 gives the impression that the assumption is newly introduced in this work.
2. The paper should provide a detailed discussion of the connection (similarity and difference) of Proposition 1 with the result involving orthogonal influence in both Gresele et al. (2021) and Buchholz et al. (2022).
3. In these statements "We argue that what is missing is a notion of independence [...] l influence that each factor exerts on the observations" and "we propose an alternative perspective on what [...] their functional effects on the generative process.", the paper should clearly acknowledge those two works above.
4. In the statements "it can be formalized as an orthogonality [...] mapping latent factors to observations" and "While previous studies have [...] independence assumption" in the introduction, references should be provided here.
5. For the statement "we take a stronger stance: we argue that orthogonal influence is an intrinsic defining property of meaningful concepts" in the introduction, there is no sufficient explanation on how this stronger stance differs from  Gresele et al. (2021) and Buchholz et al. (2022) that leverage orthogonal influence with the goal of obtaining identifiability.

**Limitations:**

I did not manage to find a discussion of the limitation.

**Strengths And Weaknesses:**

Strengths:
- Identifiability of latent variables have received considerable attention in the past few years, e.g., in the literature of nonlinear ICA and causal representation learning. This paper improves upon the line of works on leveraging function classes to establish identify the latent variables, and further relax it to handle dependent latent variables.
- The paper is well written and easy to follow. The notations are clearly defined and the assumptions are stated clearly.
-  The theoretical results seem to be sound and rigorous (although I did not carefully verify each step of the proofs).

Weaknesses:
1. Lack of examples to illustrate the key assumption (Assumption 2), which makes it difficult to understand and also hard to evaluate how general/restrictive it is.
2. The connection with highly relevant prior works are not well discussed and acknowledged in many parts of the paper. Orthogonal influences have been adopted in several previous works, but many parts of the paper gives the impression that the assumption is newly introduced in this work.

See further elaborations of the above points in the "Key Questions For Authors" section.

---

> ### Author Rebuttal · Authors · 2026-03-29
>
> **Given the strict 5000-character limit, this is a highly condensed answer. Please feel free to ask for details. See the general message for reviewers in hvD5.**
>
> **Q1:** We thank the reviewer for this suggestion and agree that connecting to standard CRL is essential to clarify our scope. To address this, we emphasize a fundamental distinction: Asm.2 is a *dataset-level property of the data support*, rather than a property of an underlying *causal generative model*. Formally, the existence of a causal model does not, by itself, guarantee that the resulting dataset will be rich enough for disentanglement. To illustrate, consider two binary latent factors $Z_1\to Z_2$: If the causal mechanism is deterministic, some factor combinations never occur; in contrast, if the dependency is probabilistic, all combinations appear with non-zero probability, satisfying Asm.2. However, even when "natural" data lacks certain combinations, interventions, either explicit or through dataset design, could ensure the support required by Asm.2. This shows that Asm.2 can be satisfied in typical CRL settings, provided the dataset is sufficiently diverse.
>
> Crucially, this highlights that causal structure and Asm.2 are orthogonal: our work does not rely on any causal assumptions, but rather focuses on the *combinatorial richness of the dataset*, regardless of how the data was generated (causally or not). This formalizes the idea that each latent factor should correspond to a *distinct, manipulable degree of freedom* ("concept slider"). If two variables can never be varied separately in the observed data, they cannot be identified as separate entities without further supervision. Thus, while Asm.2 defines the conditions under which disentanglement is theoretically possible, it may not be achieved in every real-world scenario, particularly if the dataset is not sufficiently rich or lacks the necessary interventional samples to make these separate degrees of freedom visible to the disentanglement model.
>
> **Q2:** We thank the reviewer for this remark which highlights an important subtlety. While we clarify this in the Appendix, the wording should be revised for clarity. The key point is that Prop.1 and 2 operate under different regimes. Prop.1 is set in the classical nonlinear ICA regime with unbounded latent space, where identifiability comes from statistical structure. In that setting, Gaussian variables are rotationally invariant, explaining the need for the "Gaussian" limitation. By contrast, Prop.2 relies on a geometric constraint encoded in Asm.2, namely a bounded support with full combinatorial structure. This breaks the same rotational symmetries independently of any distributional assumption, and is sufficient for identifiability.
> Regarding the statement, it aims at observing that independence implies Asm.2 at *the level of support*. Provided one considers the natural setting where each latent variable has bounded support (e.g., physical quantities with natural bounds), independence implies a Cartesian product, which directly satisfies Asm.2. In this sense, Prop.2 can be seen as a genuine generalization of the independent setting: it recovers the independent case while extending identifiability to dependent latent variables *under the same structural support condition*. We will revise the wording to make this distinction explicit.
>
> **Positioning:** We thank the reviewer for the opportunity to clarify our "stronger stance" on functional orthogonality. We fully acknowledge these pioneering papers identified the role of Jacobian's orthogonality. It was never our intent to suggest this principle originates with our work. While these references appear multiple times, we agree that mentioning them earlier (Intro/Sec 3) better serves the reader, and will revise accordingly.
>
> Our contribution is a significant theoretical extension that goes beyond these works in several fundamental ways. Prior approaches primarily view orthogonality as a *regularizer* to recover statistically independent sources. In contrast, we argue orthogonality is a *defining property* of disentangled representations, regardless of the latent distribution. This shift allows us to move beyond the “Independent” in ICA. While prior results rely on statistical independence for identifiability, we show that it can still be achieved with dependent factors under structural conditions on the latent domain, broadening applicability. Additionally, these works focus on more restricted function classes (conformal maps or specific spurious solutions), whereas we consider a broader class characterized by orthogonal Jacobians. Our framework provides a unifying perspective: Prop.1 not only recovers the result of Th.4.12 in Gresele et al. as a specific case, but also explains it, showing that the underlying mechanism is functional orthogonality itself. Overall, our framework extends and subsumes these approaches, as special cases of a more general, structurally grounded theory.

---

> > ### Author Rebuttal · Reviewer_AFsQ · 2026-04-03
> >
> > Thanks for the detailed response. Most of my main concerns have been addressed. Please incorporates the comments in the revision, especially the connection with existing works and improving the wordings. I have updated my score.

---

### Decision · Program_Chairs · 2026-04-30

**Decision:**

Accept (regular)

**Comment:**

This paper studies unsupervised disentanglement in nonlinear representational learning $x=f(z)$ where $z$ may be correlated (so, e.g. more general than ICA). The authors use the notion of functional orthogonality to prove identifiability results in this model. This work is timely and relevant, and there is a consensus to accept this for presentation at the conference.